# Improved Multi-Strategy Aquila Optimizer for Engineering Optimization Problems

**DOI:** 10.3390/biomimetics10090620

**Published:** 2025-09-15

**Authors:** Honglin Kan, Yaping Xiao, Zhiliang Gao, Xuan Zhang

**Affiliations:** School of Artificial Intelligence, Anhui Polytechnic University, Wuhu 241000, China; kanhonglin@ahpu.edu.cn (H.K.); gaozhiliang@ahpu.edu.cn (Z.G.); zhangxuan@ahpu.edu.cn (X.Z.)

**Keywords:** Aquila Optimizer, random sub-dimension update mechanism, adaptive parameter, dynamic opposition-based learning

## Abstract

The Aquila Optimizer (AO) is a novel and efficient optimization algorithm inspired by the hunting and searching behavior of Aquila. However, the AO faces limitations when tackling high-dimensional and complex optimization problems due to insufficient search capabilities and a tendency to prematurely converge to local optima, which restricts its overall performance. To address these challenges, this study proposes the Multi-Strategy Aquila Optimizer (MSAO) by integrating multiple enhancement techniques. Firstly, the MSAO introduces a random sub-dimension update mechanism, significantly enhancing its exploration capacity in high-dimensional spaces. Secondly, it incorporates memory strategy and dream-sharing strategy from the Dream Optimization Algorithm (DOA), thereby achieving a balance between global exploration and local exploitation. Additionally, the MSAO employs adaptive parameter and dynamic opposition-based learning to further refine the AO’s original update rules, making them more suitable for a multi-strategy collaborative framework. In the experiment, the MSAO outperform eight state-of-the-art algorithms, including CEC-winning and enhanced AO variants, achieving the best optimization results on 55%, 69%, 69%, and 72% of the benchmark functions, respectively, which demonstrates its outstanding performance. Furthermore, ablation experiments validate the independent contributions of each proposed strategy, and the application of MSAO to five engineering problems confirms its strong practical value and potential for broader adoption.

## 1. Introduction

Optimization problems, characterized by high nonlinearity, multimodality, and large-scale search spaces, have long posed significant challenges across scientific and engineering domains due to the proliferation of local optima. In practical applications, increasing complexity and dynamism demand algorithms with enhanced performance and adaptability. Consequently, the development of efficient and highly adaptable optimization algorithms has become a key to improving the efficiency of the system and the quality of decision making.

In recent years, metaheuristic algorithms have demonstrated unique advantages in handling complex large-scale problems by combining global exploration with local exploitation. They have been successfully applied in various domains, including computational intelligence and data mining [1,2,3,4,5,6], transportation [7], task planning [8], resource management [9,10], and UAV path planning [11,12]. (1) In computational intelligence and data mining, Manoharan et al. [1] introduced an enhanced weighted K-means Grey Wolf Optimizer to enhance clustering performance. Tang et al. [2] developed EDECO, an Enhanced Educational Competition Optimizer [13] that uses distribution-based replacement and dynamic distance balancing for efficient numerical optimization. Jia et al. [3] improved the Slime Mould Algorithm (SMA) [14] with Compound Mutation Strategy (CMS) and Restart Strategy (RS) for feature selection. Djaafar Zouache et al. [4] drew inspiration from quantum computing to combine the Firefly Algorithm (FA) [15] and Particle Swarm Optimization (PSO) [16] for feature selection tasks. (2) In transportation, Fang et al. [7] proposed the Discrete Wild Horse Optimizer (DWHO) to solve the Capacitated Vehicle Routing Problem (CVRP) [17]. (3) In task planning, Zhong et al. [8] proposed a novel Independent Success History Adaptation Competitive Differential Evolution (ISHACDE) algorithm to address functional optimization problems and Space Mission Trajectory Optimization (SMTO). (4) In resource management, Abd et al. [9] presented an improved White Shark Optimizer [18] to optimize hybrid photovoltaic, wind turbine, biomass and hydrogen storage systems. Zhang et al. [10] proposed MFSMA, a Multi-Strategy Slime Mould Algorithm, for optimal microgrid scheduling. (5) In UAV path planning, Xu et al. [11] introduced DBO-AWOA, an Adaptive Whale Optimization Algorithm that combines chaotic mapping, nonlinear convergence factors, adaptive inertia mechanisms, and dung beetle-inspired reproduction behaviors for 3D route planning. Liang et al. [12] developed the Improved Spider-Bee Optimizer (ISWO) for UAV 3D path planning.

In general, metaheuristic algorithms offer a rich toolkit to solve complex optimization challenges by emulating natural phenomena, animal behaviors, physical laws, and social processes. These algorithms fall into evolution-based, physics-based, swarm-based, and human-based categories [19].

Evolution-based algorithms draw inspiration from Darwin’s theory of natural selection and genetic inheritance. Prominent examples include the Genetic Algorithm (GA) [20], Differential Evolution (DE) [21], Genetic Programming (GP) [22], and Evolution Strategies (ES) [23]. Among these, the GA and DE are particularly popular due to their strong adaptability and straightforward implementation.

Physics-based algorithms are derived from various phenomena and principles in physics. Simulated Annealing (SA) [24] mimics the annealing process in solids to achieve global search, the Sine–Cosine Algorithm (SCA) [25] uses the periodic oscillations of trigonometric functions to balance exploration and exploitation, Young’s Double-Slit Experiment (YDSE) Optimizer [26] is inspired by the interference patterns observed in Young’s double-slit experiment, and the Sinh Cosh Optimizer (SCHO) [27] leverages the mathematical properties of hyperbolic sine and hyperbolic cosine functions to drive its exploration and exploitation mechanisms.

Human-behavior-based algorithms simulate educational or social activities to drive optimization. Teaching–Learning-Based Optimization (TLBO) [28] models the influence mechanism between teachers and learners, the Attraction–Repulsion Optimization Algorithm (AROA) [29] emulates the balance between attractive and repulsive forces observed in nature, using this mechanism as the basis for its search process, Differentiated Creative Search (DCS) [30] balances divergent and convergent thinking within a team-based framework, fostering a continuously learning and adaptive optimization environment, and the Football Team Training Algorithm (FTTA) [31] draws on the training routines and tactical drills of football teams to formulate its optimization strategy.

Swarm-based algorithms are inspired by collective foraging and collaboration behaviors in nature. Particle Swarm Optimization (PSO) [16] emulates bird flocking and foraging, the Whale Optimization Algorithm (WOA) [32] mimics the bubble net feeding strategy of humpback whales, the Grey Wolf Optimizer (GWO) [33] draws on the hierarchical pack hunting of grey wolves, and the Griffon Vulture Optimization Algorithm (GVOA) [34] models the collective intelligent foraging behavior of griffon and vulture species in nature.

Abualigah et al. [35] are the first to formalize the “high-altitude cruise and dive hunting” strategy of Aquila into a novel metaheuristic optimization algorithm, known as the Aquila Optimizer (AO). By alternating between global cruise and precise diving behaviors, the AO achieves rapid convergence while maintaining a simple and easy-to-implement structure. Its outstanding performance in numerous benchmark tests has made AO a research focal point and led to its widespread adoption for solving complex optimization problems. However, the AO exhibits several limitations: It tends to converge prematurely, becoming trapped in local optima and failing to sufficiently explore the global search space, which can result in suboptimal solutions. Its convergence speed in complex high-dimensional problems can be slow, requiring more iterations to reach the best results, and its scalability is limited, making it challenging to address large-scale optimization tasks with many variables and constraints [36]. To address these issues, researchers have proposed various improved variants of the AO in different optimization domains, including hybrids with other techniques and novel movement strategies. Compared to the original AO, these enhanced algorithms demonstrate superior adaptability and performance in a broader range of complex problems. Serdar Ekinci et al. [37] developed an enhanced AO algorithm (enAO) by innovatively integrating an improved Opposition-Based Learning (OBL) mechanism [38] with the Nelder–Mead (NM) simplex search method [39]. Kujur et al. [40] introduced a Chaotic Aquila Optimizer tailored for demand response scheduling in grid-connected residential microgrid (GCRMG) systems. Pashaei et al. [41] proposed a Mutated Binary Aquila Optimizer (MBAO) that incorporates a Time-Varying Mirrored S-shaped (TVMS) transfer function. Designed as a new wrapper-based gene selection method, MBAO aims to identify the optimal subset of informative genes. Inspired by the superior performance of the Arithmetic Optimization Algorithm (AOA) [42] and the Aquila Optimizer (AO), Zhang et al. [43] proposed a hybrid algorithm that integrates the strengths of both, denoted AOAAO.

Although various AO variants have addressed some of the shortcomings of the original AO, there are still issues that need to be resolved. Xiao et al. [44] introduced an enhanced hybrid metaheuristic, IHAOAVOA, which combines the strengths of the AO and the African Vultures Optimization Algorithm (AVOA) [45]. Although IHAOAVOA demonstrates marked improvements over both the AO and AVOA on various benchmark functions, its increased computational cost remains a potential drawback, and there is still room for performance gains on certain benchmark functions. To accelerate convergence, Wirma et al. [46] integrated chaotic mapping into the AO and employed a single-stage evolutionary strategy to balance exploration and exploitation. However, compared to other AO variants, this approach incurs additional computational overhead and memory usage for chaotic variables, resulting in higher CPU time and resource demands. Zhao et al. [47] developed a multiple-update mechanism employing heterogeneous strategies to accelerate the late-stage convergence of the AO and enhance its overall performance. However, this approach exhibits suboptimal results when applied to real-world engineering problems. Zhao et al. [48] simplified the Aquila Optimizer to enhance its convergence speed, introducing a simplified Aquila Optimizer (IAO). In benchmark functions, the IAO outperforms the original AO and other comparison algorithms. However, its performance remains suboptimal when applied to real-world engineering problems. This suggests that, although the AO algorithm and its variants have undergone continual refinement and perform excellently on benchmark functions, they still exhibit limitations: They often underperform on real-world engineering problems, and their performance on certain test functions leaves room for further improvement. Furthermore, we observe that many AO variants still exhibit performance bottlenecks when tackling high-dimensional optimization problems.

To address the shortcomings of the AO and AO variants, this paper proposes MSAO, a multi-strategy enhanced variant designed to overcome the premature convergence to local optima and its degraded performance in high-dimensional search spaces. In summary, the MSAO introduces three key innovations: the incorporation of a random sub-dimension update mechanism to effectively decompose and explore high-dimensional search spaces; the integration of the memory strategy and dream-sharing mechanism of the DOA, allowing individuals to retain historical elite information, achieving a coordinated balance between global exploration and local exploitation; and the adoption of adaptive parameter control and dynamic opposition-based learning to refine the original AO’s update rules, thus accelerating convergence and improving solution precision. The main contributions of this work can be summarized as follows:A random sub-dimension update mechanism is incorporated to effectively decompose high-dimensional search spaces. The memory strategy and dream-sharing strategy from the DOA are fused to enable individual memory retention and inter-population information sharing, achieving a coordinated balance between global exploration and local exploitation.Adaptive parameter and dynamic opposition-based learning are integrated to further refine the original AO update rules within its multi-strategy framework.In the CEC2017 benchmark suite, the MSAO significantly outperforms the original AO and seven other state-of-the-art algorithms. Ablation studies confirm the independent contributions of each enhancement, and outstanding results in five real-world engineering optimization cases underscore the practical applicability.

## 2. Related Work

### 2.1. Overview of the Original AO

Abualigah et al. introduced the Aquila Optimizer (AO), a swarm-based algorithm inspired by the predatory behavior of Aquila raptors. Aquila employs four hunting modes: a high-altitude vertical dive for broad reconnaissance; level gliding at mid-altitudes for localized exploration; a low-altitude gradual descent to refine the search in promising zones; and ground-level stalking and capture using subtle perturbations to fine-tune local prey detection. Based on predation strategies, the AO divides the search process into five phases: initialization, expanded exploration, narrowed exploration, expanded exploitation, and narrowed exploitation. Phase transitions are governed by the iteration index *t* relative to the maximum *T*: When the current iteration $t\leq \frac{2}{3}\times T$, the algorithm prioritizes exploration; otherwise, it switches to the exploitation phase. This design ensures robust global search in early iterations and enhanced local search precision later, thereby achieving an effective balance between convergence speed and solution quality.

In all mathematical expressions of this algorithm, vector variables are shown in boldface. The symbol “×” denotes multiplication between scalars or between a scalar and a vector and the symbol “∗” denotes element-wise multiplication between vectors.

#### 2.1.1. Initialization

The Aquila Optimizer employs a population-based search strategy. Initially, a set of candidate solutions *X* is randomly generated within the upper and lower bounds (UB and LB) [35]. During each iteration, the best solution identified so far is updated and regarded as the current global best.(1)$$X=\left[\begin{array}{c} X_{1}\\ X_{2}\\ \vdots \\ X_{i}\\ \vdots \\ X_{N} \end{array}\right]=\left[\begin{array}{cccccc} x_{1,1} & x_{1,2} & \ldots & x_{1,j} & \ldots & x_{1,Dim}\\ x_{2,1} & x_{2,2} & \ldots & x_{2,j} & \ldots & x_{2,Dim}\\ \vdots & \vdots & \ddots & \vdots & \ddots & \vdots \\ x_{i,1} & x_{i,2} & \ldots & x_{i,j} & \ldots & x_{i,Dim}\\ \vdots & \vdots & \vdots & \ddots & \vdots & \vdots \\ x_{N,1} & x_{N,2} & \ldots & x_{N,j} & \ldots & x_{N,Dim} \end{array}\right]$$
where *X* denotes the set of current candidate solutions, which are randomly generated within the upper and lower bounds by Equation (Equation 2). $X_{i}$ represents the decision vector (position) of the *i*th solution, *N* is the total number of candidate solutions (population size), and Dim indicates the dimensionality of the problem.(2)$$x_{i,j}=\mathrm{rand}\times \left(UB_{j}-LB_{j}\right)+LB_{j},\;i=1,2,\ldots ,N,\;j=1,2,\ldots ,\mathrm{Dim}$$
where rand denotes a random number in the interval [0,1], and $LB_{j}$ and $UB_{j}$ represent the lower and upper bounds of the *j*th decision variable, respectively.

#### 2.1.2. Expanded Exploration

During the expanded exploration phase, the algorithm mimics Aquila’s broad search behavior during high-altitude vertical dives to perform a coarse scan of the entire search space. Its mathematical formulation is as follows [35].(3)$$X\left(t+1\right)=X_{\mathrm{best}}\left(t\right)\times \left(1-\frac{t}{T}\right)+\left(X_{M}\left(t\right)-X_{\mathrm{best}}\left(t\right)\times \mathrm{rand}\right)$$
where X(t+1) denotes the new solution generated by the expanded exploration strategy at iteration t+1. $X_{best}\left(t\right)$ is the best individual found so far. $X_{M}\left(t\right)$ is the mean position of all individuals in the population at iteration *t*, defined as(4)$$X_{M}\left(t\right)=\frac{1}{N}\sum_{i=1}^{N}X_{i}\left(t\right)$$
where *N* denotes the size of the population.

#### 2.1.3. Narrowed Exploration

In the narrowed exploration phase, the algorithm emulates Aquila’s short-range contour gliding around promising regions to perform fine-grained reconnaissance and encirclement of high-potential solution areas [35]. Its mathematical formulation is as follows [35].(5)$$X\left(t+1\right)=X_{\mathrm{best}}\left(t\right)*\mathrm{Levy}\left(D\right)+\left(X_{R}\left(t\right)+\left(y-x\right)\times rand\right)$$
where X(t+1) denotes the new solution generated in iteration t+1 by the narrowed exploration strategy. $X_{R}\left(t\right)$ is an individual randomly selected from the population and the variables *x* and *y* define the spiral trajectory used during the search, as specified in Equation (Equation 6).(6)$$\begin{array}{cccccc} x & =r*\sin(\theta ),\; & y & =r*\cos(\theta )\\ \mathrm{where},\\ r & =r_{1}+U\times D_{1},\; & \theta & =-\omega \times D_{1}+\theta_{1},\; & \theta_{1} & =\frac{3\pi }{2} \end{array}$$

The parameter $r_{1}$ takes integer values from 1 to 20 to determine the fixed number of search cycles. *U* is a small constant set to 0.00565. $D_{1}$ is an integer that ranges from 1 to the dimensionality of the search space. ω is a small constant fixed at 0.005.

Furthermore, Equation (Equation 5) incorporates the Lévy flight distribution function Levy(D), which is defined as follows: (7)$$\mathrm{Levy}\left(D\right)=s\times \frac{u\times \sigma }{{\left|v\right|}^{\frac{1}{\beta }}}$$(8)$$\sigma =\frac{\tau \left(1+\beta \right)\times \sin\left(\frac{\beta \pi }{2}\right)}{\tau \left(\frac{1+\beta }{2}\right)\times \beta \times 2^{\left(\frac{\beta -1}{2}\right)}}$$
where Levy(D) denotes the Lévy flight distribution function in a space of *D*. *s* = 0.1 is a constant. *u* and *v* are random numbers uniformly distributed in [0,1]. β = 0.5 is the exponent parameter of the Lévy distribution, and ρ is the scale parameter of the Lévy distribution.

#### 2.1.4. Expanded Exploitation

In the expanded exploitation phase, Aquila emulates a low-altitude gradual descent behavior: Individuals progressively approach the target region and perform precise strikes through slow dive maneuvers, thereby intensifying the in-depth exploitation of high-quality solution areas [35]. Its mathematical formulation is as follows [35].(9)$$X\left(t+1\right)=\left(X_{\mathrm{best}}\left(t\right)-X_{M}\left(t\right)\right)\times \alpha -rand+\left(\left(\mathrm{UB}-\mathrm{LB}\right)\times rand+\mathrm{LB}\right)\times \delta$$
where X(t+1) denotes the new solution generated in iteration t+1 by the expanded exploitation strategy. α and δ serve as fixed parameter settings to adjust the search behavior.

#### 2.1.5. Narrowed Exploitation

In the narrowed exploitation phase, the algorithm emulates Aquila’s stalking and capture behavior to perform fine-grained localized encirclement of high-quality solution regions [35]. Its mathematical formulation is as follows [35].(10)$$X\left(t+1\right)=QF\times X_{\mathrm{best}}\left(t\right)-\left(G_{1}\times X\left(t\right)\times rand\right)-\left(G_{2}\times \mathrm{Levy}\left(D\right)+rand\times G_{1}\right)$$
where X(t+1) denotes the new solution generated in iteration t+1 by the narrowed exploitation strategy. The quality function QF and the adjustment functions $G_{1}$ and $G_{2}$ are defined as follows: (11)$$QF\left(t\right)=t^{\frac{2\times rand-1}{{\left(1-T\right)}^{2}}},\;G_{1}=2\times rand-1,\;G_{2}=2\times \left(1-\frac{t}{T}\right)$$

#### 2.1.6. Algorithm Procedure

Algorithm 1 details the complete execution workflow of the AO algorithm.

The algorithm begins by randomly generating *N* individuals in the search space and initializing the iteration counter *t* = 1. Then, it evaluates the fitness of all individuals and logs the best global solution $X_{\mathrm{best}}$ to guide subsequent position updates. It dynamically alternates among four behavioral strategies: during the first $\frac{2}{3}$ of iterations (exploration phase), High Soar with Vertical Stoop (Expanded Exploration, Equation (Equation 3)) for broad global sampling and Contour Flight with Short Glide (Narrowed Exploration, Equation (Equation 5)) to refine the search region; and in the final $\frac{1}{3}$ of the iterations (exploitation phase), Low Flight with Slow Descent (Expanded Exploitation, Equation (Equation 9)) for fine-grained local search around promising areas and Swoop and Grab Prey (Narrowed Exploitation, Equation (Equation 10)) to converge on and polish the best solutions. After each update, boundary handling ensures feasibility. If the termination criterion is met, the algorithm returns the final global best $X_{\mathrm{best}}$. Otherwise, *t* increases and the cycle repeats.
**Algorithm 1:** Pseudo-code of AO
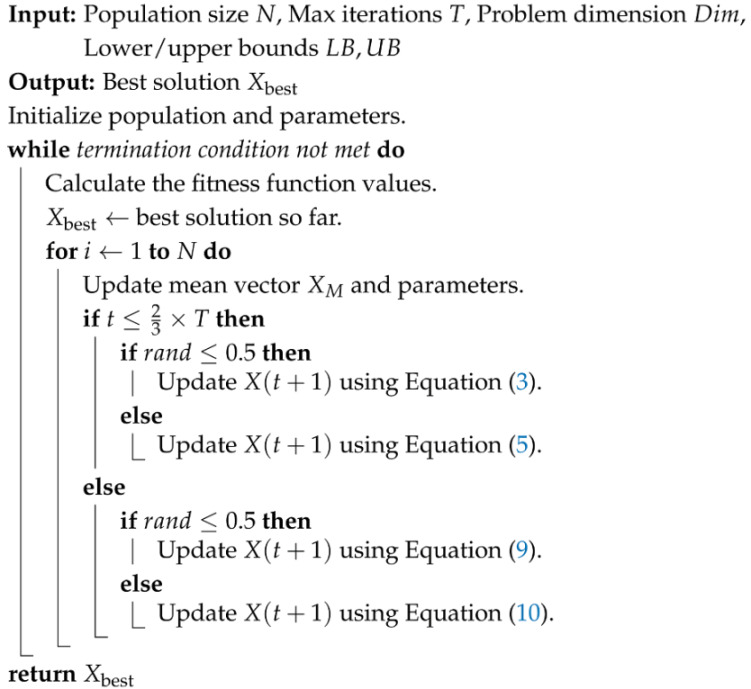


The two parameters used in this algorithm, 2/3 and 0.5, are adopted directly from the original AO literature [35] without modification. In particular, the parameter $\frac{2}{3}$ governs the transition point between the exploration and exploitation phases: When the current iteration $t\leq \frac{2}{3}\times T$, the algorithm prioritizes exploration. Otherwise, it switches to the exploitation phase. This design maintains robust global search in the early iterations and enhances local search precision in later stages, thereby balancing convergence speed and solution quality. The parameter 0.5 serves as a randomness threshold, allowing a 50% probability of switching between the two strategies, which ensures an even execution of exploration and exploitation.

### 2.2. Overview of the DOA

Since the MSAO draws on the Dream Optimization Algorithm (DOA), we provide a brief overview of the DOA. Lang et al. [49] introduced the DOA, inspired by the human dreaming process, which involves memory retention, forgetting, and the logical organization of ideas. These features parallel the cycle of exploration and exploitation in metaheuristic optimization. The DOA incorporates a forgetting and supplementation strategy that ensures a balance between broad search and focused refinement, and a dream-sharing strategy that helps the method overcome local traps.

As depicted in Figure 1, the DOA operates in two phases: exploration and exploitation. During the exploration phase, the algorithm divides the population into five subpopulations and applies the memory strategy to reset each subpopulation’s individuals to their current best position. Then, it employs the forgetting and supplementation strategy, which combines global and local search by selectively removing old solutions and introducing new ones. Additionally, the dream-sharing strategy is activated in this phase to enhance escape from local optima. In the exploitation phase, no further subpopulation partitioning occurs. The algorithm first applies the memory strategy to reset all individuals to the current best global position and then proceeds with the forgetting and supplementation strategy alone, omitting the dream-sharing strategy. In Section 3.4, we will provide a detailed comparison of the algorithmic workflows of the DOA and MSAO.

## 3. The Proposed MSOA

This section presents the Multi-Strategy Aquila Optimizer (MSAO), developed to address the original AO’s limitations in convergence accuracy and its propensity to become trapped in local optima when solving high-dimensional optimization problems. To achieve this, we integrate several critical enhancements within the AO framework to markedly strengthen its global exploration and local exploitation capabilities. Each of these improvements will be described in detail in the following sections.

In all mathematical expressions of this algorithm, vector variables are shown in boldface. The symbol “×” denotes multiplication between scalars or between a scalar and a vector and the symbol “∗” denotes element-wise multiplication between vectors.

### 3.1. Exploration Phase

During the exploration phase, the MSAO implements a random sub-dimension update mechanism by partitioning the population into five subpopulations. Initially, the each positions of subpopulation positions are reinitialized to their best individual position from the previous iteration using the memory strategy. Then, for each subpopulation *q* (q=1,2,3,4,5), $k_{q}$ dimensions are randomly selected from the set 1,2,…,Dim, denoted as $K_{1}$, $K_{2}$, …, $K_{k_{q}}$. For each individual $X_{i}$, only the components in these $k_{q}$ dimensions are updated according to the enhanced exploration rule, while all other dimensions remain unchanged. This procedure is applied sequentially across individuals, ensuring that all subpopulations collaboratively explore their respective subspaces within a single iteration, thus increasing global search diversity and accelerating convergence.

#### 3.1.1. Memory Strategy

The memory strategy, originating from the DOA [49], is defined mathematically in Equation (Equation 12). For any individual $X_{i}$ in subpopulation *q*, its position at the beginning of iteration t+1 is reset to the best solution found by that subpopulation in iteration *t*.(12)$$X_{i}^{t+1}=X_{\mathrm{bestq}}^{t}$$
where $X_{i}^{t+1}$ denotes the position of the *i*th individual in generation t+1. $X_{bestq}^{t}$ represents the best individual in the *q*th subpopulation during generation *t*.

#### 3.1.2. The Improved Expanded Exploration

To accelerate convergence and enhance solution accuracy, an adaptive convergence parameter η is introduced during the expanded exploration phase. This parameter enables gradual decay in the early stages of iteration, facilitating broad global exploration [50]. As the iteration progresses, the decay rate increases, promoting faster convergence. In contrast, the original AO’s convergence parameter may lead to premature convergence. Based on this nonlinear factor, the improved update formula for the expanded exploration is presented in Equation (Equation 14).(13)$$\eta ={\left(1-\frac{t}{T}\right)}^{\frac{2t}{T}}$$(14)$$x_{i,j}^{t+1}=\eta \times x_{bestq,j}^{t}+\left(x_{M,j}^{t}-x_{bestq,j}^{t}\times \mathrm{rand}\right),\;j=K_{1},K_{2},\ldots ,K_{k_{q}}$$
where $x_{i,j}^{t+1}$ denotes the value of the *i*th individual in the *j*th dimension at iteration t+1. $x_{bestq,j}^{t}$ represents the value of the best individual in group *q* in the *j*th dimension at iteration *t*. $x_{M,j}^{t}$ is the mean position of the population in the *j*th dimension at iteration *t*.

#### 3.1.3. The Improved Narrowed Exploration

To better align the narrowed exploration with the multi-strategy framework, this study replaces the original random perturbation mechanism with a dynamic opposition-based learning strategy [51], thus more effectively guiding the algorithm’s exploration and exploitation in the search space. The control factor for this strategy is defined in Equation (Equation 15), the dynamic opposition-based learning formula is given in Equation (Equation 16), and the resulting improved narrowed exploration update rule is presented in Equation (Equation 17).(15)$$\lambda =\exp\left(-\tan^{2}\left(\frac{1.2\;t}{T}\right)\right)$$(16)$$X_{o}=\lambda \times \left(\mathrm{UB}+\mathrm{LB}-X_{i}^{t}\right)$$(17)$$x_{i,j}^{t+1}=x_{bestq,j}^{t}\times Levy{\left(D\right)}_{j}+\left(x_{O,j}^{t}+\left(y_{j}-x_{j}\right)\times rand\right),\;j=K_{1},K_{2},\ldots ,K_{k_{q}}$$
where $x_{O,j}^{t}$ denotes the position in the *j*th dimension at iteration *t* obtained by the dynamic opposition-based learning strategy.

#### 3.1.4. Dream-Sharing Strategy

The dream-sharing strategy, derived from the DOA [49], enables each individual to randomly “borrow” positional information from other members of its subpopulation along selected dimensions, thereby enhancing the algorithm’s ability to escape local optima. This strategy is executed in parallel with the improved exploration update rules. Its mathematical formulation is given by(18)$$x_{i,j}^{t+1}=\left\{\begin{array}{cc} x_{m,j}^{t+1}, & m\leq i\\ x_{m,j}^{t}, & i<m\leq N \end{array}\right.\;j=K_{1},K_{2},\ldots ,K_{k_{q}}$$

### 3.2. Exploitation Phase

During the exploitation phase, the MSAO continues to employ the random sub-dimension update mechanism without subpopulation partitioning. Specifically, the algorithm first reinitializes all individual positions to that of the best solution from the previous iteration using the memory strategy. It then updates the selected sub-dimensions of each individual according to the enhanced exploration update rule, while keeping the other dimensions unchanged, thereby preserving global alignment while enabling localized fine-grained exploitation.

#### 3.2.1. Memory Strategy

During the exploitation phase, each individual $X_{i}$ has its position at the beginning of the iteration t+1 reset to the best global solution from iteration *t*. Its mathematical formulation is as follows [49].(19)$$X_{i}^{t+1}=X_{\mathrm{best}}^{t}$$
where $X_{best}^{t}$ represents the best individual during generation *t*.

#### 3.2.2. The Improved Expanded Exploitation

To accelerate iterations in high-dimensional settings, we streamline the expanded exploitation update strategy and introduce an adaptive parameter ω. This enhancement maintains a minimalist design of the parameter, while improving the numerical stability and high-dimensional handling [50]. The definition of ω is provided in Equation (Equation 20), and the refined expanded exploitation update rule is given in Equation (Equation 21).(20)$$\omega =\frac{1}{2}\times \left(\cos\left(\pi \times \frac{t}{T}\right)+1\right)$$(21)$$x_{i,j}^{t+1}=x_{best,j}^{t}+\left(\left(UB_{j}-LB_{j}\right)\times rand+LB_{j}\right)\times \omega ),\;j=K_{1},K_{2},\ldots ,K_{k_{q}}$$

#### 3.2.3. The Improved Narrowed Exploitation

The update strategy for the narrowed exploitation phase remains largely unchanged, with the addition of a random sub-dimension update mechanism to enhance high-dimensional performance. Its mathematical formulation is as follows: (22)$$x_{i,j}^{t+1}=QF\times x_{best,j}^{t}-\left(G_{1}\times x_{i,j}^{t}\times rand\right)-\left(G_{2}\times Levy{\left(D\right)}_{j}+rand\times G_{1}\right),j=K_{1},K_{2},\ldots ,K_{k_{q}}$$

### 3.3. Parameters Setting

The remaining key parameter settings for the MSAO are as follows: The threshold for switching between the exploration and exploitation phases is critical to overall performance, and the default value of the original AO is unsuitable. Therefore, in subsequent experiments, we conduct a sensitivity analysis to identify the optimal threshold that maximizes the benefits of prior empirical knowledge. In this study, the phase switching parameter is set to 0.6. $k_{q}$ and $k_{r}$ denote the number of randomly selected sub-dimensions in the exploration and exploitation phases, respectively. Their calculation formulas are given in Equations (Equation 23) and (Equation 24).(23)$$k_{q}=\mathrm{randi}\left(\left\lceil \frac{Dim}{8\times q}\right\rceil ,\max\left\{2,\left\lceil \frac{Dim}{3\times q}\right\rceil \right\}\right),\;q=1,2,3,4,5$$(24)$$k_{r}=\mathrm{randi}\left(2,\max\left\{2,\left\lceil \frac{Dim}{3}\right\rceil \right\}\right)$$

The parameter *u* is used to balance the enhanced exploration update rule and the dream-sharing strategy during the exploration phase. If rand < *u*, the algorithm applies the forgetting-and-replenishment mechanism; otherwise, it executes the dream-sharing strategy. Following the empirical setting in the DOA, *u* is set to 0.9.

### 3.4. The Detail of MSAO

The pseudo-code for the proposed MSAO algorithm is presented in Algorithm 2, and its flowchart is illustrated in Figure 2.
**Algorithm 2:** Pseudo-code of MSAO 
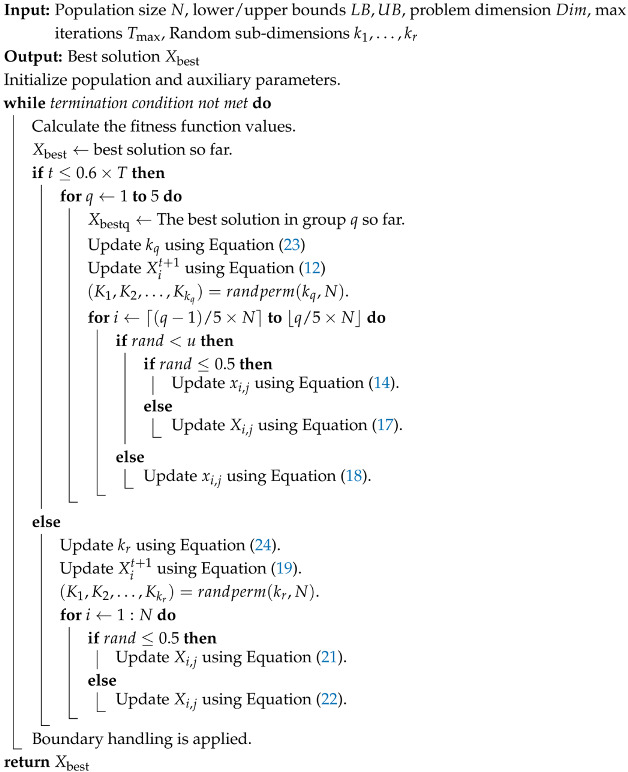


The algorithm begins by randomly generating *N* individuals within the search space and evenly dividing them into five subpopulations to ensure initial diversity and enable parallel local searches. The iteration counter *t* is initialized to 1. The fitness of all individuals is then evaluated and the best global solution $X_{\mathrm{best}}$ is recorded to guide the subsequent memory strategy and positional updates. During the first 60% of the iterations (exploration phase), the algorithm extracts the best solution of each subpopulation $X_{\mathrm{bestq}}$ and applies the memory strategy to reset all individuals in that subpopulation to their best solution, thereby rapidly exploiting promising regions. Then, it randomly switches between the expanded exploration update strategy (Equation (Equation 14)) and the narrowed exploration update strategy (Equation (Equation 17)) to balance global and local search and activates the dream-sharing mechanism (Equation (Equation 18)) to enhance diversity through inter-information exchange between subpopulations. In the remaining 40% of the iterations (exploitation phase), the algorithm focuses on the best global solution $X_{\mathrm{best}}$, using the memory strategy (Equation (Equation 19)) to pull all individuals back into its neighborhood and fully utilize global information. It selects between the expanded exploitation update strategy (Equation (Equation 21)) and the narrowed exploitation update strategy (Equation (Equation 22)) based on a random probability, flexibly adjusting step sizes for a finer-grained local search. After position update, boundary handling ensures feasibility. If the termination criterion is met, the final best global solution $X_{\mathrm{best}}$ is returned. Otherwise, *t* is incremented and the process repeats.

Figure 1 and Figure 2 illustrate that both the DOA and MSAO divide the algorithmic workflow into exploration and exploitation phases, with the MSAO adopting the DOA’s memory strategy and dream-sharing strategy to enhance performance. However, they differ markedly in their primary update procedures. The DOA relies primarily on the forgetting and supplementation strategy in both phases. In contrast, the MSAO adopts two distinct update rules in each phase. During the exploration phase, it utilizes Expanded Exploration and Narrowed Exploration, while in the exploitation phase, it applies Expanded Exploitation and Narrowed Exploitation. This richer set of phase-specific update mechanisms further enhances search efficiency and solution accuracy.

### 3.5. Complexity Analysis of MSAO

The time complexity of the MSAO is determined by the population size *N*, initialization, fitness evaluations and individual updates, the number of iterations *T*, and the dimensionality of the problem *D*. Initializing the population incurs O(N) complexity, whereas the fitness evaluation complexity depends on the specific problem and is not detailed here. Individual updates involve O(T×N×D) for position updates and O(T×N) for other per-generation operations. Hence, the overall time complexity of MSAO can be expressed as: O(N)+O(T×N)+O(T×N×D)=O(T×N×(D+1)).

## 4. Experimental Results and Analysis

In this experiment, the MSAO is evaluated on 29 benchmark functions of the CEC2017, excluding the unstable F2 function to ensure consistency of the result [52], detailed in Table 1. All benchmark functions are defined in the search space [−100, 100], and $f_{\min}$ denotes the theoretical minimum value of each test function. To assess adaptability across problem scales, we conduct experiments in 10, 30, 50, and 100 dimensions, with a fixed population size of 30 and a maximum of 1000 iterations. Each configuration is independently run 30 times to evaluate reliability. The MSAO is compared with its five AO variants and six state-of-the-art metaheuristic algorithms: the original AO [35] and its enhanced variants LOBLAO [53], TEAO [54], SGAO [55], MMSIAO [56], and IAO [57]; the classic WOA [32] and HHO [58]; the novel AOO [59] and DOA [49]; and the CEC-winning LSHADE [60]. The algorithm parameters are set according to original sources or widely accepted values, detailed in Table 2, and all methods use the same population size and iteration count to ensure a fair and rigorous comparison.

### 4.1. Sensitivity Analysis

This section presents a systematic sensitivity and performance evaluation of the phase-switching threshold between exploration and exploitation. This threshold critically determines the allocation of computational effort between early global search and later local refinement, affecting the convergence speed and final precision [61]. To investigate its impact on the MSAO’s performance, we test threshold values of 0.5, 0.6, 0.7, 0.8, and 0.9, corresponding to 50%, 60%, 70%, 80%, and 90% of the total iteration. The results are summarized in Table 3, with the best performing thresholds for each dimension highlighted in bold. The “w/t/l” stands for win/tie/loss. For example, the “2/6/21” in the first row and second column of the table indicates that with the phase-switch threshold set to 0.5 in the 10D CEC2017 tests, the algorithm outperformed all competitors on 2 functions, tied on six, and underperformed on 21 of the 29 benchmark functions.

The data reveal that increasing the threshold from 0.5 to 0.6 yields a marked performance improvement, whereas further increases lead to degradation. Although both 0.6 and 0.7 perform well, 0.6 offers the most stable and overall superior performance across dimensions. Consequently, we set the phase switching threshold to 0.6 to achieve an optimal balance between the two phases.

### 4.2. Ablation Experiments

To validate the individual contributions of each enhancement in the MSAO, we conduct ablation experiments. The MSAO comprises three key components: the memory strategy combined with dream-sharing strategy, the random sub-dimension update mechanism, and the tailored update rules that leverage these mechanisms. We created multiple variants of the algorithm by selectively including or excluding each component to assess their respective impacts on performance and to examine their synergistic effects. The results show that omitting any single strategy degrades performance, highlighting that these three components work cooperatively to achieve the MSAO’s optimal performance. Table 4 summarizes the combinations of strategies for each variant, where “∘” indicates inclusion and “×” indicates exclusion of a given component.

Figure 3, Figure 4, Figure 5 and Figure 6 present heatmaps of Friedman test rankings for these variants. In these heatmaps, darker cells correspond to better average rankings and, thus, superior performance. In Figure 3 and Figure 4, the original AO shows the lightest cells, while each single-strategy variant exhibits darker cells, indicating performance improvements over the AO. Two-strategy combinations yield even darker cells, whereas the full three-strategy integration, the MSAO, shows the darkest cells, denoting the best performance. This shows that the three strategies synergize in more than an additive fashion, mutually reinforcing each other under the complex characteristics of different test functions to substantially enhance global exploration and local exploitation. Similar trends are observed when the dimensionality increases to 50 and 100, confirming that the strategy combination remains highly effective in high-dimensional settings. In summary, analysis of cell darkness (average Friedman rankings) leads to three conclusions: (1) Any single strategy surpasses the original AO, (2) multi-strategy fusion further elevates performance, and (3) the fully integrated three-strategy MSAO achieves the top results, validating the effectiveness of the multi-strategy cooperative optimization framework.

### 4.3. Qualitative Analysis

To further validate the MSAO, we perform qualitative analyses on unimodal, simple multimodal, hybrid, and composition functions from the CEC2017. The results are shown in Figure 7.

Four visualization metrics are used to intuitively assess performance: search history, average fitness curve, first-dimensional trajectory, and the convergence curve of the best candidate solution. In the search history plots, red dots denote global optima and blue dots indicate the best solution found at each iteration. These polts show that the MSAO explores effectively, using its memory strategy for rapid convergence, demonstrating strong global exploration and local development capabilities. The average fitness curve exhibits a pronounced steep decline in the early phase, indicating that the population, leveraging the memory strategy, dream-sharing strategy, and enhanced position-update mechanisms, rapidly escapes local optima and converges efficiently. In the mid-to-late stages, the curve levels off, demonstrating that the population has locked onto a global or near-global optimum region and, aided by the random sub-dimension update mechanism and refined update rules, performs fine-grained local searches to produce high-quality solutions. The first-dimension trajectory illustrates position oscillations: The initial rapid oscillations reflect the random sub-dimension update mechanism swiftly identifying the global optimum region, and subsequent reduced oscillations indicate the fine-tuned search. The convergence curves confirm that the MSAO quickly reduces fitness values early on and stabilizes at high-quality solutions, further validating its efficient performance.

### 4.4. Performance Analysis of MSAO and AO’s Variants in CEC2017

In this section, we compare the MSAO against five AO variants on the CEC2017 to verify the MSAO’s performance. Table 5, Table 6, Table 7 and Table 8 summarize the results for 10D, 30D, 50D, and 100D.

Across the four dimensions (10D, 30D, 50D, and 100D), the MSAO achieves the lowest average fitness on most of the benchmark functions: 25 (86%) in 10D, 22 (76%) in 30D, 23 (79%) in 50D, and 28 (97%) in 100D. Furthermore, the MSAO’s average rankings consistently rank first at 1.34, 1.28, 1.34, and 1.03 in these dimensions, followed by the LOBLAO (average rankings consistently rank second at 2.34, 2.41, 2.48, and 2.17 in these dimensions) and the TEAO (average rankings consistently rank third at 3.41, 3.72, 3.48, and 3.48 in these dimensions). Given their strong performance and status as recent AO’s variants, the LOBLAO and TEAO will serve as representative AO variants for comparative evaluation in subsequent experiments. Furthermore, the MSAO exhibits lower standard deviations in all dimensions, indicating a more stable convergence. In summary, the MSAO not only achieves superior average fitness across most test functions but also demonstrates higher stability, outperforming current state-of-the-art AO variants.

### 4.5. Performance Analysis of MSAO in CEC2017

In this section, we provide a comprehensive analysis of the MSAO’s performance compared to eight leading optimization algorithms in the CEC2017 benchmark suite. Table 9, Table 10, Table 11 and Table 12 summarize results for 10D, 30D, 50D, and 100D problems. Figure 8 shows the convergence curves and Figure 9 shows the box plots [62].

In 10D and 30D (Table 9 and Table 10), the MSAO achieves the best average fitness ranking in 16 (55%) and 20 (69%) of the 29 benchmark functions, respectively. Specifically, for unimodal functions (F1–F3), the MSAO ranked first in F1 in 10D and in F3 in 30D. For simple multimodal functions (F4–F10), it dominated with a win rate of 86% in 10D, which slightly decreased to 57% in 30D. For hybrid functions (F11–F20), its win rate increases from 30% in 10D to 90% in 30D. For composition functions (F21–F30), the MSAO achieved 60% wins in 10D and 70% in 30D. For more than half of low-dimensional problems, the MSAO not only finds lower objective values than all competing algorithms but also maintains consistency with smaller standard deviations, demonstrating its exceptional ability to quickly and reliably locate high-quality solutions in low-dimensional search spaces. In 50D and 100D in higher dimensions (Table 11 and Table 12), the MSAO achieves the best average fitness in 20 (69%) and 21 (72%) of the 29 benchmark functions, respectively. In 50D, the MSAO ranks first on F5, F6, F7, F8, and F10 (71%) in the simple multimodal functions. For hybrid functions (F11 – F20), it takes the top spot in F12, F13, F15, F16, F17, F18, F19, and F20 (80%). In composition functions (F21–F30), the MSAO wins in F21, F22, F23, F24, F26, F29, and F30(70%). Although it does not claim first place in the unimodal functions F1 and F3, it secures second on F1. In 100D, the win rates remain 71% in simple multimodal and 70% in composition functions, while hybrid functions rise to 90%. This outstanding result in high dimensions, in which it wins nearly two thirds of all test functions, demonstrates that the integration of three strategies significantly enhances global search capability in high dimensions. Moreover, the MSAO shows lower standard deviations than the original AO and other competitors in these experiments, confirming its exceptional stability and robustness in high dimensional optimization problems.

As shown in Figure 8, the MSAO’s convergence curves in 10D, 30D, 50D, and 100D exhibit markedly steeper declines, faster convergence speeds, and lower fitness values than its counterparts. In the initial phase, the MSAO’s rapid descent is based on its memory strategy. By retaining and leveraging elite solutions from previous iterations, the algorithm quickly targets high-potential regions and avoids aimless exploration. Concurrently, the random sub-dimension update mechanism breaks free from full-dimensional search constraints, maintaining robust global exploration in high-dimensional landscapes and enabling rapid escape from local optima. In the mid-to-late stages, the MSAO’s curve levels off with significantly reduced oscillations compared to other methods, indicating that near convergence, its refined update rules, such as adaptive parameter control and dynamic opposition learning, effectively shrink step sizes for precise local exploitation, thus securing superior solutions and preventing premature convergence. In Figure 9, boxes are generally lower, narrower, and more concentrated, further confirming their superiority in consistency and stability of the results.

In summary, the MSAO not only significantly outperforms the original AO and other algorithms on low-dimensional optimization problems but also demonstrates exceptional performance and robustness in high-dimensional optimization problems. The experiments confirm that the memory strategy combined with dream-sharing strategy, the random sub-dimension update mechanism, and the tailored update rules that leverage these mechanisms effectively address the limitations of the original AO in high-dimensional search, offering an efficient and reliable solution for complex, large-scale optimization challenges.

### 4.6. Nonparametric Test Analysis

In this section, we perform nonparametric statistical analyses [63] (Wilcoxon rank sum test and Friedman test) to compare the performance of the MSAO against other algorithms. Table 13, Table 14, Table 15 and Table 16 present the results of the Wilcoxon test in various dimensional settings, showing statistically significant differences compared to the original AO and other competing algorithms (*p* < 0.05). The statistics of “wins/ties/losses” (w/t/l) overwhelmingly favor the MSAO, which generally achieves 27 to 29 wins (93% to 100%) against eight competing algorithms in 29 benchmark functions; even its lowest win count is 16 (55%), with only 0 to 7 ties and, at most, three losses, further corroborating its superior performance in most benchmark functions. Table 17, Table 18, Table 19 and Table 20 report the Friedman test results, showing that the MSAO’s average ranking mainly ranges from 1.4 to 2.4, while the runner-up LSHADE’s ranking mainly ranges from 2.7 to 4.4. Nonparametric statistical analyses confirm that the MSAO delivers outstanding optimization capability under the selected test conditions.

### 4.7. Engineering Optimization Experiments

To further assess the practical applicability of the MSOA, we compare it with other algorithms on five representative engineering design problems: the tension/compression spring design problem [64], pressure vessel design problem [65], three-bar truss design problem [64], welded beam design problem [66], and speed reducer design problem [67]. These engineering design problems have been widely adopted as benchmarks for evaluating optimization algorithms [68,69,70]. Each engineering design problem comprises an objective function to be minimized (typically weight or cost), subject to several nonlinear constraints. A smaller objective value indicates a better design and, consequently, corresponds to a better rank. Through comprehensive testing and the performance evaluation of these real-world engineering cases, the exceptional performance of the MSAO in engineering optimization is validated. The results are presented in Table 21, Table 22, Table 23, Table 24 and Table 25.

#### 4.7.1. Tension/Compression Spring Design Problem

The tension/compression spring design problem aims to minimize the weight of the spring. It involves four nonlinear constraints and three design variables: the wire diameter $x_{1}$, mean coil diameter $x_{2}$, and number of active coils $x_{3}$. The mathematical model is defined as follows: (25)$$\begin{array}{cc} \; & \mathrm{Minimize}:\\ \; & \;f\left(x\right)=x_{1}^{2}x_{2}\left(2+x_{3}\right)\\ \; & \mathrm{Subject}\;\mathrm{to}:\\ \; & \;g_{1}\left(x\right)=1-\frac{x_{2}^{3}x_{3}}{71785x_{1}^{4}}\leq 0\\ \; & \;g_{2}\left(x\right)=\frac{4x_{2}^{2}-x_{1}x_{2}}{12566(x_{2}x_{1}^{3}-x_{1}^{4})}+\frac{1}{5108x_{1}^{2}}-1\leq 0\\ \; & \;g_{3}\left(x\right)=1-\frac{140.45x_{1}}{x_{2}^{2}x_{3}}\leq 0\\ \; & \;g_{4}\left(x\right)=\frac{x_{1}+x_{2}}{1.5}-1\leq 0\\ \; & \mathrm{With}\;\mathrm{bounds}:\\ \; & \;0.05\leq x_{1}\leq 2.00\\ \; & \;0.25\leq x_{2}\leq 1.30\\ \; & \;2.00\leq x_{3}\leq 15.0 \end{array}$$

Table 21 presents the optimization results for the tension/compression spring design problem. The MSAO achieves the best optimal value of 0.012671 for the tension/compression spring design problem, ranking first. It is closely followed by LSHADE (0.012681, second place) and the AOO (0.012721, third place), with all three algorithms showing strong search capabilities for this problem. In contrast, the original AO (0.015784, eighth place), LOBLAO (0.015593, seventh place), and TEAO (0.015830, ninth place) perform poorly, primarily because they struggle to balance the three design variables while satisfying four nonlinear strength and deformation constraints. Although the DOA (0.014085, sixth place) and HHO (0.013924, fifth place) show improvements over them, they still fall short of the MSAO’s performance.

#### 4.7.2. Pressure Vessel Design Problem

The pressure vessel design problem aims to minimize the total manufacturing cost, which comprises welding, material, and forming expenses. The problem involves four constraints and four design variables: shell thickness $z_{1}$, head thickness $z_{2}$, inner radius $x_{3}$, and vessel length excluding heads $x_{4}$. The mathematical model is defined as follows: (26)$$\begin{array}{cc} \; & \mathrm{Minimize}:\\ \; & \;f\left(x\right)=1.7781z_{2}x_{3}^{2}+0.6224z_{1}x_{3}x_{4}+3.1661z_{1}^{2}x_{4}+19.84z_{1}^{2}x_{3}\\ \; & \mathrm{Subject}\;\mathrm{to}:\\ \; & \;g_{1}\left(x\right)=0.00954x_{3}\leq z_{2}\\ \; & \;g_{2}\left(x\right)=0.0193x_{3}\leq z_{1}\\ \; & \;g_{3}\left(x\right)=x_{4}\leq 240\\ \; & \;g_{4}\left(x\right)=-\pi x_{3}^{2}x_{4}-\frac{4}{3}\pi x_{3}^{3}\leq -1296000\\ \; & \mathrm{Where}:\\ \; & \;z_{1}=0.0625x_{1}\\ \; & \;z_{2}=0.0625x_{2}\\ \; & \mathrm{With}\;\mathrm{bounds}:\\ \; & \;10\leq x_{4},x_{3}\leq 200\\ \; & \;1\leq x_{2},x_{1}\leq 99 \end{array}$$

Table 22 presents the optimization results for the pressure vessel design problem, in which the MSAO achieves the best optimal value of 5821.851, ranking first. It is followed by LSHADE (5848.727, second place) and the AOO (5892.490, third place). In contrast, the WOA (5950.671, fourth place), LOBLAO (6282.705, fifth place), and TEAO (6307.177, sixth place) demonstrate reasonable search capability, but still exhibit shortcomings and underperform overall. The HHO (6755.636, seventh place) and DOA (7209.205, eighth place) adopt conservative boundary handling, leading to suboptimal design variables. The original AO performs the worst (7644.994, ninth place), struggling to finely optimize material thickness and dimensional requirements within the feasible region.

#### 4.7.3. Three-Bar Truss Design Problem

The three-bar truss design problem, originating in civil engineering, features a complex feasible region defined by stress constraints. The primary objective is to minimize the total weight of the truss members, subject to three nonlinear inequality constraints based on member stresses, resulting in a linear objective function optimization. The mathematical formulation is as follows:(27)$$\begin{array}{cc} \; & \mathrm{Minimize}:\\ \; & \;f\left(x\right)=l\left(x_{2}+2\sqrt{2}x_{1}\right)\\ \; & \mathrm{Subject}\;\mathrm{to}:\\ \; & \;g_{1}\left(x\right)=\frac{x_{2}}{2x_{2}x_{1}+\sqrt{2}x_{1}^{2}}p-\sigma \leq 0\\ \; & \;g_{2}\left(x\right)=\frac{x_{2}+\sqrt{2}x_{1}}{2x_{2}x_{1}+\sqrt{2}x_{1}^{2}}p-\sigma \leq 0\\ \; & \;g_{3}\left(x\right)=\frac{1}{x_{1}+\sqrt{2}x_{2}}p-\sigma \leq 0\\ \; & \mathrm{Where},\\ \; & \;l=100,\;p=2,\;\mathrm{and}\;\;\sigma =2\\ \; & \mathrm{With}\;\mathrm{bounds}:\\ \; & \;0\leq x_{1},\;x_{2}\leq 1 \end{array}$$

Table 23 presents the optimization results for the three-bar truss design problem, in which the MSAO achieves the best optimal value of 263.8958 for the three-bar truss design problem, ranking first. LSHADE (263.8972, second place) and the AOO (263.8972, third) follow closely. Mid-level performers include the WOA (263.8973, fourth place), DOA (263.8998, fifth place), and TEAO (263.9626, sixth place), which yield similar results but exhibit slightly less precision in exploring the constraint compared to the top three. The HHO (263.9721, seventh place), AO (264.0865, eighth place), and LOBLAO (264.2021, ninth place) demonstrate higher optimal values due to less refined adjustments near the stress limit constraints.

#### 4.7.4. Welded Beam Design Problem

The welded beam design problem aims to minimize the manufacturing cost of a welded beam. It features five constraints and four design variables that define the weld and beam geometry: weld thickness $x_{1}$, beam height $x_{2}$, beam length $x_{3}$, and beam width $x_{4}$. The mathematical formulation is as follows:(28)$$\begin{array}{cc} \; & \mathrm{Minimize}:\\ \; & \;f\left(x\right)=0.04811x_{3}x_{4}\left(x_{2}+14\right)+1.10471x_{1}^{2}x_{2}\\ \; & \mathrm{Subject}\;\mathrm{to}:\\ \; & \;g_{1}\left(x\right)=x_{1}-x_{4}\leq 0\\ \; & \;g_{2}\left(x\right)=\delta \left(x\right)-\delta_{max}\leq 0\\ \; & \;g_{3}\left(x\right)=P\leq P_{c}\left(x\right)\\ \; & \;g_{4}\left(x\right)=\tau_{max}\geq \tau \left(x\right)\\ \; & \;g_{5}\left(x\right)=\sigma \left(x\right)-\sigma_{max}\leq 0\\ \; & \mathrm{Where},\\ \; & \;\tau =\sqrt{\tau^{\prime 2}+\tau^{\prime \prime 2}+2\tau^{\prime }\tau^{\prime \prime }\frac{x_{2}}{2R}},\;\tau^{\prime }=\frac{P}{\sqrt{2}x_{2}x_{1}},\;\tau^{\prime \prime }=\frac{RM}{J},\\ \; & \;M=P\left(\frac{x_{2}}{2}+L\right),\\ \; & \;R=\sqrt{\frac{x_{2}^{2}}{4}+{\left(\frac{x_{1}+x_{3}}{2}\right)}^{2}},\;J=2\left(\sqrt{2}x_{1}x_{2}\left(\frac{x_{2}^{2}}{4}+{\left(\frac{x_{1}+x_{3}}{2}\right)}^{2}\right)\right),\\ \; & \;\sigma \left(x\right)=\frac{6PL}{x_{4}x_{3}^{2}},\;\delta \left(x\right)=\frac{6PL^{3}}{Ex_{3}^{2}x_{4}},\;P_{c}\left(x\right)=\frac{4.013Ex_{3}x_{4}^{3}}{6L^{2}}\left(1-\frac{x_{3}}{2L}\sqrt{\frac{E}{4G}}\right),\\ \; & \;L=14\;\mathrm{in},\;P=6000\;\mathrm{lb},\;E=30\times 10^{6}\;\mathrm{psi},\;\sigma_{max}=30,\;000\;\mathrm{psi}\\ \; & \;\tau_{max}=13,\;600\;\mathrm{psi},\;G=12\times 10^{6}\;\mathrm{psi},\;\delta_{max}=0.25\;\mathrm{in}\\ \; & \mathrm{With}\;\mathrm{bounds}:\\ \; & \;0.1\leq x_{3},x_{2}\leq 10\\ \; & \;0.1\leq x_{4}\leq 2\\ \; & \;0.125\leq x_{1}\leq 2 \end{array}$$

Table 24 presents the optimization results for the welded beam design problem, in which the MSAO achieves the lowest manufacturing cost of 1.692794 for the welded beam design problem, ranking first. It is followed by LSHADE (1.695725, second place) and the WOA (1.697121, third place). The AOO (1.826056, fourth place) demonstrates reasonable search capability, but still exhibits shortcomings and underperforms overall. The HHO (2.231482, fifth place) and LOBLAO (2.324208, sixth place) exhibit a similar problem, resulting in higher costs. The DOA (2.340397, seventh place) and AO (2.414442, eighth place) handle constraint boundaries conservatively, leading to suboptimal cost solutions. The TEAO performs worst (2.827475, ninth place), as its fixed parameter update rules struggle with the precise adjustments required in a complex, multi-constraint environment.

#### 4.7.5. Speed Reducer Design Problem

The speed reducer design problem involves the structural optimization of a gearbox for a small aircraft engine. Its mathematical formulation is given by(29)$$\begin{array}{cc} \; & \mathrm{Minimize}:\\ \; & \;f\left(x\right)=0.7854x_{2}^{2}x_{1}\left(14.9334x_{3}-43.0934+3.3333x_{3}^{2}\right)\\ \; & \;+0.7854\left(x_{5}x_{7}^{2}+x_{4}x_{6}^{2}\right)-1.508x_{1}\left(x_{7}^{2}+x_{6}^{2}\right)+7.477\left(x_{7}^{3}+x_{6}^{3}\right)\\ \; & \mathrm{Subject}\;\mathrm{to}:\\ \; & \;g_{1}\left(x\right)=-x_{1}x_{2}^{2}x_{3}+27\leq 0\\ \; & \;g_{2}\left(x\right)=-x_{1}x_{2}^{2}x_{3}^{2}+397.5\leq 0\\ \; & \;g_{3}\left(x\right)=-x_{2}x_{6}^{4}x_{3}x_{4}^{-3}+1.93\leq 0\\ \; & \;g_{4}\left(x\right)=-x_{2}x_{7}^{4}x_{3}x_{5}^{-3}+1.93\leq 0\\ \; & \;g_{5}\left(x\right)=10x_{6}^{-3}\sqrt{16.91\times 10^{6}+{\left(745x_{4}x_{2}^{-1}x_{3}^{-1}\right)}^{2}}-1100\leq 0\\ \; & \;g_{6}\left(x\right)=10x_{7}^{-3}\sqrt{157.5\times 10^{6}+{\left(745x_{5}x_{2}^{-1}x_{3}^{-1}\right)}^{2}}-850\leq 0,\\ \; & \;g_{7}\left(x\right)=x_{2}x_{3}-40\leq 0\\ \; & \;g_{8}\left(x\right)=-x_{1}x_{2}^{-1}+5\leq 0\\ \; & \;g_{9}\left(x\right)=x_{1}x_{2}^{-1}-12\leq 0\\ \; & \;g_{10}\left(x\right)=1.5x_{6}-x_{4}+1.9\leq 0\\ \; & \;g_{11}\left(x\right)=1.1x_{7}-x_{5}+1.9\leq 0\\ \; & \mathrm{With}\;\mathrm{bounds}:\\ \; & \;0.7\leq x_{2}\leq 0.8,\;17\leq x_{3}\leq 28,\;2.6\leq x_{1}\leq 3.6,\\ \; & \;5\leq x_{7}\leq 5.5,\;7.3\leq x_{5},x_{4}\leq 8.3,\;2.9\leq x_{6}\leq 3.9 \end{array}$$

Table 25 presents the optimization results for the speed reducer design problem, in which the MSAO achieves the best objective value of 2500.976, ranking first. It is followed by LSHADE (2592.060, second place) and the DOA (2995.152, third place), all demonstrating strong optimization capabilities. The WOA (2995.287, fourth), AOO (3000.988, fifth place), HHO (3025.564, sixth place), and TEAO (3047.838, seventh place) deliver similar performances, showcasing solid global search abilities but slightly lacking in fine-tuning under multiple geometric and stress constraints. The AO (3102.403, eighth place) and LOBLAO (3180.600, ninth place) perform comparatively worse.

## 5. Conclusions

This paper introduces a Multi-Strategy Aquila Optimizer (MSAO) that integrates a memory strategy and dream-sharing strategy, a random sub-dimension update mechanism, and enhanced position update rules to bolster performance on high-dimensional, complex optimization problems. First, the random sub-dimension update mechanism effectively decomposes the high-dimensional search space, significantly strengthening the algorithm’s ability to handle large-scale problems. Second, by combining the memory strategy and dream-sharing strategy from the DOA, individuals can both leverage past elite solutions and flexibly share information on each subpopulation, achieving an organic balance between global exploration and local exploitation. Finally, adaptive parameter control and dynamic opposition-based learning are employed to deeply refine the AO’s update rules, markedly accelerating convergence and improving solution precision. Compared with other AO variants, the MSAO achieves average rankings of 1.31 in the 10 dimension, 1.34 in the 30 dimension, 1.38 in the 50 dimension, and 1.45 in the 100 dimension. It is in the top ranking in each case, clearly demonstrating its superior performance over existing AO variants. Compared with other state-of-the-art algorithms, the MSAO delivered 16 (55%) and 20 (69%) best solutions out of 29 benchmark functions in the CEC2017 10D and 30D tests, respectively. When the dimensionality increases to 50D and 100D, the number of best solutions increases to 20 (69%) and 21 (72%). These results demonstrate that the MSAO’s three-strategy integration significantly enhances global search capability, effectively addressing the AO’s shortcomings in high-dimensional optimization. Ablation studies demonstrate the independent contributions of each module, thereby validating the effectiveness of each strategy. Across five real-world engineering design problems, the MSAO consistently achieved the top rank, demonstrating its superiority in engineering optimization and effectively addressing the shortcomings of existing AO variants in practical applications.

However, the performance in hybrid functions indicates that there is room for improvement. This limitation probably stems from the rigid division of the MSOA’s exploration and exploitation into two separate phases under a fixed threshold. In hybrid functions, subregions with diverse characteristics frequently overlap and cannot be addressed by inflexible phase transitions. As a result, if the algorithm becomes trapped in a local optimum in one region, it may shift phases too early or place undue emphasis on that area, thereby overlooking other promising regions of the search space. Future work will investigate a novel mechanism to more flexibly coordinate the exploration and exploitation phases, thereby overcoming this limitation, and focus on developing large-scale and multi-objective versions and applying the MSAO to path planning, image segmentation, data clustering, hyperparameter tuning, and wireless sensor network coverage to fully evaluate its generality and engineering value.

## Figures and Tables

**Figure 1 biomimetics-10-00620-f001:**
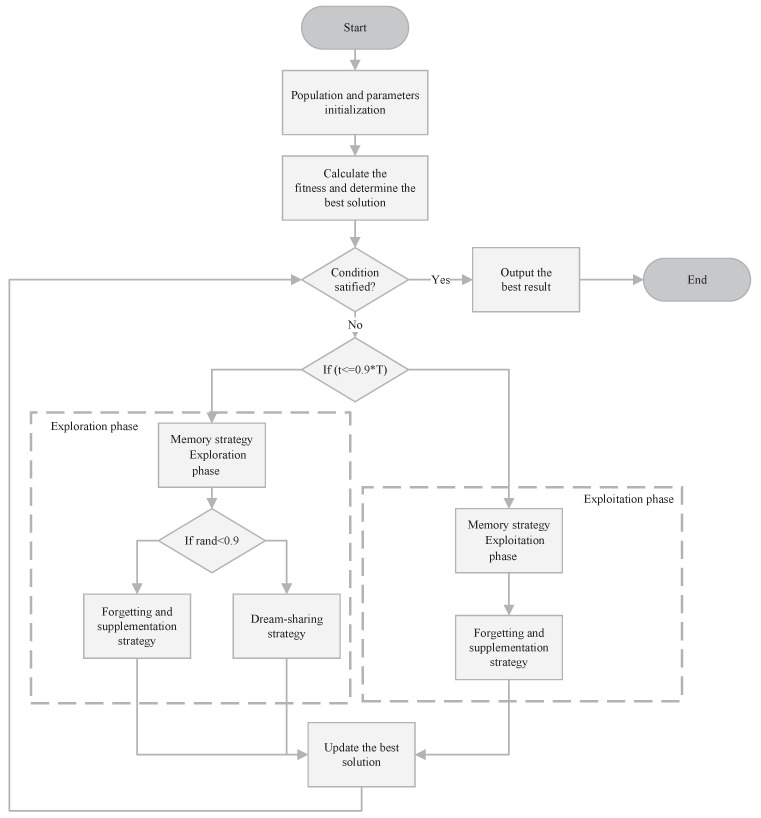
The flowchart of DOA.

**Figure 2 biomimetics-10-00620-f002:**
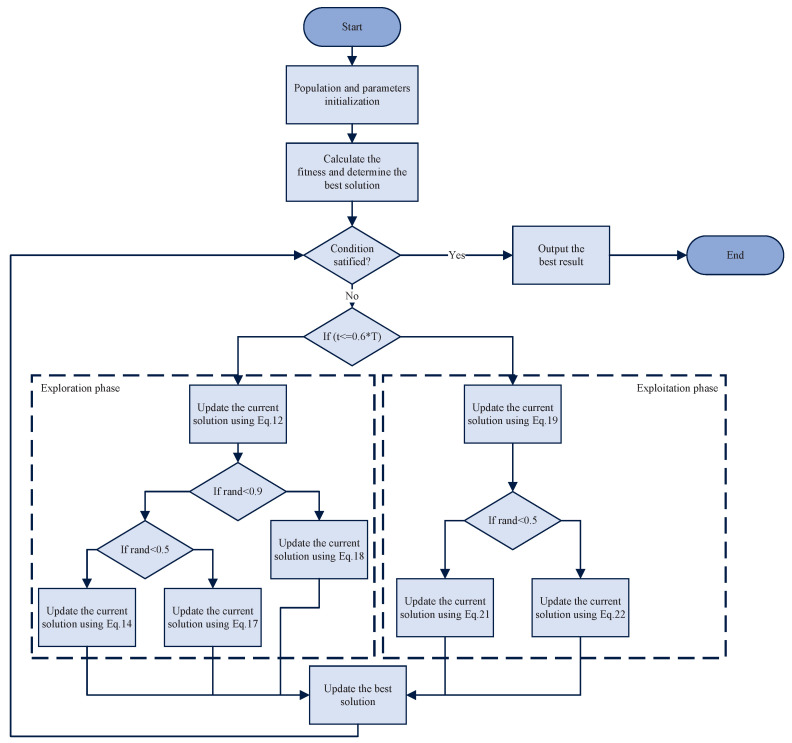
The flowchart of MSAO.

**Figure 3 biomimetics-10-00620-f003:**
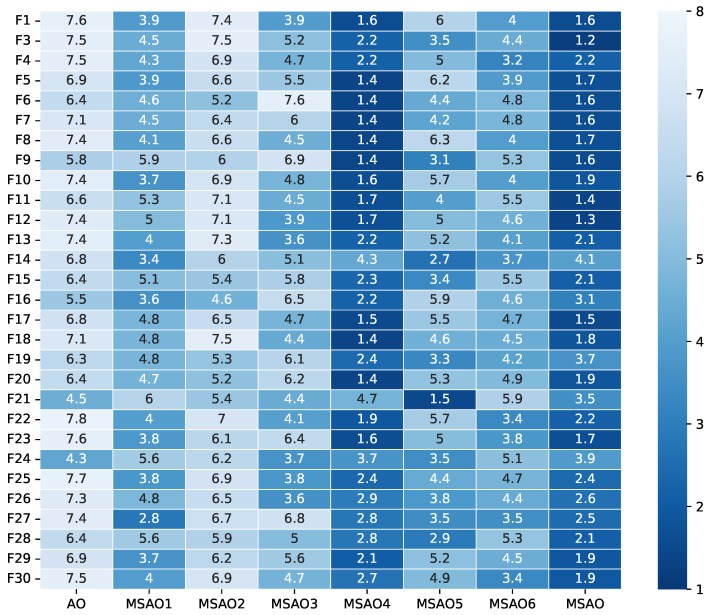
Friedman test rankings for the ablation experiments on CEC2017 (10D).

**Figure 4 biomimetics-10-00620-f004:**
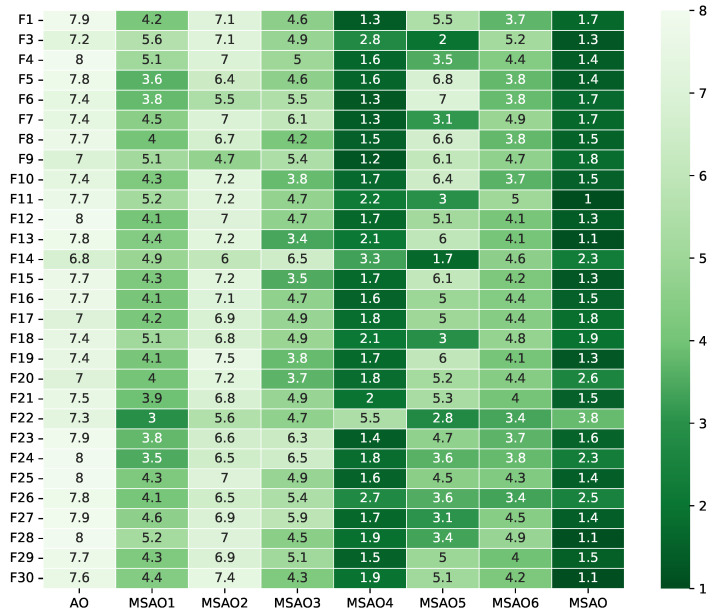
Friedman test rankings for the ablation experiments on CEC2017 (30D).

**Figure 5 biomimetics-10-00620-f005:**
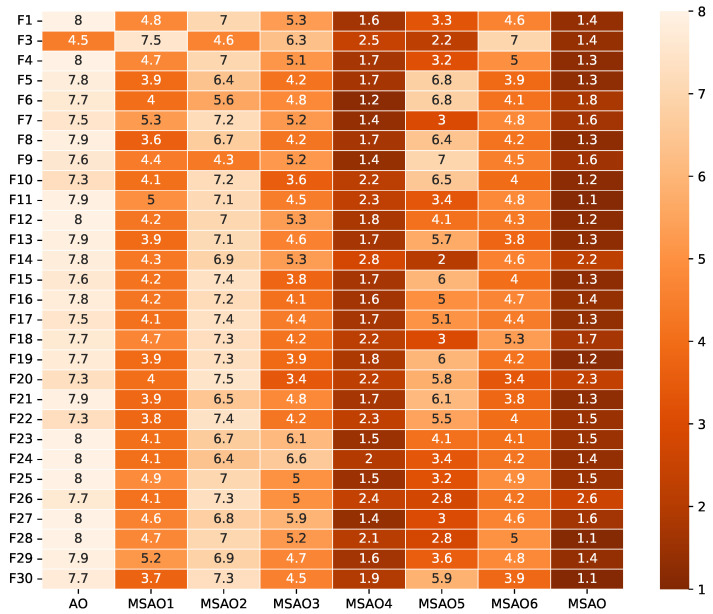
Friedman test rankings for the ablation experiments on CEC2017 (50D).

**Figure 6 biomimetics-10-00620-f006:**
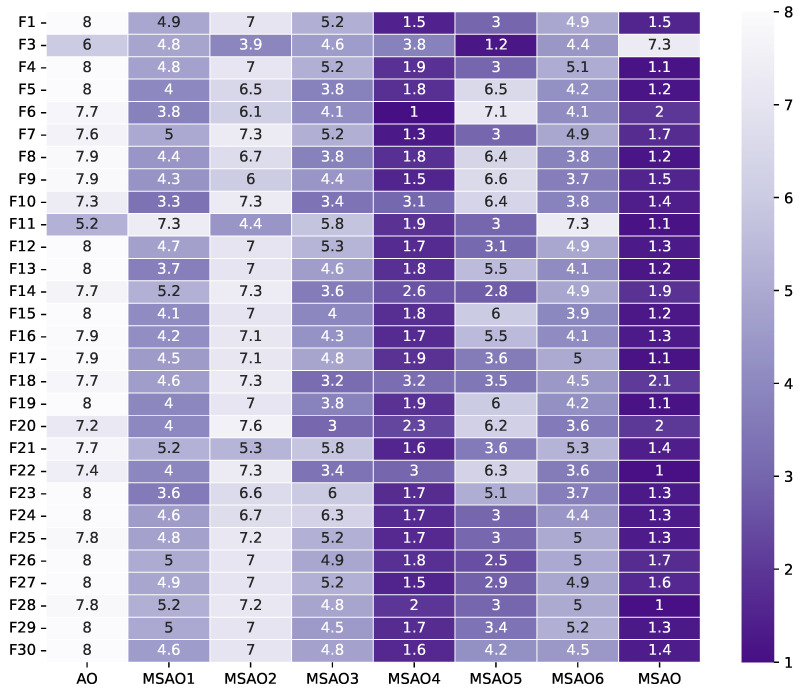
Friedman test rankings for the ablation experiments on CEC2017 (100D).

**Figure 7 biomimetics-10-00620-f007:**
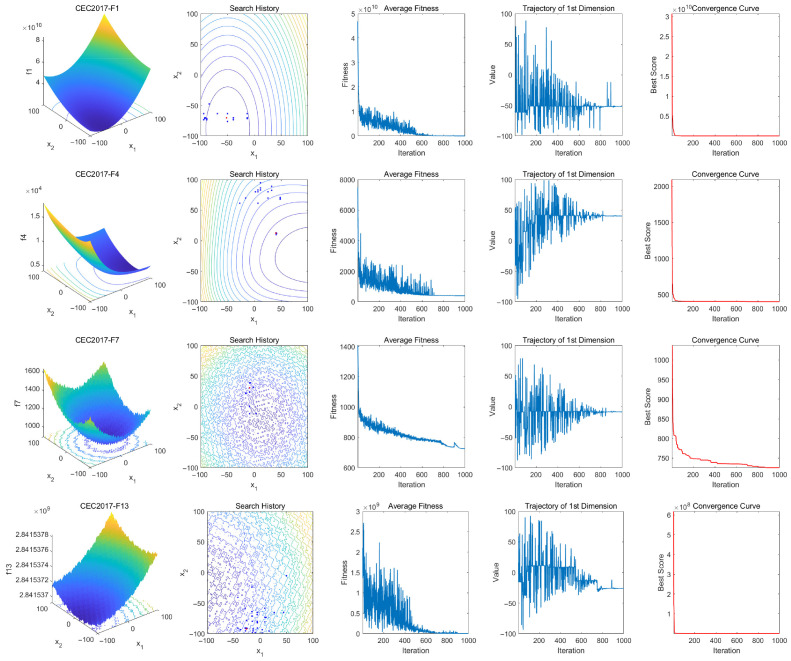
Qualitative analysis of MSOA on CEC2017.

**Figure 8 biomimetics-10-00620-f008:**
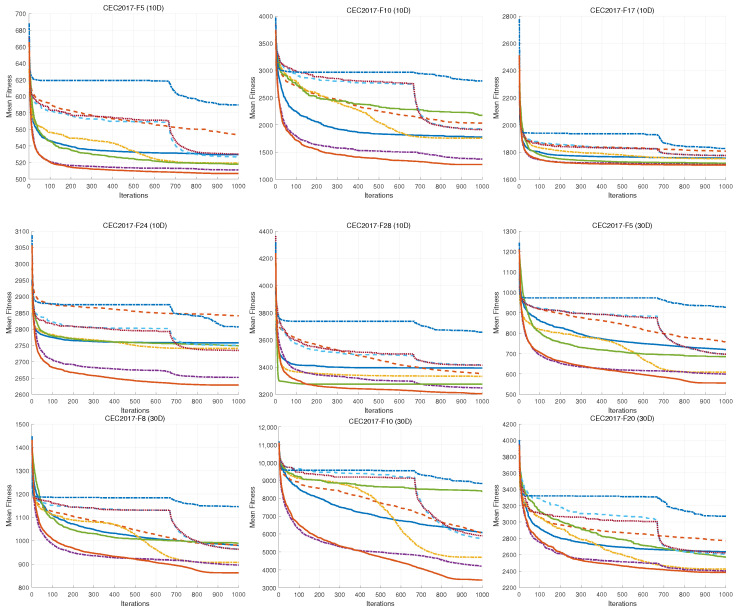
Convergence curves of different algorithms on CEC2017.

**Figure 9 biomimetics-10-00620-f009:**
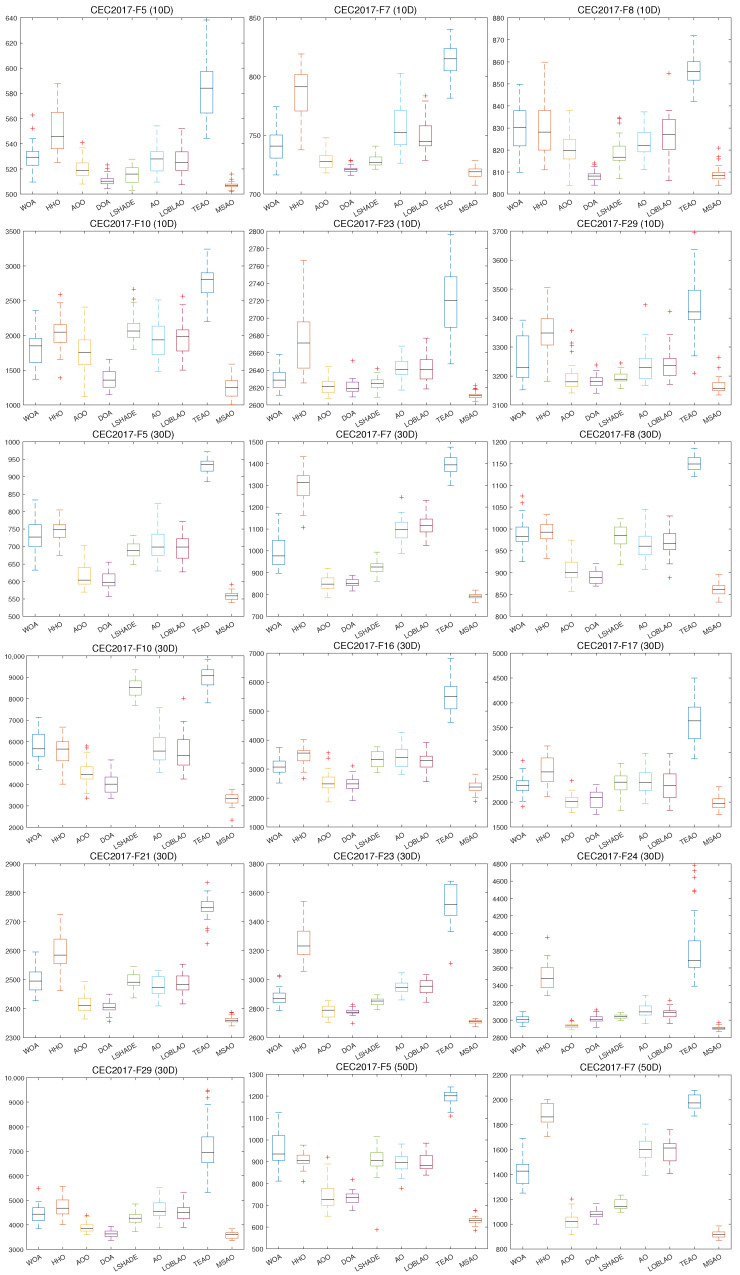
Box plot of different algorithms on CEC2017.

**Table 1 biomimetics-10-00620-t001:** Descriptions of CEC2017 benchmark test functions.

	Function	Range	Dim	$f_{\min}$
Unimodal functions				
F1	Shifted and Rotated Bent Cigar Function	[−100, 100]	10/30/50/100	100
F3	Shifted and Rotated Zakharov Function	[−100, 100]	10/30/50/100	200
Simple multimodal functions				
F4	Shifted and Rotated Rosenbrock’s Function	[−100, 100]	10/30/50/100	300
F5	Shifted and Rotated Rastrigin’s Function	[−100, 100]	10/30/50/100	400
F6	Shifted and Rotated Schaffer’s F7 Function	[−100, 100]	10/30/50/100	500
F7	Shifted and Rotated Lunacek Bi-Rastrigin’s Function	[−100, 100]	10/30/50/100	600
F8	Shifted and Rotated Non-Continuous Rastrigin’s Function	[−100, 100]	10/30/50/100	700
F9	Shifted and Rotated Levy Function	[−100, 100]	10/30/50/100	800
F10	Shifted and Rotated Schwefel’s Function	[−100, 100]	10/30/50/100	900
Hybrid functions				
F11	Hybrid Function 1 (N = 3)	[−100, 100]	10/30/50/100	1000
F12	Hybrid Function 2 (N = 3)	[−100, 100]	10/30/50/100	1100
F13	Hybrid Function 3 (N = 3)	[−100, 100]	10/30/50/100	1200
F14	Hybrid Function 4 (N = 4)	[−100, 100]	10/30/50/100	1300
F15	Hybrid Function 5 (N = 4)	[−100, 100]	10/30/50/100	1400
F16	Hybrid Function 6 (N = 4)	[−100, 100]	10/30/50/100	1500
F17	Hybrid Function 7 (N = 5)	[−100, 100]	10/30/50/100	1600
F18	Hybrid Function 8 (N = 5)	[−100, 100]	10/30/50/100	1700
F19	Hybrid Function 9 (N = 5)	[−100, 100]	10/30/50/100	1800
F20	Hybrid Function 10 (N = 6)	[−100, 100]	10/30/50/100	1900
Composition functions				
F21	Composition Function 1 (N = 3)	[−100, 100]	10/30/50/100	2000
F22	Composition Function 2 (N = 3)	[−100, 100]	10/30/50/100	2100
F23	Composition Function 3 (N = 4)	[−100, 100]	10/30/50/100	2200
F24	Composition Function 4 (N = 4)	[−100, 100]	10/30/50/100	2300
F25	Composition Function 5 (N = 5)	[−100, 100]	10/30/50/100	2400
F26	Composition Function 6 (N = 5)	[−100, 100]	10/30/50/100	2500
F27	Composition Function 7 (N = 6)	[−100, 100]	10/30/50/100	2600
F28	Composition Function 8 (N = 6)	[−100, 100]	10/30/50/100	2700
F29	Composition Function 9 (N = 3)	[−100, 100]	10/30/50/100	2800
F30	Composition Function 10 (N = 3)	[−100, 100]	10/30/50/100	2900

**Table 2 biomimetics-10-00620-t002:** Setting parameters for contrast algorithms.

Algorithms	Name of the Parameter	Value of the Parameter
WOA	a, a2, b	[0, 2], [−1, −2], 1
HHO	E0, E1, q, r	[−1, 1], [0, 2], [0, 1], [0, 1]
AOO	switch threshold	0.5
DOA	u	0.9
LSHADE	Memorysize, p, H	5, 5, 6
AO	alpha, delta	0.1, 0.1
SGAO	b, c	1, 15
MMSIAO	S	0.5
IAO	Kv	2
LOBLAO	µr	0.5
TEAO	switch threshold	0.5

**Table 3 biomimetics-10-00620-t003:** Sensitivity analysis on CEC2017.

Dim (w/t/l)	0.5	0.6	0.7	0.8	0.9
10	2/6/21	**10/9/10**	4/12/15	1/10/18	1/8/20
30	3/4/22	7/5/17	**10/4/15**	3/3/23	1/3/25
50	3/3/23	**9/6/14**	5/10/14	3/5/21	2/3/24
100	3/5/21	**9/7/13**	4/8/17	2/4/23	1/5/13
Total	11/18/87	**35/27/54**	23/34/61	9/22/85	5/19/82

Bold values indicate the best performance under each dimension.

**Table 4 biomimetics-10-00620-t004:** Versions of various MSOAs.

Algorithm	Key Component I	Key Component II	Key Component III
AO	×	×	×
MSAO1	∘	×	×
MSAO2	×	∘	×
MSAO3	×	×	∘
MSAO4	∘	∘	×
MSAO5	∘	×	∘
MSAO6	×	∘	∘
MSAO	∘	∘	∘

**Table 5 biomimetics-10-00620-t005:** Experimental results of MSAO and AO variants on CEC2017 (10D).

		SGAO	MMSIAO	IAO	LOBLAO	TEAO	MSAO
F1	Ave	$2.88\times 10^{6}$	$4.46\times 10^{6}$	$5.29\times 10^{10}$	$8.25\times 10^{5}$	$7.50\times 10^{9}$	$1.09\times 10^{5}$
	Std	$2.94\times 10^{3}$	$3.85\times 10^{3}$	$4.86\times 10^{6}$	$7.27\times 10^{5}$	$2.50\times 10^{9}$	$9.80\times 10^{4}$
	Rank	3	4	6	2	5	1
F3	Ave	$3.00\times 10^{2}$	$3.00\times 10^{2}$	$1.17\times 10^{4}$	$6.04\times 10^{2}$	$9.19\times 10^{3}$	$3.36\times 10^{2}$
	Std	$1.18\times 10^{-10}$	$2.33\times 10^{0}$	$5.70\times 10^{2}$	$1.93\times 10^{2}$	$2.64\times 10^{3}$	$2.76\times 10^{1}$
	Rank	1	1	6	4	5	3
F4	Ave	$9.03\times 10^{2}$	$9.04\times 10^{2}$	$4.45\times 10^{2}$	$4.21\times 10^{2}$	$8.19\times 10^{2}$	$4.03\times 10^{2}$
	Std	$2.02\times 10^{0}$	$2.23\times 10^{0}$	$4.50\times 10^{1}$	$2.29\times 10^{1}$	$3.33\times 10^{2}$	$2.79\times 10^{0}$
	Rank	5	6	3	2	4	1
F5	Ave	$6.44\times 10^{2}$	$6.29\times 10^{2}$	$5.94\times 10^{2}$	$5.28\times 10^{2}$	$5.91\times 10^{2}$	$5.12\times 10^{2}$
	Std	$1.81\times 10^{1}$	$8.46\times 10^{0}$	$1.72\times 10^{1}$	$1.07\times 10^{1}$	$1.57\times 10^{1}$	$5.43\times 10^{0}$
	Rank	6	5	4	2	3	1
F6	Ave	$6.53\times 10^{2}$	$6.49\times 10^{2}$	$6.71\times 10^{2}$	$6.18\times 10^{2}$	$6.48\times 10^{2}$	$6.00\times 10^{2}$
	Std	$9.70\times 10^{0}$	$6.27\times 10^{0}$	$1.03\times 10^{1}$	$6.17\times 10^{0}$	$6.88\times 10^{0}$	$8.87\times 10^{-2}$
	Rank	5	4	6	2	3	1
F7	Ave	$8.17\times 10^{2}$	$8.19\times 10^{2}$	$8.38\times 10^{2}$	$7.52\times 10^{2}$	$8.15\times 10^{2}$	$7.21\times 10^{2}$
	Std	$2.19\times 10^{1}$	$1.51\times 10^{1}$	$2.13\times 10^{1}$	$1.50\times 10^{1}$	$1.23\times 10^{1}$	$3.66\times 10^{0}$
	Rank	4	5	6	2	3	1
F8	Ave	$8.62\times 10^{2}$	$8.26\times 10^{2}$	$8.80\times 10^{2}$	$8.24\times 10^{2}$	$8.59\times 10^{2}$	$8.09\times 10^{2}$
	Std	$7.24\times 10^{0}$	$5.50\times 10^{0}$	$6.70\times 10^{0}$	$9.34\times 10^{0}$	$8.29\times 10^{0}$	$2.68\times 10^{0}$
	Rank	5	3	6	2	4	1
F9	Ave	$1.47\times 10^{3}$	$9.58\times 10^{2}$	$1.48\times 10^{3}$	$1.02\times 10^{3}$	$1.42\times 10^{3}$	$9.00\times 10^{2}$
	Std	$1.74\times 10^{2}$	$5.61\times 10^{1}$	$2.24\times 10^{2}$	$8.86\times 10^{1}$	$1.78\times 10^{2}$	$2.67\times 10^{-1}$
	Rank	5	2	6	3	4	1
F10	Ave	$2.90\times 10^{3}$	$2.85\times 10^{3}$	$3.11\times 10^{3}$	$1.89\times 10^{3}$	$2.82\times 10^{3}$	$1.33\times 10^{3}$
	Std	$3.60\times 10^{2}$	$2.74\times 10^{2}$	$2.63\times 10^{2}$	$2.79\times 10^{2}$	$2.43\times 10^{2}$	$1.83\times 10^{2}$
	Rank	5	4	6	2	3	1
F11	Ave	$2.15\times 10^{3}$	$2.13\times 10^{3}$	$2.20\times 10^{3}$	$1.19\times 10^{3}$	$1.70\times 10^{3}$	$1.11\times 10^{3}$
	Std	$4.56\times 10^{1}$	$2.08\times 10^{1}$	$7.20\times 10^{1}$	$5.25\times 10^{1}$	$1.08\times 10^{3}$	$4.47\times 10^{0}$
	Rank	5	4	6	2	3	1
F12	Ave	$1.70\times 10^{6}$	$8.32\times 10^{6}$	$5.59\times 10^{8}$	$3.68\times 10^{6}$	$1.28\times 10^{8}$	$2.16\times 10^{4}$
	Std	$1.41\times 10^{4}$	$5.06\times 10^{3}$	$4.36\times 10^{6}$	$3.40\times 10^{6}$	$1.31\times 10^{8}$	$2.07\times 10^{4}$
	Rank	2	4	6	3	5	1
F13	Ave	$1.69\times 10^{4}$	$1.98\times 10^{4}$	$1.83\times 10^{4}$	$1.63\times 10^{4}$	$1.23\times 10^{4}$	$7.72\times 10^{3}$
	Std	$1.07\times 10^{4}$	$7.17\times 10^{3}$	$1.15\times 10^{4}$	$1.15\times 10^{4}$	$1.02\times 10^{4}$	$5.25\times 10^{3}$
	Rank	4	6	5	3	2	1
F14	Ave	$2.52\times 10^{3}$	$2.49\times 10^{3}$	$2.64\times 10^{3}$	$2.27\times 10^{3}$	$1.79\times 10^{3}$	$3.90\times 10^{3}$
	Std	$5.28\times 10^{1}$	$3.27\times 10^{1}$	$1.43\times 10^{2}$	$1.16\times 10^{3}$	$1.04\times 10^{3}$	$3.04\times 10^{3}$
	Rank	4	3	5	2	1	6
F15	Ave	$1.64\times 10^{3}$	$1.63\times 10^{3}$	$6.60\times 10^{3}$	$5.23\times 10^{3}$	$5.63\times 10^{3}$	$2.14\times 10^{3}$
	Std	$1.22\times 10^{2}$	$1.21\times 10^{2}$	$2.77\times 10^{3}$	$2.79\times 10^{3}$	$2.88\times 10^{3}$	$6.68\times 10^{2}$
	Rank	2	1	6	4	5	3
F16	Ave	$2.05\times 10^{3}$	$2.06\times 10^{3}$	$2.09\times 10^{3}$	$1.81\times 10^{3}$	$2.05\times 10^{3}$	$1.70\times 10^{3}$
	Std	$1.59\times 10^{2}$	$1.30\times 10^{2}$	$1.41\times 10^{2}$	$1.30\times 10^{2}$	$1.25\times 10^{2}$	$1.12\times 10^{2}$
	Rank	3	5	6	2	3	1
F17	Ave	$1.88\times 10^{3}$	$1.86\times 10^{3}$	$1.88\times 10^{3}$	$1.77\times 10^{3}$	$1.82\times 10^{3}$	$1.71\times 10^{3}$
	Std	$5.28\times 10^{1}$	$2.55\times 10^{1}$	$2.97\times 10^{1}$	$2.17\times 10^{1}$	$3.69\times 10^{1}$	$8.97\times 10^{0}$
	Rank	5	4	5	2	3	1
F18	Ave	$1.86\times 10^{4}$	$1.88\times 10^{4}$	$2.60\times 10^{4}$	$2.84\times 10^{4}$	$1.86\times 10^{4}$	$3.92\times 10^{3}$
	Std	$1.30\times 10^{4}$	$1.05\times 10^{4}$	$9.99\times 10^{3}$	$1.46\times 10^{4}$	$1.45\times 10^{4}$	$1.70\times 10^{3}$
	Rank	2	4	5	6	2	1
F19	Ave	$3.04\times 10^{4}$	$3.15\times 10^{4}$	$9.93\times 10^{4}$	$1.48\times 10^{4}$	$2.56\times 10^{4}$	$4.26\times 10^{3}$
	Std	$5.93\times 10^{3}$	$4.91\times 10^{3}$	$1.14\times 10^{4}$	$3.28\times 10^{4}$	$4.06\times 10^{4}$	$2.84\times 10^{3}$
	Rank	5	4	6	2	3	1
F20	Ave	$2.10\times 10^{3}$	$2.12\times 10^{3}$	$2.28\times 10^{3}$	$2.12\times 10^{3}$	$2.23\times 10^{3}$	$2.00\times 10^{3}$
	Std	$8.19\times 10^{1}$	$5.90\times 10^{1}$	$5.11\times 10^{1}$	$5.71\times 10^{1}$	$5.72\times 10^{1}$	$1.63\times 10^{0}$
	Rank	2	3	6	3	5	1
F21	Ave	$2.43\times 10^{3}$	$2.35\times 10^{3}$	$2.35\times 10^{3}$	$2.28\times 10^{3}$	$2.33\times 10^{3}$	$2.29\times 10^{3}$
	Std	$5.41\times 10^{1}$	$3.32\times 10^{1}$	$5.91\times 10^{1}$	$5.67\times 10^{1}$	$5.05\times 10^{1}$	$5.24\times 10^{1}$
	Rank	6	4	4	1	3	2
F22	Ave	$2.85\times 10^{3}$	$2.31\times 10^{3}$	$2.31\times 10^{3}$	$2.31\times 10^{3}$	$2.81\times 10^{3}$	$2.29\times 10^{3}$
	Std	$2.28\times 10^{2}$	$4.29\times 10^{0}$	$9.53\times 10^{0}$	$7.41\times 10^{0}$	$2.02\times 10^{2}$	$3.47\times 10^{1}$
	Rank	6	2	2	2	5	1
F23	Ave	$2.74\times 10^{3}$	$2.75\times 10^{3}$	$2.77\times 10^{3}$	$2.64\times 10^{3}$	$2.73\times 10^{3}$	$2.62\times 10^{3}$
	Std	$1.92\times 10^{1}$	$1.10\times 10^{1}$	$4.53\times 10^{1}$	$1.40\times 10^{1}$	$3.43\times 10^{1}$	$7.19\times 10^{0}$
	Rank	4	5	6	2	3	1
F24	Ave	$2.85\times 10^{3}$	$2.85\times 10^{3}$	$2.96\times 10^{3}$	$2.74\times 10^{3}$	$2.82\times 10^{3}$	$2.69\times 10^{3}$
	Std	$8.72\times 10^{1}$	$6.75\times 10^{1}$	$1.24\times 10^{2}$	$8.25\times 10^{1}$	$7.39\times 10^{1}$	$9.94\times 10^{1}$
	Rank	4	4	6	2	3	1
F25	Ave	$2.92\times 10^{3}$	$3.33\times 10^{3}$	$3.34\times 10^{3}$	$2.92\times 10^{3}$	$3.21\times 10^{3}$	$2.90\times 10^{3}$
	Std	$6.42\times 10^{1}$	$2.40\times 10^{1}$	$2.37\times 10^{1}$	$5.98\times 10^{1}$	$1.28\times 10^{2}$	$7.85\times 10^{1}$
	Rank	2	5	6	2	4	1
F26	Ave	$4.31\times 10^{3}$	$4.02\times 10^{3}$	$4.65\times 10^{3}$	$3.03\times 10^{3}$	$3.99\times 10^{3}$	$2.81\times 10^{3}$
	Std	$4.45\times 10^{2}$	$1.18\times 10^{2}$	$6.49\times 10^{2}$	$1.73\times 10^{2}$	$3.08\times 10^{2}$	$1.40\times 10^{2}$
	Rank	5	4	6	2	3	1
F27	Ave	$3.22\times 10^{3}$	$3.20\times 10^{3}$	$3.25\times 10^{3}$	$3.10\times 10^{3}$	$3.18\times 10^{3}$	$3.10\times 10^{3}$
	Std	$3.15\times 10^{1}$	$5.00\times 10^{0}$	$4.32\times 10^{1}$	$5.74\times 10^{0}$	$2.75\times 10^{1}$	$8.36\times 10^{0}$
	Rank	5	4	6	1	3	1
F28	Ave	$4.31\times 10^{3}$	$4.26\times 10^{3}$	$4.38\times 10^{3}$	$3.38\times 10^{3}$	$3.70\times 10^{3}$	$3.24\times 10^{3}$
	Std	$1.37\times 10^{2}$	$1.42\times 10^{2}$	$1.64\times 10^{2}$	$1.02\times 10^{2}$	$1.51\times 10^{2}$	$1.35\times 10^{2}$
	Rank	5	4	6	2	3	1
F29	Ave	$3.51\times 10^{3}$	$3.52\times 10^{3}$	$3.46\times 10^{3}$	$3.23\times 10^{3}$	$3.43\times 10^{3}$	$3.19\times 10^{3}$
	Std	$1.03\times 10^{2}$	$4.19\times 10^{1}$	$8.01\times 10^{1}$	$4.64\times 10^{1}$	$1.14\times 10^{2}$	$2.97\times 10^{1}$
	Rank	5	6	4	2	3	1
F30	Ave	$6.71\times 10^{6}$	$6.69\times 10^{6}$	$7.26\times 10^{6}$	$5.69\times 10^{5}$	$6.03\times 10^{6}$	$6.92\times 10^{4}$
	Std	$4.77\times 10^{5}$	$1.48\times 10^{5}$	$5.87\times 10^{6}$	$8.01\times 10^{5}$	$5.67\times 10^{6}$	$6.09\times 10^{4}$
	Rank	5	4	6	2	3	1

**Table 6 biomimetics-10-00620-t006:** Experimental results of MSAO and AO variants on CEC2017 (30D).

		SGAO	MMSIAO	IAO	LOBLAO	TEAO	MSAO
F1	Ave	$6.95\times 10^{8}$	$2.64\times 10^{9}$	$6.77\times 10^{10}$	$5.02\times 10^{8}$	$4.59\times 10^{10}$	$9.03\times 10^{6}$
	Std	$9.54\times 10^{3}$	$5.31\times 10^{5}$	$2.99\times 10^{8}$	$2.24\times 10^{8}$	$4.90\times 10^{9}$	$3.29\times 10^{6}$
	Rank	3	4	6	2	5	1
F3	Ave	$1.35\times 10^{5}$	$1.21\times 10^{5}$	$1.75\times 10^{5}$	$5.46\times 10^{4}$	$8.23\times 10^{4}$	$1.70\times 10^{4}$
	Std	$8.67\times 10^{3}$	$9.38\times 10^{3}$	$5.85\times 10^{3}$	$5.97\times 10^{3}$	$5.29\times 10^{3}$	$7.24\times 10^{3}$
	Rank	5	4	6	2	3	1
F4	Ave	$4.99\times 10^{2}$	$7.16\times 10^{2}$	$7.40\times 10^{2}$	$6.48\times 10^{2}$	$1.22\times 10^{4}$	$5.09\times 10^{2}$
	Std	$2.37\times 10^{1}$	$2.33\times 10^{1}$	$1.07\times 10^{2}$	$5.82\times 10^{1}$	$1.56\times 10^{3}$	$2.13\times 10^{1}$
	Rank	1	4	5	3	6	2
F5	Ave	$1.44\times 10^{3}$	$1.89\times 10^{3}$	$1.58\times 10^{3}$	$7.10\times 10^{2}$	$9.32\times 10^{2}$	$5.99\times 10^{2}$
	Std	$4.21\times 10^{1}$	$3.51\times 10^{1}$	$3.48\times 10^{1}$	$4.07\times 10^{1}$	$2.55\times 10^{1}$	$2.39\times 10^{1}$
	Rank	4	6	5	2	3	1
F6	Ave	$7.06\times 10^{2}$	$7.16\times 10^{2}$	$7.65\times 10^{2}$	$6.52\times 10^{2}$	$6.94\times 10^{2}$	$6.02\times 10^{2}$
	Std	$8.24\times 10^{0}$	$7.83\times 10^{0}$	$5.14\times 10^{0}$	$6.79\times 10^{0}$	$6.21\times 10^{0}$	$4.52\times 10^{-1}$
	Rank	4	5	6	2	3	1
F7	Ave	$1.48\times 10^{3}$	$1.42\times 10^{3}$	$1.41\times 10^{3}$	$1.09\times 10^{3}$	$1.38\times 10^{3}$	$8.51\times 10^{2}$
	Std	$9.61\times 10^{1}$	$5.15\times 10^{1}$	$7.86\times 10^{1}$	$6.09\times 10^{1}$	$3.30\times 10^{1}$	$1.33\times 10^{1}$
	Rank	6	5	4	2	3	1
F8	Ave	$1.72\times 10^{3}$	$1.31\times 10^{3}$	$1.78\times 10^{3}$	$9.64\times 10^{2}$	$1.14\times 10^{3}$	$8.84\times 10^{2}$
	Std	$2.20\times 10^{1}$	$2.39\times 10^{1}$	$1.88\times 10^{1}$	$2.77\times 10^{1}$	$1.87\times 10^{1}$	$1.80\times 10^{1}$
	Rank	5	4	6	2	3	1
F9	Ave	$1.17\times 10^{4}$	$1.68\times 10^{4}$	$1.31\times 10^{4}$	$6.78\times 10^{3}$	$1.16\times 10^{4}$	$1.64\times 10^{3}$
	Std	$5.40\times 10^{2}$	$7.54\times 10^{2}$	$9.58\times 10^{2}$	$1.14\times 10^{3}$	$1.22\times 10^{3}$	$5.25\times 10^{2}$
	Rank	4	6	5	2	3	1
F10	Ave	$1.15\times 10^{4}$	$1.08\times 10^{4}$	$1.39\times 10^{4}$	$5.55\times 10^{3}$	$9.06\times 10^{3}$	$4.12\times 10^{3}$
	Std	$4.67\times 10^{2}$	$8.29\times 10^{2}$	$5.20\times 10^{2}$	$7.51\times 10^{2}$	$5.05\times 10^{2}$	$5.62\times 10^{2}$
	Rank	5	4	6	2	3	1
F11	Ave	$7.24\times 10^{3}$	$7.28\times 10^{3}$	$9.60\times 10^{3}$	$2.30\times 10^{3}$	$7.12\times 10^{3}$	$1.21\times 10^{3}$
	Std	$5.63\times 10^{1}$	$7.02\times 10^{1}$	$6.00\times 10^{2}$	$7.30\times 10^{2}$	$1.50\times 10^{3}$	$3.56\times 10^{1}$
	Rank	4	5	6	2	3	1
F12	Ave	$9.71\times 10^{5}$	$1.82\times 10^{6}$	$1.42\times 10^{10}$	$1.16\times 10^{8}$	$1.37\times 10^{10}$	$2.10\times 10^{6}$
	Std	$1.44\times 10^{6}$	$1.04\times 10^{6}$	$1.07\times 10^{8}$	$1.04\times 10^{8}$	$2.72\times 10^{9}$	$9.40\times 10^{5}$
	Rank	1	2	6	4	5	3
F13	Ave	$1.84\times 10^{4}$	$1.74\times 10^{10}$	$1.38\times 10^{6}$	$3.06\times 10^{6}$	$9.00\times 10^{9}$	$7.92\times 10^{4}$
	Std	$1.50\times 10^{4}$	$1.66\times 10^{4}$	$1.93\times 10^{6}$	$8.55\times 10^{6}$	$4.22\times 10^{9}$	$3.48\times 10^{4}$
	Rank	1	6	3	4	5	2
F14	Ave	$1.66\times 10^{6}$	$1.80\times 10^{6}$	$1.71\times 10^{6}$	$8.60\times 10^{5}$	$1.64\times 10^{6}$	$2.29\times 10^{5}$
	Std	$4.71\times 10^{4}$	$6.80\times 10^{4}$	$1.38\times 10^{6}$	$9.72\times 10^{5}$	$1.35\times 10^{6}$	$2.09\times 10^{5}$
	Rank	4	6	5	2	3	1
F15	Ave	$8.67\times 10^{3}$	$7.64\times 10^{3}$	$7.52\times 10^{4}$	$1.10\times 10^{5}$	$2.35\times 10^{8}$	$8.09\times 10^{3}$
	Std	$1.07\times 10^{4}$	$5.48\times 10^{3}$	$3.14\times 10^{4}$	$6.29\times 10^{4}$	$1.88\times 10^{8}$	$4.61\times 10^{3}$
	Rank	3	1	4	5	6	2
F16	Ave	$5.99\times 10^{3}$	$5.84\times 10^{3}$	$5.94\times 10^{3}$	$3.32\times 10^{3}$	$5.35\times 10^{3}$	$2.40\times 10^{3}$
	Std	$3.81\times 10^{2}$	$3.37\times 10^{2}$	$4.21\times 10^{2}$	$4.23\times 10^{2}$	$6.96\times 10^{2}$	$2.49\times 10^{2}$
	Rank	6	4	5	2	3	1
F17	Ave	$3.76\times 10^{3}$	$4.26\times 10^{3}$	$3.65\times 10^{3}$	$2.46\times 10^{3}$	$3.56\times 10^{3}$	$2.08\times 10^{3}$
	Std	$3.05\times 10^{2}$	$2.18\times 10^{2}$	$3.12\times 10^{2}$	$2.52\times 10^{2}$	$3.52\times 10^{2}$	$1.71\times 10^{2}$
	Rank	4	6	5	2	3	1
F18	Ave	$6.00\times 10^{6}$	$5.24\times 10^{6}$	$7.13\times 10^{6}$	$4.34\times 10^{6}$	$2.88\times 10^{7}$	$5.51\times 10^{5}$
	Std	$2.48\times 10^{5}$	$5.51\times 10^{5}$	$7.33\times 10^{6}$	$3.72\times 10^{6}$	$2.70\times 10^{7}$	$5.74\times 10^{5}$
	Rank	4	3	5	2	6	1
F19	Ave	$6.32\times 10^{3}$	$6.41\times 10^{8}$	$1.74\times 10^{9}$	$1.99\times 10^{6}$	$3.74\times 10^{8}$	$1.21\times 10^{4}$
	Std	$4.31\times 10^{3}$	$5.25\times 10^{3}$	$1.71\times 10^{6}$	$2.17\times 10^{6}$	$2.66\times 10^{8}$	$7.95\times 10^{3}$
	Rank	1	5	6	3	4	2
F20	Ave	$3.79\times 10^{3}$	$3.53\times 10^{3}$	$3.76\times 10^{3}$	$2.62\times 10^{3}$	$3.07\times 10^{3}$	$2.41\times 10^{3}$
	Std	$3.09\times 10^{2}$	$1.40\times 10^{2}$	$2.33\times 10^{2}$	$1.73\times 10^{2}$	$2.16\times 10^{2}$	$1.31\times 10^{2}$
	Rank	6	4	5	2	3	1
F21	Ave	$2.74\times 10^{3}$	$2.73\times 10^{3}$	$2.89\times 10^{3}$	$2.50\times 10^{3}$	$2.73\times 10^{3}$	$2.40\times 10^{3}$
	Std	$7.03\times 10^{1}$	$2.87\times 10^{1}$	$4.21\times 10^{1}$	$4.96\times 10^{1}$	$3.05\times 10^{1}$	$2.17\times 10^{1}$
	Rank	5	3	6	2	3	1
F22	Ave	$1.15\times 10^{4}$	$1.32\times 10^{4}$	$1.63\times 10^{4}$	$2.57\times 10^{3}$	$9.11\times 10^{3}$	$3.43\times 10^{3}$
	Std	$2.27\times 10^{3}$	$4.32\times 10^{1}$	$8.36\times 10^{2}$	$1.68\times 10^{2}$	$9.22\times 10^{2}$	$1.60\times 10^{3}$
	Rank	4	5	6	1	3	2
F23	Ave	$3.55\times 10^{3}$	$3.55\times 10^{3}$	$3.65\times 10^{3}$	$2.96\times 10^{3}$	$3.52\times 10^{3}$	$2.78\times 10^{3}$
	Std	$8.41\times 10^{1}$	$4.51\times 10^{1}$	$1.16\times 10^{2}$	$6.45\times 10^{1}$	$1.50\times 10^{2}$	$2.55\times 10^{1}$
	Rank	4	4	6	2	3	1
F24	Ave	$3.88\times 10^{3}$	$3.85\times 10^{3}$	$3.91\times 10^{3}$	$3.09\times 10^{3}$	$3.80\times 10^{3}$	$3.00\times 10^{3}$
	Std	$7.59\times 10^{1}$	$5.85\times 10^{1}$	$1.64\times 10^{2}$	$6.71\times 10^{1}$	$3.39\times 10^{2}$	$3.68\times 10^{1}$
	Rank	5	4	6	2	3	1
F25	Ave	$2.90\times 10^{3}$	$2.90\times 10^{3}$	$5.01\times 10^{3}$	$2.99\times 10^{3}$	$4.65\times 10^{3}$	$2.90\times 10^{3}$
	Std	$1.80\times 10^{1}$	$2.10\times 10^{1}$	$4.43\times 10^{1}$	$3.36\times 10^{1}$	$2.95\times 10^{2}$	$1.74\times 10^{1}$
	Rank	1	1	6	4	5	1
F26	Ave	$1.02\times 10^{4}$	$1.13\times 10^{4}$	$1.77\times 10^{4}$	$6.09\times 10^{3}$	$1.06\times 10^{4}$	$4.34\times 10^{3}$
	Std	$1.80\times 10^{3}$	$1.61\times 10^{3}$	$1.56\times 10^{3}$	$1.54\times 10^{3}$	$5.59\times 10^{2}$	$1.02\times 10^{3}$
	Rank	3	5	6	2	4	1
F27	Ave	$4.49\times 10^{3}$	$4.47\times 10^{3}$	$4.59\times 10^{3}$	$3.36\times 10^{3}$	$4.16\times 10^{3}$	$3.24\times 10^{3}$
	Std	$4.15\times 10^{1}$	$3.99\times 10^{1}$	$1.17\times 10^{2}$	$7.06\times 10^{1}$	$4.02\times 10^{2}$	$1.25\times 10^{1}$
	Rank	5	4	6	2	3	1
F28	Ave	$3.23\times 10^{3}$	$3.27\times 10^{3}$	$3.65\times 10^{3}$	$3.42\times 10^{3}$	$6.53\times 10^{3}$	$3.25\times 10^{3}$
	Std	$3.12\times 10^{1}$	$3.15\times 10^{1}$	$8.95\times 10^{1}$	$5.48\times 10^{1}$	$3.67\times 10^{2}$	$2.56\times 10^{1}$
	Rank	1	3	6	4	5	2
F29	Ave	$7.41\times 10^{3}$	$7.19\times 10^{3}$	$7.29\times 10^{3}$	$4.68\times 10^{3}$	$6.77\times 10^{3}$	$3.70\times 10^{3}$
	Std	$3.49\times 10^{2}$	$3.35\times 10^{2}$	$6.72\times 10^{2}$	$3.45\times 10^{2}$	$6.92\times 10^{2}$	$1.61\times 10^{2}$
	Rank	6	4	5	2	3	1
F30	Ave	$4.38\times 10^{9}$	$2.40\times 10^{9}$	$2.48\times 10^{10}$	$1.51\times 10^{7}$	$1.43\times 10^{9}$	$3.05\times 10^{4}$
	Std	$1.20\times 10^{5}$	$1.52\times 10^{4}$	$2.49\times 10^{7}$	$1.12\times 10^{7}$	$1.40\times 10^{9}$	$1.79\times 10^{4}$
	Rank	5	4	6	2	3	1

**Table 7 biomimetics-10-00620-t007:** Experimental results of MSAO and AO variants on CEC2017 (50D).

		SGAO	MMSIAO	IAO	LOBLAO	TEAO	MSAO
F1	Ave	$1.35\times 10^{6}$	$1.27\times 10^{11}$	$6.16\times 10^{11}$	$5.35\times 10^{9}$	$1.05\times 10^{11}$	$2.01\times 10^{8}$
	Std	$1.66\times 10^{6}$	$1.86\times 10^{9}$	$1.55\times 10^{9}$	$1.59\times 10^{9}$	$5.70\times 10^{9}$	$4.01\times 10^{7}$
	Rank	1	5	6	3	4	2
F3	Ave	$2.59\times 10^{5}$	$2.57\times 10^{5}$	$2.89\times 10^{5}$	$2.56\times 10^{5}$	$2.42\times 10^{5}$	$9.23\times 10^{4}$
	Std	$3.47\times 10^{4}$	$4.44\times 10^{4}$	$2.06\times 10^{4}$	$5.47\times 10^{4}$	$5.92\times 10^{4}$	$1.21\times 10^{4}$
	Rank	5	4	6	3	2	1
F4	Ave	$5.70\times 10^{4}$	$7.80\times 10^{4}$	$1.71\times 10^{5}$	$1.57\times 10^{3}$	$3.06\times 10^{4}$	$6.29\times 10^{2}$
	Std	$4.71\times 10^{1}$	$1.85\times 10^{2}$	$4.02\times 10^{2}$	$3.83\times 10^{2}$	$5.04\times 10^{3}$	$4.28\times 10^{1}$
	Rank	4	5	6	2	3	1
F5	Ave	$1.60\times 10^{3}$	$1.24\times 10^{3}$	$1.21\times 10^{3}$	$8.95\times 10^{2}$	$1.20\times 10^{3}$	$7.30\times 10^{2}$
	Std	$2.25\times 10^{1}$	$3.25\times 10^{1}$	$4.09\times 10^{1}$	$4.58\times 10^{1}$	$2.23\times 10^{1}$	$2.82\times 10^{1}$
	Rank	6	5	4	2	3	1
F6	Ave	$7.62\times 10^{2}$	$7.67\times 10^{2}$	$7.74\times 10^{2}$	$6.70\times 10^{2}$	$7.06\times 10^{2}$	$6.06\times 10^{2}$
	Std	$4.95\times 10^{0}$	$4.83\times 10^{0}$	$5.03\times 10^{0}$	$8.07\times 10^{0}$	$6.19\times 10^{0}$	$1.05\times 10^{0}$
	Rank	4	5	6	2	3	1
F7	Ave	$1.63\times 10^{4}$	$1.40\times 10^{4}$	$1.76\times 10^{4}$	$1.58\times 10^{3}$	$1.98\times 10^{3}$	$1.08\times 10^{3}$
	Std	$1.24\times 10^{2}$	$1.22\times 10^{2}$	$9.68\times 10^{1}$	$8.65\times 10^{1}$	$5.72\times 10^{1}$	$3.68\times 10^{1}$
	Rank	5	4	6	2	3	1
F8	Ave	$1.66\times 10^{3}$	$1.65\times 10^{3}$	$1.74\times 10^{3}$	$1.21\times 10^{3}$	$1.52\times 10^{3}$	$1.05\times 10^{3}$
	Std	$3.72\times 10^{1}$	$4.13\times 10^{1}$	$3.44\times 10^{1}$	$4.72\times 10^{1}$	$2.96\times 10^{1}$	$2.49\times 10^{1}$
	Rank	5	4	6	2	3	1
F9	Ave	$4.25\times 10^{4}$	$4.13\times 10^{4}$	$4.67\times 10^{4}$	$2.48\times 10^{4}$	$4.01\times 10^{4}$	$8.10\times 10^{3}$
	Std	$1.07\times 10^{3}$	$1.14\times 10^{3}$	$2.94\times 10^{3}$	$4.05\times 10^{3}$	$3.45\times 10^{3}$	$2.31\times 10^{3}$
	Rank	5	4	6	2	3	1
F10	Ave	$1.57\times 10^{4}$	$1.58\times 10^{4}$	$1.85\times 10^{4}$	$1.00\times 10^{4}$	$1.55\times 10^{4}$	$7.58\times 10^{3}$
	Std	$1.07\times 10^{3}$	$9.68\times 10^{2}$	$8.81\times 10^{2}$	$1.19\times 10^{3}$	$6.83\times 10^{2}$	$6.04\times 10^{2}$
	Rank	4	5	6	2	3	1
F11	Ave	$2.32\times 10^{4}$	$2.59\times 10^{4}$	$3.59\times 10^{4}$	$3.26\times 10^{3}$	$2.27\times 10^{4}$	$1.49\times 10^{3}$
	Std	$6.44\times 10^{1}$	$2.22\times 10^{2}$	$7.08\times 10^{2}$	$6.29\times 10^{2}$	$2.18\times 10^{3}$	$1.39\times 10^{2}$
	Rank	4	5	6	2	3	1
F12	Ave	$5.80\times 10^{9}$	$3.26\times 10^{9}$	$8.63\times 10^{10}$	$1.07\times 10^{9}$	$7.24\times 10^{10}$	$1.97\times 10^{7}$
	Std	$4.62\times 10^{6}$	$3.02\times 10^{7}$	$7.27\times 10^{8}$	$5.15\times 10^{8}$	$1.38\times 10^{10}$	$9.26\times 10^{6}$
	Rank	4	3	6	2	5	1
F13	Ave	$3.65\times 10^{4}$	$5.22\times 10^{4}$	$6.12\times 10^{10}$	$6.44\times 10^{7}$	$4.00\times 10^{10}$	$5.25\times 10^{5}$
	Std	$3.46\times 10^{4}$	$4.27\times 10^{4}$	$1.31\times 10^{8}$	$5.44\times 10^{7}$	$1.30\times 10^{10}$	$2.41\times 10^{5}$
	Rank	1	2	6	4	5	3
F14	Ave	$5.94\times 10^{7}$	$5.65\times 10^{7}$	$6.20\times 10^{7}$	$3.84\times 10^{6}$	$5.41\times 10^{7}$	$1.24\times 10^{6}$
	Std	$2.45\times 10^{5}$	$2.83\times 10^{5}$	$1.28\times 10^{7}$	$3.71\times 10^{6}$	$4.12\times 10^{7}$	$1.01\times 10^{6}$
	Rank	5	4	6	2	3	1
F15	Ave	$2.11\times 10^{4}$	$2.93\times 10^{4}$	$1.07\times 10^{10}$	$1.02\times 10^{6}$	$4.65\times 10^{9}$	$6.47\times 10^{4}$
	Std	$3.00\times 10^{4}$	$1.53\times 10^{4}$	$6.77\times 10^{9}$	$6.06\times 10^{5}$	$1.90\times 10^{9}$	$3.82\times 10^{4}$
	Rank	1	2	6	4	5	3
F16	Ave	$9.68\times 10^{3}$	$9.61\times 10^{3}$	$1.42\times 10^{4}$	$4.76\times 10^{3}$	$8.87\times 10^{3}$	$3.25\times 10^{3}$
	Std	$4.70\times 10^{2}$	$5.32\times 10^{2}$	$7.20\times 10^{2}$	$6.01\times 10^{2}$	$1.08\times 10^{3}$	$3.63\times 10^{2}$
	Rank	5	4	6	2	3	1
F17	Ave	$1.55\times 10^{4}$	$1.75\times 10^{4}$	$1.99\times 10^{4}$	$3.84\times 10^{3}$	$1.07\times 10^{4}$	$2.84\times 10^{3}$
	Std	$4.30\times 10^{2}$	$3.37\times 10^{2}$	$4.10\times 10^{2}$	$3.85\times 10^{2}$	$6.23\times 10^{3}$	$3.02\times 10^{2}$
	Rank	4	5	6	2	3	1
F18	Ave	$9.44\times 10^{5}$	$1.73\times 10^{6}$	$1.60\times 10^{8}$	$1.43\times 10^{7}$	$1.24\times 10^{8}$	$3.03\times 10^{6}$
	Std	$8.91\times 10^{5}$	$1.08\times 10^{6}$	$1.01\times 10^{7}$	$8.36\times 10^{6}$	$5.52\times 10^{7}$	$1.84\times 10^{6}$
	Rank	1	2	6	4	5	3
F19	Ave	$2.61\times 10^{4}$	$2.99\times 10^{4}$	$3.82\times 10^{9}$	$3.48\times 10^{6}$	$3.59\times 10^{9}$	$3.79\times 10^{4}$
	Std	$1.46\times 10^{4}$	$1.22\times 10^{4}$	$3.83\times 10^{6}$	$3.27\times 10^{6}$	$1.21\times 10^{9}$	$1.39\times 10^{4}$
	Rank	1	2	6	4	5	3
F20	Ave	$4.49\times 10^{3}$	$4.42\times 10^{3}$	$4.50\times 10^{3}$	$3.36\times 10^{3}$	$4.27\times 10^{3}$	$2.96\times 10^{3}$
	Std	$2.83\times 10^{2}$	$2.60\times 10^{2}$	$3.24\times 10^{2}$	$3.28\times 10^{2}$	$2.61\times 10^{2}$	$2.38\times 10^{2}$
	Rank	5	4	6	2	3	1
F21	Ave	$3.77\times 10^{3}$	$3.65\times 10^{3}$	$3.92\times 10^{3}$	$2.75\times 10^{3}$	$3.21\times 10^{3}$	$2.53\times 10^{3}$
	Std	$7.74\times 10^{1}$	$6.18\times 10^{1}$	$9.80\times 10^{1}$	$5.69\times 10^{1}$	$7.63\times 10^{1}$	$2.79\times 10^{1}$
	Rank	5	4	6	2	3	1
F22	Ave	$1.85\times 10^{4}$	$1.83\times 10^{4}$	$1.96\times 10^{4}$	$1.16\times 10^{4}$	$1.75\times 10^{4}$	$9.26\times 10^{3}$
	Std	$1.02\times 10^{3}$	$9.58\times 10^{2}$	$1.04\times 10^{3}$	$1.71\times 10^{3}$	$5.24\times 10^{2}$	$1.54\times 10^{3}$
	Rank	5	4	6	2	3	1
F23	Ave	$4.57\times 10^{3}$	$4.59\times 10^{3}$	$5.10\times 10^{3}$	$3.53\times 10^{3}$	$4.56\times 10^{3}$	$3.03\times 10^{3}$
	Std	$1.69\times 10^{2}$	$1.18\times 10^{2}$	$1.96\times 10^{2}$	$1.11\times 10^{2}$	$2.37\times 10^{2}$	$3.56\times 10^{1}$
	Rank	4	5	6	2	3	1
F24	Ave	$4.85\times 10^{3}$	$4.87\times 10^{3}$	$4.82\times 10^{3}$	$3.55\times 10^{3}$	$4.79\times 10^{3}$	$3.28\times 10^{3}$
	Std	$1.18\times 10^{2}$	$1.18\times 10^{2}$	$2.66\times 10^{2}$	$1.23\times 10^{2}$	$3.97\times 10^{2}$	$8.21\times 10^{1}$
	Rank	5	6	4	2	3	1
F25	Ave	$1.41\times 10^{4}$	$1.42\times 10^{4}$	$1.79\times 10^{4}$	$3.69\times 10^{3}$	$1.37\times 10^{4}$	$3.15\times 10^{3}$
	Std	$2.90\times 10^{1}$	$6.25\times 10^{1}$	$2.47\times 10^{2}$	$1.54\times 10^{2}$	$1.08\times 10^{3}$	$3.44\times 10^{1}$
	Rank	4	5	6	2	3	1
F26	Ave	$1.72\times 10^{4}$	$1.78\times 10^{4}$	$1.85\times 10^{4}$	$9.01\times 10^{3}$	$1.63\times 10^{4}$	$6.95\times 10^{3}$
	Std	$1.10\times 10^{3}$	$2.74\times 10^{3}$	$1.79\times 10^{3}$	$2.26\times 10^{3}$	$6.08\times 10^{2}$	$3.51\times 10^{2}$
	Rank	4	5	6	2	3	1
F27	Ave	$6.90\times 10^{3}$	$6.85\times 10^{3}$	$7.85\times 10^{3}$	$4.27\times 10^{3}$	$6.14\times 10^{3}$	$3.50\times 10^{3}$
	Std	$2.63\times 10^{2}$	$2.62\times 10^{2}$	$7.37\times 10^{2}$	$2.49\times 10^{2}$	$7.42\times 10^{2}$	$5.97\times 10^{1}$
	Rank	5	4	6	2	3	1
F28	Ave	$3.37\times 10^{3}$	$3.63\times 10^{3}$	$1.40\times 10^{4}$	$4.84\times 10^{3}$	$1.21\times 10^{4}$	$3.44\times 10^{3}$
	Std	$3.86\times 10^{1}$	$1.52\times 10^{2}$	$4.22\times 10^{2}$	$3.59\times 10^{2}$	$6.22\times 10^{2}$	$4.79\times 10^{1}$
	Rank	1	3	6	4	5	2
F29	Ave	$5.42\times 10^{4}$	$5.30\times 10^{4}$	$7.88\times 10^{4}$	$6.71\times 10^{3}$	$3.15\times 10^{4}$	$4.15\times 10^{3}$
	Std	$5.07\times 10^{2}$	$5.04\times 10^{2}$	$1.51\times 10^{3}$	$7.60\times 10^{2}$	$1.37\times 10^{4}$	$2.85\times 10^{2}$
	Rank	5	4	6	2	3	1
F30	Ave	$2.30\times 10^{6}$	$2.10\times 10^{6}$	$8.63\times 10^{9}$	$1.50\times 10^{8}$	$6.75\times 10^{9}$	$1.79\times 10^{6}$
	Std	$4.93\times 10^{5}$	$1.11\times 10^{6}$	$6.51\times 10^{7}$	$4.96\times 10^{7}$	$1.81\times 10^{9}$	$2.65\times 10^{5}$
	Rank	3	2	6	4	5	1

**Table 8 biomimetics-10-00620-t008:** Experimental results of MSAO and AO variants on CEC2017 (100D).

		LOBLAO	TEAO	SGAO	MMSIAO	IAO	MSAO
F1	Ave	$1.70\times 10^{11}$	$1.18\times 10^{11}$	$1.31\times 10^{12}$	$5.58\times 10^{9}$	$1.06\times 10^{11}$	$2.13\times 10^{8}$
	Std	$2.88\times 10^{8}$	$1.08\times 10^{10}$	$7.70\times 10^{9}$	$1.46\times 10^{9}$	$5.48\times 10^{9}$	$4.95\times 10^{7}$
	Rank	5	4	6	2	3	1
F3	Ave	$3.14\times 10^{5}$	$3.82\times 10^{5}$	$3.79\times 10^{5}$	$2.55\times 10^{5}$	$2.64\times 10^{5}$	$9.29\times 10^{4}$
	Std	$4.20\times 10^{4}$	$9.19\times 10^{4}$	$1.15\times 10^{4}$	$6.29\times 10^{4}$	$7.36\times 10^{4}$	$1.90\times 10^{4}$
	Rank	4	6	5	2	3	1
F4	Ave	$4.05\times 10^{4}$	$3.75\times 10^{4}$	$4.97\times 10^{4}$	$1.56\times 10^{3}$	$3.07\times 10^{4}$	$6.27\times 10^{2}$
	Std	$7.99\times 10^{1}$	$1.31\times 10^{3}$	$1.47\times 10^{3}$	$3.45\times 10^{2}$	$4.45\times 10^{3}$	$5.02\times 10^{1}$
	Rank	5	4	6	2	3	1
F5	Ave	$1.34\times 10^{3}$	$2.38\times 10^{3}$	$2.58\times 10^{3}$	$8.92\times 10^{2}$	$1.20\times 10^{3}$	$7.32\times 10^{2}$
	Std	$5.84\times 10^{1}$	$5.85\times 10^{1}$	$6.23\times 10^{1}$	$3.87\times 10^{1}$	$2.86\times 10^{1}$	$2.81\times 10^{1}$
	Rank	4	5	6	2	3	1
F6	Ave	$7.65\times 10^{2}$	$7.67\times 10^{2}$	$7.84\times 10^{2}$	$6.69\times 10^{2}$	$7.07\times 10^{2}$	$6.06\times 10^{2}$
	Std	$3.08\times 10^{0}$	$4.13\times 10^{0}$	$4.13\times 10^{0}$	$5.86\times 10^{0}$	$4.99\times 10^{0}$	$1.11\times 10^{0}$
	Rank	4	5	6	2	3	1
F7	Ave	$2.14\times 10^{3}$	$2.05\times 10^{3}$	$3.42\times 10^{3}$	$1.61\times 10^{3}$	$1.98\times 10^{3}$	$1.08\times 10^{3}$
	Std	$1.69\times 10^{2}$	$1.52\times 10^{2}$	$1.77\times 10^{2}$	$9.64\times 10^{1}$	$5.26\times 10^{1}$	$2.99\times 10^{1}$
	Rank	5	4	6	2	3	1
F8	Ave	$1.91\times 10^{3}$	$1.80\times 10^{3}$	$2.06\times 10^{3}$	$1.22\times 10^{3}$	$1.53\times 10^{3}$	$1.04\times 10^{3}$
	Std	$7.10\times 10^{1}$	$7.85\times 10^{1}$	$6.38\times 10^{1}$	$4.28\times 10^{1}$	$3.03\times 10^{1}$	$2.68\times 10^{1}$
	Rank	5	4	6	2	3	1
F9	Ave	$3.46\times 10^{4}$	$2.78\times 10^{4}$	$5.75\times 10^{4}$	$2.55\times 10^{4}$	$3.96\times 10^{4}$	$7.54\times 10^{3}$
	Std	$1.39\times 10^{3}$	$2.42\times 10^{3}$	$4.78\times 10^{3}$	$4.16\times 10^{3}$	$3.04\times 10^{3}$	$2.21\times 10^{3}$
	Rank	4	3	6	2	5	1
F10	Ave	$2.76\times 10^{4}$	$2.74\times 10^{4}$	$3.13\times 10^{4}$	$9.92\times 10^{3}$	$1.55\times 10^{4}$	$7.93\times 10^{3}$
	Std	$1.42\times 10^{3}$	$1.31\times 10^{3}$	$2.15\times 10^{3}$	$8.08\times 10^{2}$	$7.18\times 10^{2}$	$6.83\times 10^{2}$
	Rank	5	4	6	2	3	1
F11	Ave	$2.94\times 10^{4}$	$2.52\times 10^{4}$	$2.20\times 10^{4}$	$3.28\times 10^{3}$	$2.20\times 10^{4}$	$1.62\times 10^{3}$
	Std	$1.12\times 10^{4}$	$4.03\times 10^{4}$	$4.38\times 10^{4}$	$5.75\times 10^{2}$	$1.88\times 10^{3}$	$3.79\times 10^{2}$
	Rank	6	5	3	2	3	1
F12	Ave	$8.39\times 10^{10}$	$1.95\times 10^{10}$	$1.35\times 10^{11}$	$1.19\times 10^{9}$	$7.74\times 10^{10}$	$2.01\times 10^{7}$
	Std	$4.15\times 10^{7}$	$2.42\times 10^{9}$	$3.76\times 10^{9}$	$6.32\times 10^{8}$	$1.16\times 10^{10}$	$6.67\times 10^{6}$
	Rank	5	3	6	2	4	1
F13	Ave	$6.02\times 10^{4}$	$3.25\times 10^{6}$	$4.88\times 10^{10}$	$9.10\times 10^{7}$	$3.94\times 10^{10}$	$6.52\times 10^{5}$
	Std	$5.91\times 10^{4}$	$1.17\times 10^{7}$	$2.92\times 10^{8}$	$1.60\times 10^{8}$	$1.25\times 10^{10}$	$3.36\times 10^{5}$
	Rank	1	3	6	4	5	2
F14	Ave	$1.56\times 10^{7}$	$5.01\times 10^{7}$	$9.39\times 10^{7}$	$5.47\times 10^{6}$	$4.52\times 10^{7}$	$1.33\times 10^{6}$
	Std	$8.40\times 10^{5}$	$1.23\times 10^{6}$	$4.57\times 10^{6}$	$4.15\times 10^{6}$	$3.18\times 10^{7}$	$9.42\times 10^{5}$
	Rank	3	5	6	2	4	1
F15	Ave	$2.02\times 10^{5}$	$7.53\times 10^{5}$	$3.17\times 10^{10}$	$1.33\times 10^{6}$	$5.22\times 10^{9}$	$9.06\times 10^{4}$
	Std	$2.03\times 10^{4}$	$2.23\times 10^{5}$	$1.67\times 10^{7}$	$1.05\times 10^{6}$	$2.66\times 10^{9}$	$4.91\times 10^{4}$
	Rank	2	3	6	4	5	1
F16	Ave	$1.29\times 10^{4}$	$1.25\times 10^{4}$	$1.55\times 10^{4}$	$4.66\times 10^{3}$	$8.45\times 10^{3}$	$3.25\times 10^{3}$
	Std	$6.97\times 10^{2}$	$9.29\times 10^{2}$	$1.28\times 10^{3}$	$4.67\times 10^{2}$	$9.51\times 10^{2}$	$3.88\times 10^{2}$
	Rank	5	4	6	2	3	1
F17	Ave	$9.08\times 10^{3}$	$6.45\times 10^{3}$	$1.09\times 10^{7}$	$3.75\times 10^{3}$	$8.82\times 10^{3}$	$2.87\times 10^{3}$
	Std	$7.72\times 10^{2}$	$8.82\times 10^{2}$	$4.72\times 10^{3}$	$4.52\times 10^{2}$	$3.58\times 10^{3}$	$2.64\times 10^{2}$
	Rank	5	3	6	2	4	1
F18	Ave	$1.79\times 10^{8}$	$1.87\times 10^{8}$	$1.98\times 10^{8}$	$1.42\times 10^{7}$	$1.25\times 10^{8}$	$3.65\times 10^{6}$
	Std	$1.06\times 10^{6}$	$1.83\times 10^{6}$	$6.32\times 10^{6}$	$8.90\times 10^{6}$	$5.86\times 10^{7}$	$2.83\times 10^{6}$
	Rank	4	5	6	2	3	1
F19	Ave	$2.99\times 10^{7}$	$2.42\times 10^{7}$	$1.94\times 10^{7}$	$2.26\times 10^{6}$	$3.67\times 10^{9}$	$3.74\times 10^{4}$
	Std	$5.40\times 10^{4}$	$5.71\times 10^{5}$	$3.28\times 10^{7}$	$1.97\times 10^{6}$	$1.38\times 10^{9}$	$1.39\times 10^{4}$
	Rank	5	4	3	2	6	1
F20	Ave	$5.93\times 10^{3}$	$5.96\times 10^{3}$	$6.02\times 10^{3}$	$3.29\times 10^{3}$	$4.23\times 10^{3}$	$2.96\times 10^{3}$
	Std	$6.40\times 10^{2}$	$6.18\times 10^{2}$	$4.26\times 10^{2}$	$2.71\times 10^{2}$	$2.35\times 10^{2}$	$2.74\times 10^{2}$
	Rank	4	5	6	2	3	1
F21	Ave	$5.57\times 10^{3}$	$5.61\times 10^{3}$	$6.36\times 10^{3}$	$2.76\times 10^{3}$	$3.22\times 10^{3}$	$2.53\times 10^{3}$
	Std	$2.25\times 10^{2}$	$1.99\times 10^{2}$	$1.70\times 10^{2}$	$7.12\times 10^{1}$	$5.66\times 10^{1}$	$3.00\times 10^{1}$
	Rank	4	5	6	2	3	1
F22	Ave	$3.59\times 10^{4}$	$3.66\times 10^{4}$	$4.54\times 10^{4}$	$1.17\times 10^{4}$	$1.73\times 10^{4}$	$9.57\times 10^{3}$
	Std	$1.25\times 10^{3}$	$1.34\times 10^{3}$	$1.67\times 10^{3}$	$2.32\times 10^{3}$	$6.99\times 10^{2}$	$6.99\times 10^{2}$
	Rank	4	5	6	2	3	1
F23	Ave	$7.18\times 10^{3}$	$7.15\times 10^{3}$	$7.68\times 10^{3}$	$3.53\times 10^{3}$	$4.52\times 10^{3}$	$3.03\times 10^{3}$
	Std	$1.98\times 10^{2}$	$1.79\times 10^{2}$	$3.18\times 10^{2}$	$1.56\times 10^{2}$	$2.23\times 10^{2}$	$6.49\times 10^{1}$
	Rank	5	4	6	2	3	1
F24	Ave	$5.14\times 10^{4}$	$5.08\times 10^{4}$	$8.37\times 10^{4}$	$3.60\times 10^{3}$	$4.80\times 10^{3}$	$3.28\times 10^{3}$
	Std	$2.95\times 10^{2}$	$3.06\times 10^{2}$	$8.35\times 10^{2}$	$1.11\times 10^{2}$	$4.73\times 10^{2}$	$6.28\times 10^{1}$
	Rank	5	4	6	2	3	1
F25	Ave	$3.68\times 10^{4}$	$4.90\times 10^{4}$	$6.71\times 10^{4}$	$3.71\times 10^{3}$	$1.40\times 10^{4}$	$3.15\times 10^{3}$
	Std	$8.19\times 10^{1}$	$5.53\times 10^{2}$	$5.25\times 10^{2}$	$1.73\times 10^{2}$	$8.76\times 10^{2}$	$3.39\times 10^{1}$
	Rank	4	5	6	2	3	1
F26	Ave	$5.54\times 10^{4}$	$5.71\times 10^{4}$	$6.17\times 10^{4}$	$9.06\times 10^{3}$	$1.63\times 10^{4}$	$6.91\times 10^{3}$
	Std	$3.40\times 10^{3}$	$4.77\times 10^{3}$	$1.78\times 10^{3}$	$2.40\times 10^{3}$	$7.72\times 10^{2}$	$3.50\times 10^{2}$
	Rank	4	5	6	2	3	1
F27	Ave	$7.13\times 10^{3}$	$8.25\times 10^{3}$	$8.23\times 10^{3}$	$4.31\times 10^{3}$	$6.49\times 10^{3}$	$3.50\times 10^{3}$
	Std	$2.54\times 10^{2}$	$2.80\times 10^{2}$	$8.72\times 10^{2}$	$2.32\times 10^{2}$	$7.03\times 10^{2}$	$7.59\times 10^{1}$
	Rank	4	6	5	2	3	1
F28	Ave	$1.89\times 10^{4}$	$1.30\times 10^{4}$	$9.19\times 10^{4}$	$4.72\times 10^{3}$	$1.21\times 10^{4}$	$3.43\times 10^{3}$
	Std	$1.11\times 10^{2}$	$1.11\times 10^{3}$	$8.66\times 10^{2}$	$3.39\times 10^{2}$	$8.51\times 10^{2}$	$4.62\times 10^{1}$
	Rank	5	3	6	2	4	1
F29	Ave	$8.08\times 10^{5}$	$9.05\times 10^{4}$	$1.36\times 10^{6}$	$7.17\times 10^{3}$	$4.06\times 10^{4}$	$4.04\times 10^{3}$
	Std	$7.75\times 10^{2}$	$8.97\times 10^{2}$	$1.25\times 10^{3}$	$9.37\times 10^{2}$	$3.35\times 10^{4}$	$2.51\times 10^{2}$
	Rank	5	4	6	2	3	1
F30	Ave	$1.36\times 10^{11}$	$9.55\times 10^{7}$	$1.06\times 10^{11}$	$1.42\times 10^{8}$	$5.42\times 10^{9}$	$1.84\times 10^{6}$
	Std	$1.21\times 10^{6}$	$3.42\times 10^{8}$	$4.84\times 10^{9}$	$4.41\times 10^{7}$	$1.58\times 10^{9}$	$3.36\times 10^{5}$
	Rank	6	2	5	3	4	1

**Table 9 biomimetics-10-00620-t009:** Experimental results of MSAO and other algorithms on CEC2017 (10D).

		WOA	HHO	AOO	DOA	LSHADE	AO	LOBLAO	TEAO	MSAO
F1	Ave	$2.56\times 10^{5}$	$5.71\times 10^{5}$	$2.92\times 10^{3}$	$1.09\times 10^{5}$	$3.83\times 10^{3}$	$8.08\times 10^{5}$	$8.25\times 10^{5}$	$7.50\times 10^{9}$	$9.58\times 10^{2}$
	Std	$1.20\times 10^{6}$	$2.56\times 10^{5}$	$2.35\times 10^{3}$	$9.80\times 10^{4}$	$1.93\times 10^{4}$	$4.65\times 10^{5}$	$7.27\times 10^{5}$	$2.50\times 10^{9}$	$1.67\times 10^{3}$
	Rank	5	6	2	4	3	7	8	9	1
F3	Ave	$4.39\times 10^{2}$	$3.06\times 10^{2}$	$3.00\times 10^{2}$	$3.36\times 10^{2}$	$5.90\times 10^{2}$	$7.51\times 10^{2}$	$6.04\times 10^{2}$	$9.19\times 10^{3}$	$3.00\times 10^{2}$
	Std	$1.17\times 10^{2}$	$7.13\times 10^{0}$	$1.12\times 10^{-4}$	$2.76\times 10^{1}$	$7.45\times 10^{2}$	$3.00\times 10^{2}$	$1.93\times 10^{2}$	$2.64\times 10^{3}$	$5.51\times 10^{-2}$
	Rank	5	3	1	4	6	8	7	9	2
F4	Ave	$4.18\times 10^{2}$	$4.19\times 10^{2}$	$4.05\times 10^{2}$	$4.03\times 10^{2}$	$4.06\times 10^{2}$	$4.16\times 10^{2}$	$4.21\times 10^{2}$	$8.19\times 10^{2}$	$4.02\times 10^{2}$
	Std	$2.97\times 10^{1}$	$2.86\times 10^{1}$	$1.76\times 10^{0}$	$2.79\times 10^{0}$	$1.14\times 10^{0}$	$2.14\times 10^{1}$	$2.29\times 10^{1}$	$3.33\times 10^{2}$	$2.07\times 10^{0}$
	Rank	6	7	3	2	4	5	8	9	1
F5	Ave	$5.23\times 10^{2}$	$5.54\times 10^{2}$	$5.19\times 10^{2}$	$5.12\times 10^{2}$	$5.18\times 10^{2}$	$5.27\times 10^{2}$	$5.28\times 10^{2}$	$5.91\times 10^{2}$	$5.07\times 10^{2}$
	Std	$7.94\times 10^{0}$	$1.69\times 10^{1}$	$7.54\times 10^{0}$	$5.43\times 10^{0}$	$7.32\times 10^{0}$	$8.23\times 10^{0}$	$1.07\times 10^{1}$	$1.57\times 10^{1}$	$2.32\times 10^{0}$
	Rank	5	8	4	2	3	6	7	9	1
F6	Ave	$6.04\times 10^{2}$	$6.39\times 10^{2}$	$6.02\times 10^{2}$	$6.00\times 10^{2}$	$6.00\times 10^{2}$	$6.17\times 10^{2}$	$6.18\times 10^{2}$	$6.48\times 10^{2}$	$6.00\times 10^{2}$
	Std	$3.90\times 10^{0}$	$9.47\times 10^{0}$	$2.43\times 10^{0}$	$8.87\times 10^{-2}$	$8.08\times 10^{-5}$	$6.59\times 10^{0}$	$6.17\times 10^{0}$	$6.88\times 10^{0}$	$1.32\times 10^{-3}$
	Rank	5	8	4	3	1	6	7	9	2
F7	Ave	$7.43\times 10^{2}$	$7.89\times 10^{2}$	$7.30\times 10^{2}$	$7.21\times 10^{2}$	$7.30\times 10^{2}$	$7.50\times 10^{2}$	$7.52\times 10^{2}$	$8.15\times 10^{2}$	$7.19\times 10^{2}$
	Std	$1.51\times 10^{1}$	$1.96\times 10^{1}$	$9.32\times 10^{0}$	$3.66\times 10^{0}$	$6.17\times 10^{0}$	$1.38\times 10^{1}$	$1.50\times 10^{1}$	$1.23\times 10^{1}$	$4.23\times 10^{0}$
	Rank	5	8	3	2	4	6	7	9	1
F8	Ave	$8.24\times 10^{2}$	$8.32\times 10^{2}$	$8.21\times 10^{2}$	$8.09\times 10^{2}$	$8.18\times 10^{2}$	$8.25\times 10^{2}$	$8.24\times 10^{2}$	$8.59\times 10^{2}$	$8.08\times 10^{2}$
	Std	$1.09\times 10^{1}$	$8.09\times 10^{0}$	$8.08\times 10^{0}$	$2.68\times 10^{0}$	$6.73\times 10^{0}$	$9.05\times 10^{0}$	$9.34\times 10^{0}$	$8.29\times 10^{0}$	$2.69\times 10^{0}$
	Rank	5	8	4	2	3	7	6	9	1
F9	Ave	$9.69\times 10^{2}$	$1.44\times 10^{3}$	$9.01\times 10^{2}$	$9.00\times 10^{2}$	$9.00\times 10^{2}$	$1.03\times 10^{3}$	$1.02\times 10^{3}$	$1.42\times 10^{3}$	$9.00\times 10^{2}$
	Std	$1.44\times 10^{2}$	$2.35\times 10^{2}$	$1.80\times 10^{0}$	$2.67\times 10^{-1}$	$1.15\times 10^{-1}$	$7.87\times 10^{1}$	$8.86\times 10^{1}$	$1.78\times 10^{2}$	$1.63\times 10^{-2}$
	Rank	5	9	4	3	2	7	6	8	1
F10	Ave	$1.68\times 10^{3}$	$2.06\times 10^{3}$	$1.67\times 10^{3}$	$1.33\times 10^{3}$	$2.08\times 10^{3}$	$1.84\times 10^{3}$	$1.89\times 10^{3}$	$2.82\times 10^{3}$	$1.31\times 10^{3}$
	Std	$3.15\times 10^{2}$	$3.09\times 10^{2}$	$2.74\times 10^{2}$	$1.83\times 10^{2}$	$2.23\times 10^{2}$	$2.33\times 10^{2}$	$2.79\times 10^{2}$	$2.43\times 10^{2}$	$1.45\times 10^{2}$
	Rank	4	7	3	2	8	5	6	9	1
F11	Ave	$1.13\times 10^{3}$	$1.17\times 10^{3}$	$1.13\times 10^{3}$	$1.11\times 10^{3}$	$1.10\times 10^{3}$	$1.19\times 10^{3}$	$1.19\times 10^{3}$	$1.70\times 10^{3}$	$1.11\times 10^{3}$
	Std	$2.79\times 10^{1}$	$5.79\times 10^{1}$	$2.84\times 10^{1}$	$4.47\times 10^{0}$	$1.36\times 10^{0}$	$7.41\times 10^{1}$	$5.25\times 10^{1}$	$1.08\times 10^{3}$	$2.51\times 10^{0}$
	Rank	4	6	5	3	1	8	7	9	2
F12	Ave	$8.32\times 10^{5}$	$3.81\times 10^{6}$	$1.06\times 10^{6}$	$2.16\times 10^{4}$	$2.69\times 10^{4}$	$4.54\times 10^{6}$	$3.68\times 10^{6}$	$1.28\times 10^{8}$	$1.67\times 10^{4}$
	Std	$8.67\times 10^{5}$	$3.83\times 10^{6}$	$1.36\times 10^{6}$	$2.07\times 10^{4}$	$4.95\times 10^{4}$	$4.74\times 10^{6}$	$3.40\times 10^{6}$	$1.31\times 10^{8}$	$1.24\times 10^{4}$
	Rank	4	7	5	2	3	8	6	9	1
F13	Ave	$1.36\times 10^{4}$	$1.22\times 10^{4}$	$1.25\times 10^{4}$	$7.72\times 10^{3}$	$1.31\times 10^{3}$	$1.50\times 10^{4}$	$1.63\times 10^{4}$	$1.23\times 10^{4}$	$2.85\times 10^{3}$
	Std	$1.18\times 10^{4}$	$8.46\times 10^{3}$	$9.08\times 10^{3}$	$5.25\times 10^{3}$	$3.58\times 10^{0}$	$9.16\times 10^{3}$	$1.15\times 10^{4}$	$1.02\times 10^{4}$	$2.72\times 10^{3}$
	Rank	7	4	6	3	1	8	9	5	2
F14	Ave	$1.61\times 10^{3}$	$1.63\times 10^{3}$	$1.50\times 10^{3}$	$3.90\times 10^{3}$	$1.42\times 10^{3}$	$2.17\times 10^{3}$	$2.27\times 10^{3}$	$1.79\times 10^{3}$	$1.80\times 10^{3}$
	Std	$6.33\times 10^{2}$	$1.53\times 10^{2}$	$3.93\times 10^{1}$	$3.04\times 10^{3}$	$6.07\times 10^{0}$	$9.49\times 10^{2}$	$1.16\times 10^{3}$	$1.04\times 10^{3}$	$5.14\times 10^{2}$
	Rank	3	4	2	9	1	7	8	5	6
F15	Ave	$2.77\times 10^{3}$	$6.88\times 10^{3}$	$1.92\times 10^{3}$	$2.14\times 10^{3}$	$1.50\times 10^{3}$	$5.54\times 10^{3}$	$5.23\times 10^{3}$	$5.63\times 10^{3}$	$2.20\times 10^{3}$
	Std	$1.48\times 10^{3}$	$2.00\times 10^{3}$	$3.28\times 10^{2}$	$6.68\times 10^{2}$	$9.74\times 10^{-1}$	$2.77\times 10^{3}$	$2.79\times 10^{3}$	$2.88\times 10^{3}$	$8.52\times 10^{2}$
	Rank	5	9	2	3	1	7	6	8	4
F16	Ave	$1.78\times 10^{3}$	$1.92\times 10^{3}$	$1.72\times 10^{3}$	$1.70\times 10^{3}$	$1.61\times 10^{3}$	$1.79\times 10^{3}$	$1.81\times 10^{3}$	$2.05\times 10^{3}$	$1.66\times 10^{3}$
	Std	$1.32\times 10^{2}$	$1.31\times 10^{2}$	$1.02\times 10^{2}$	$1.12\times 10^{2}$	$1.09\times 10^{1}$	$1.24\times 10^{2}$	$1.30\times 10^{2}$	$1.25\times 10^{2}$	$6.13\times 10^{1}$
	Rank	5	8	4	3	1	6	7	9	2
F17	Ave	$1.76\times 10^{3}$	$1.78\times 10^{3}$	$1.76\times 10^{3}$	$1.71\times 10^{3}$	$1.72\times 10^{3}$	$1.77\times 10^{3}$	$1.77\times 10^{3}$	$1.82\times 10^{3}$	$1.71\times 10^{3}$
	Std	$4.66\times 10^{1}$	$3.82\times 10^{1}$	$2.05\times 10^{1}$	$8.97\times 10^{0}$	$9.36\times 10^{0}$	$2.30\times 10^{1}$	$2.17\times 10^{1}$	$3.69\times 10^{1}$	$9.23\times 10^{0}$
	Rank	5	8	4	2	3	7	6	9	1
F18	Ave	$1.93\times 10^{4}$	$1.88\times 10^{4}$	$1.79\times 10^{4}$	$3.92\times 10^{3}$	$1.82\times 10^{3}$	$2.86\times 10^{4}$	$2.84\times 10^{4}$	$1.86\times 10^{4}$	$4.19\times 10^{3}$
	Std	$1.22\times 10^{4}$	$1.16\times 10^{4}$	$1.12\times 10^{4}$	$1.70\times 10^{3}$	$6.27\times 10^{0}$	$1.33\times 10^{4}$	$1.46\times 10^{4}$	$1.45\times 10^{4}$	$2.15\times 10^{3}$
	Rank	7	6	4	2	1	9	8	5	3
F19	Ave	$9.72\times 10^{3}$	$1.10\times 10^{4}$	$2.96\times 10^{3}$	$4.26\times 10^{3}$	$1.90\times 10^{3}$	$8.18\times 10^{3}$	$1.48\times 10^{4}$	$2.56\times 10^{4}$	$2.60\times 10^{3}$
	Std	$7.98\times 10^{3}$	$9.40\times 10^{3}$	$1.66\times 10^{3}$	$2.84\times 10^{3}$	$5.14\times 10^{-1}$	$5.60\times 10^{3}$	$3.28\times 10^{4}$	$4.06\times 10^{4}$	$1.21\times 10^{3}$
	Rank	6	7	3	4	1	5	8	9	2
F20	Ave	$2.05\times 10^{3}$	$2.19\times 10^{3}$	$2.07\times 10^{3}$	$2.00\times 10^{3}$	$2.01\times 10^{3}$	$2.11\times 10^{3}$	$2.12\times 10^{3}$	$2.23\times 10^{3}$	$2.00\times 10^{3}$
	Std	$3.86\times 10^{1}$	$8.35\times 10^{1}$	$4.56\times 10^{1}$	$1.63\times 10^{0}$	$9.19\times 10^{0}$	$6.33\times 10^{1}$	$5.71\times 10^{1}$	$5.72\times 10^{1}$	$4.34\times 10^{0}$
	Rank	4	8	5	2	3	6	7	9	1
F21	Ave	$2.33\times 10^{3}$	$2.31\times 10^{3}$	$2.30\times 10^{3}$	$2.29\times 10^{3}$	$2.31\times 10^{3}$	$2.31\times 10^{3}$	$2.28\times 10^{3}$	$2.33\times 10^{3}$	$2.28\times 10^{3}$
	Std	$2.56\times 10^{1}$	$6.97\times 10^{1}$	$5.56\times 10^{1}$	$5.24\times 10^{1}$	$3.72\times 10^{1}$	$4.14\times 10^{1}$	$5.67\times 10^{1}$	$5.05\times 10^{1}$	$5.30\times 10^{1}$
	Rank	8	6	4	3	5	7	2	9	1
F22	Ave	$2.30\times 10^{3}$	$2.45\times 10^{3}$	$2.30\times 10^{3}$	$2.29\times 10^{3}$	$2.30\times 10^{3}$	$2.31\times 10^{3}$	$2.31\times 10^{3}$	$2.81\times 10^{3}$	$2.29\times 10^{3}$
	Std	$1.74\times 10^{1}$	$3.73\times 10^{2}$	$1.36\times 10^{1}$	$3.47\times 10^{1}$	$9.05\times 10^{0}$	$8.41\times 10^{0}$	$7.41\times 10^{0}$	$2.02\times 10^{2}$	$2.39\times 10^{1}$
	Rank	3	8	4	1	5	6	7	9	2
F23	Ave	$2.63\times 10^{3}$	$2.68\times 10^{3}$	$2.62\times 10^{3}$	$2.62\times 10^{3}$	$2.63\times 10^{3}$	$2.64\times 10^{3}$	$2.64\times 10^{3}$	$2.73\times 10^{3}$	$2.61\times 10^{3}$
	Std	$1.11\times 10^{1}$	$2.77\times 10^{1}$	$6.28\times 10^{0}$	$7.19\times 10^{0}$	$7.65\times 10^{0}$	$1.29\times 10^{1}$	$1.40\times 10^{1}$	$3.43\times 10^{1}$	$4.14\times 10^{0}$
	Rank	5	8	3	2	4	6	7	9	1
F24	Ave	$2.75\times 10^{3}$	$2.84\times 10^{3}$	$2.73\times 10^{3}$	$2.69\times 10^{3}$	$2.75\times 10^{3}$	$2.76\times 10^{3}$	$2.74\times 10^{3}$	$2.82\times 10^{3}$	$2.65\times 10^{3}$
	Std	$4.90\times 10^{1}$	$8.38\times 10^{1}$	$6.39\times 10^{1}$	$9.94\times 10^{1}$	$7.44\times 10^{0}$	$5.04\times 10^{1}$	$8.25\times 10^{1}$	$7.39\times 10^{1}$	$1.21\times 10^{2}$
	Rank	6	9	3	2	5	7	4	8	1
F25	Ave	$2.94\times 10^{3}$	$2.93\times 10^{3}$	$2.91\times 10^{3}$	$2.90\times 10^{3}$	$2.93\times 10^{3}$	$2.94\times 10^{3}$	$2.92\times 10^{3}$	$3.21\times 10^{3}$	$2.89\times 10^{3}$
	Std	$1.92\times 10^{1}$	$4.61\times 10^{1}$	$6.25\times 10^{1}$	$7.85\times 10^{1}$	$2.17\times 10^{1}$	$2.01\times 10^{1}$	$5.98\times 10^{1}$	$1.28\times 10^{2}$	$7.38\times 10^{1}$
	Rank	8	5	3	2	6	7	4	9	1
F26	Ave	$3.22\times 10^{3}$	$3.75\times 10^{3}$	$3.01\times 10^{3}$	$2.81\times 10^{3}$	$2.99\times 10^{3}$	$2.98\times 10^{3}$	$3.03\times 10^{3}$	$3.99\times 10^{3}$	$2.87\times 10^{3}$
	Std	$4.84\times 10^{2}$	$6.43\times 10^{2}$	$3.56\times 10^{2}$	$1.40\times 10^{2}$	$1.33\times 10^{2}$	$2.12\times 10^{2}$	$1.73\times 10^{2}$	$3.08\times 10^{2}$	$9.56\times 10^{1}$
	Rank	7	8	5	1	4	3	6	9	2
F27	Ave	$3.11\times 10^{3}$	$3.18\times 10^{3}$	$3.10\times 10^{3}$	$3.10\times 10^{3}$	$3.07\times 10^{3}$	$3.10\times 10^{3}$	$3.10\times 10^{3}$	$3.18\times 10^{3}$	$3.09\times 10^{3}$
	Std	$2.80\times 10^{1}$	$5.74\times 10^{1}$	$1.97\times 10^{1}$	$8.36\times 10^{0}$	$6.66\times 10^{-1}$	$5.01\times 10^{0}$	$5.74\times 10^{0}$	$2.75\times 10^{1}$	$2.46\times 10^{0}$
	Rank	7	8	4	3	1	5	6	9	2
F28	Ave	$3.39\times 10^{3}$	$3.43\times 10^{3}$	$3.28\times 10^{3}$	$3.24\times 10^{3}$	$3.27\times 10^{3}$	$3.39\times 10^{3}$	$3.38\times 10^{3}$	$3.70\times 10^{3}$	$3.16\times 10^{3}$
	Std	$2.00\times 10^{2}$	$1.35\times 10^{2}$	$1.64\times 10^{2}$	$1.35\times 10^{2}$	$3.84\times 10^{0}$	$7.88\times 10^{1}$	$1.02\times 10^{2}$	$1.51\times 10^{2}$	$1.13\times 10^{2}$
	Rank	7	8	4	2	3	6	5	9	1
F29	Ave	$3.26\times 10^{3}$	$3.30\times 10^{3}$	$3.19\times 10^{3}$	$3.19\times 10^{3}$	$3.19\times 10^{3}$	$3.23\times 10^{3}$	$3.23\times 10^{3}$	$3.43\times 10^{3}$	$3.16\times 10^{3}$
	Std	$6.89\times 10^{1}$	$7.10\times 10^{1}$	$3.42\times 10^{1}$	$2.97\times 10^{1}$	$2.42\times 10^{1}$	$4.99\times 10^{1}$	$4.64\times 10^{1}$	$1.14\times 10^{2}$	$1.77\times 10^{1}$
	Rank	7	8	4	3	2	6	5	9	1
F30	Ave	$2.00\times 10^{5}$	$9.86\times 10^{5}$	$4.01\times 10^{5}$	$6.92\times 10^{4}$	$3.28\times 10^{3}$	$7.33\times 10^{5}$	$5.69\times 10^{5}$	$6.03\times 10^{6}$	$3.81\times 10^{4}$
	Std	$5.25\times 10^{5}$	$1.10\times 10^{6}$	$5.26\times 10^{5}$	$6.09\times 10^{4}$	$1.50\times 10^{2}$	$1.16\times 10^{6}$	$8.01\times 10^{5}$	$5.67\times 10^{6}$	$4.33\times 10^{4}$
	Rank	4	8	5	3	1	7	6	9	2

**Table 10 biomimetics-10-00620-t010:** Experimental results of MSAO and other algorithms on CEC2017 (30D).

		WOA	HHO	AOO	DOA	LSHADE	AO	LOBLAO	TEAO	MSAO
F1	Ave	$6.35\times 10^{8}$	$2.87\times 10^{7}$	$7.88\times 10^{3}$	$9.03\times 10^{6}$	$4.51\times 10^{6}$	$5.20\times 10^{8}$	$5.02\times 10^{8}$	$4.59\times 10^{10}$	$1.20\times 10^{4}$
	Std	$5.96\times 10^{8}$	$7.29\times 10^{6}$	$5.63\times 10^{3}$	$3.29\times 10^{6}$	$1.75\times 10^{7}$	$2.61\times 10^{8}$	$2.24\times 10^{8}$	$4.90\times 10^{9}$	$5.14\times 10^{3}$
	Rank	8	5	1	4	3	7	6	9	2
F3	Ave	$5.54\times 10^{4}$	$3.82\times 10^{4}$	$4.93\times 10^{3}$	$1.70\times 10^{4}$	$2.11\times 10^{5}$	$5.64\times 10^{4}$	$5.46\times 10^{4}$	$8.23\times 10^{4}$	$5.51\times 10^{4}$
	Std	$1.32\times 10^{4}$	$6.30\times 10^{3}$	$1.91\times 10^{3}$	$7.24\times 10^{3}$	$6.41\times 10^{4}$	$8.23\times 10^{3}$	$5.97\times 10^{3}$	$5.29\times 10^{3}$	$1.70\times 10^{4}$
	Rank	6	3	1	2	9	7	4	8	5
F4	Ave	$5.87\times 10^{2}$	$5.65\times 10^{2}$	$4.96\times 10^{2}$	$5.09\times 10^{2}$	$4.43\times 10^{2}$	$6.63\times 10^{2}$	$6.48\times 10^{2}$	$1.22\times 10^{4}$	$4.76\times 10^{2}$
	Std	$5.28\times 10^{1}$	$3.59\times 10^{1}$	$1.73\times 10^{1}$	$2.13\times 10^{1}$	$2.52\times 10^{1}$	$7.73\times 10^{1}$	$5.82\times 10^{1}$	$1.56\times 10^{3}$	$2.45\times 10^{1}$
	Rank	6	5	3	4	1	8	7	9	2
F5	Ave	$7.04\times 10^{2}$	$7.62\times 10^{2}$	$6.11\times 10^{2}$	$5.99\times 10^{2}$	$6.91\times 10^{2}$	$7.09\times 10^{2}$	$7.10\times 10^{2}$	$9.32\times 10^{2}$	$5.59\times 10^{2}$
	Std	$4.54\times 10^{1}$	$3.25\times 10^{1}$	$2.38\times 10^{1}$	$2.39\times 10^{1}$	$1.89\times 10^{1}$	$3.53\times 10^{1}$	$4.07\times 10^{1}$	$2.55\times 10^{1}$	$1.06\times 10^{1}$
	Rank	5	8	3	2	4	6	7	9	1
F6	Ave	$6.47\times 10^{2}$	$6.65\times 10^{2}$	$6.25\times 10^{2}$	$6.02\times 10^{2}$	$6.00\times 10^{2}$	$6.56\times 10^{2}$	$6.52\times 10^{2}$	$6.94\times 10^{2}$	$6.00\times 10^{2}$
	Std	$1.15\times 10^{1}$	$5.41\times 10^{0}$	$8.27\times 10^{0}$	$4.52\times 10^{-1}$	$8.96\times 10^{-2}$	$7.79\times 10^{0}$	$6.79\times 10^{0}$	$6.21\times 10^{0}$	$5.79\times 10^{-2}$
	Rank	5	8	4	3	1	7	6	9	2
F7	Ave	$1.03\times 10^{3}$	$1.27\times 10^{3}$	$8.58\times 10^{2}$	$8.51\times 10^{2}$	$9.20\times 10^{2}$	$1.11\times 10^{3}$	$1.09\times 10^{3}$	$1.38\times 10^{3}$	$7.98\times 10^{2}$
	Std	$7.09\times 10^{1}$	$5.92\times 10^{1}$	$3.57\times 10^{1}$	$1.33\times 10^{1}$	$1.83\times 10^{1}$	$4.95\times 10^{1}$	$6.09\times 10^{1}$	$3.30\times 10^{1}$	$1.49\times 10^{1}$
	Rank	5	8	3	2	4	7	6	9	1
F8	Ave	$9.97\times 10^{2}$	$9.81\times 10^{2}$	$9.12\times 10^{2}$	$8.84\times 10^{2}$	$9.85\times 10^{2}$	$9.69\times 10^{2}$	$9.64\times 10^{2}$	$1.14\times 10^{3}$	$8.63\times 10^{2}$
	Std	$5.11\times 10^{1}$	$2.80\times 10^{1}$	$3.31\times 10^{1}$	$1.80\times 10^{1}$	$4.50\times 10^{1}$	$2.76\times 10^{1}$	$2.77\times 10^{1}$	$1.87\times 10^{1}$	$1.40\times 10^{1}$
	Rank	8	6	3	2	7	5	4	9	1
F9	Ave	$6.37\times 10^{3}$	$8.04\times 10^{3}$	$2.85\times 10^{3}$	$1.64\times 10^{3}$	$9.09\times 10^{2}$	$6.70\times 10^{3}$	$6.78\times 10^{3}$	$1.16\times 10^{4}$	$1.28\times 10^{3}$
	Std	$2.97\times 10^{3}$	$1.02\times 10^{3}$	$1.15\times 10^{3}$	$5.25\times 10^{2}$	$3.29\times 10^{1}$	$1.44\times 10^{3}$	$1.14\times 10^{3}$	$1.22\times 10^{3}$	$3.19\times 10^{2}$
	Rank	5	8	4	3	1	6	7	9	2
F10	Ave	$5.71\times 10^{3}$	$6.20\times 10^{3}$	$4.76\times 10^{3}$	$4.12\times 10^{3}$	$8.49\times 10^{3}$	$5.83\times 10^{3}$	$5.55\times 10^{3}$	$9.06\times 10^{3}$	$3.50\times 10^{3}$
	Std	$6.30\times 10^{2}$	$6.35\times 10^{2}$	$6.30\times 10^{2}$	$5.62\times 10^{2}$	$4.18\times 10^{2}$	$7.81\times 10^{2}$	$7.51\times 10^{2}$	$5.05\times 10^{2}$	$4.01\times 10^{2}$
	Rank	5	7	3	2	8	6	4	9	1
F11	Ave	$1.49\times 10^{3}$	$1.29\times 10^{3}$	$1.26\times 10^{3}$	$1.21\times 10^{3}$	$1.19\times 10^{3}$	$2.53\times 10^{3}$	$2.30\times 10^{3}$	$7.12\times 10^{3}$	$1.17\times 10^{3}$
	Std	$1.49\times 10^{2}$	$4.92\times 10^{1}$	$5.41\times 10^{1}$	$3.56\times 10^{1}$	$2.92\times 10^{1}$	$6.05\times 10^{2}$	$7.30\times 10^{2}$	$1.50\times 10^{3}$	$3.15\times 10^{1}$
	Rank	6	5	4	3	2	8	7	9	1
F12	Ave	$5.23\times 10^{7}$	$2.30\times 10^{7}$	$8.19\times 10^{6}$	$2.10\times 10^{6}$	$2.86\times 10^{7}$	$9.26\times 10^{7}$	$1.16\times 10^{8}$	$1.37\times 10^{10}$	$8.57\times 10^{5}$
	Std	$3.01\times 10^{7}$	$2.06\times 10^{7}$	$8.05\times 10^{6}$	$9.40\times 10^{5}$	$3.95\times 10^{7}$	$7.51\times 10^{7}$	$1.04\times 10^{8}$	$2.72\times 10^{9}$	$7.56\times 10^{5}$
	Rank	6	4	3	2	5	7	8	9	1
F13	Ave	$1.78\times 10^{5}$	$8.59\times 10^{5}$	$9.77\times 10^{4}$	$7.92\times 10^{4}$	$3.05\times 10^{5}$	$2.23\times 10^{6}$	$3.06\times 10^{6}$	$9.00\times 10^{9}$	$4.21\times 10^{3}$
	Std	$1.31\times 10^{5}$	$8.09\times 10^{5}$	$5.62\times 10^{4}$	$3.48\times 10^{4}$	$6.90\times 10^{5}$	$2.85\times 10^{6}$	$8.55\times 10^{6}$	$4.22\times 10^{9}$	$5.00\times 10^{3}$
	Rank	4	6	3	2	5	7	8	9	1
F14	Ave	$5.09\times 10^{5}$	$4.81\times 10^{5}$	$3.53\times 10^{4}$	$2.29\times 10^{5}$	$1.85\times 10^{4}$	$9.94\times 10^{5}$	$8.60\times 10^{5}$	$1.64\times 10^{6}$	$1.17\times 10^{5}$
	Std	$4.85\times 10^{5}$	$5.50\times 10^{5}$	$3.01\times 10^{4}$	$2.09\times 10^{5}$	$2.32\times 10^{4}$	$8.32\times 10^{5}$	$9.72\times 10^{5}$	$1.35\times 10^{6}$	$1.25\times 10^{5}$
	Rank	6	5	2	4	1	8	7	9	3
F15	Ave	$6.05\times 10^{4}$	$8.06\times 10^{4}$	$7.01\times 10^{4}$	$8.09\times 10^{3}$	$4.85\times 10^{4}$	$1.26\times 10^{5}$	$1.10\times 10^{5}$	$2.35\times 10^{8}$	$3.43\times 10^{3}$
	Std	$2.66\times 10^{4}$	$5.12\times 10^{4}$	$5.77\times 10^{4}$	$4.61\times 10^{3}$	$7.72\times 10^{4}$	$8.17\times 10^{4}$	$6.29\times 10^{4}$	$1.88\times 10^{8}$	$2.34\times 10^{3}$
	Rank	4	6	5	2	3	8	7	9	1
F16	Ave	$3.00\times 10^{3}$	$3.44\times 10^{3}$	$2.60\times 10^{3}$	$2.40\times 10^{3}$	$3.37\times 10^{3}$	$3.27\times 10^{3}$	$3.32\times 10^{3}$	$5.35\times 10^{3}$	$2.36\times 10^{3}$
	Std	$3.42\times 10^{2}$	$3.89\times 10^{2}$	$3.63\times 10^{2}$	$2.49\times 10^{2}$	$3.30\times 10^{2}$	$4.15\times 10^{2}$	$4.23\times 10^{2}$	$6.96\times 10^{2}$	$2.07\times 10^{2}$
	Rank	4	8	3	2	7	5	6	9	1
F17	Ave	$2.34\times 10^{3}$	$2.65\times 10^{3}$	$2.12\times 10^{3}$	$2.08\times 10^{3}$	$2.35\times 10^{3}$	$2.35\times 10^{3}$	$2.46\times 10^{3}$	$3.56\times 10^{3}$	$2.00\times 10^{3}$
	Std	$2.39\times 10^{2}$	$2.92\times 10^{2}$	$1.70\times 10^{2}$	$1.71\times 10^{2}$	$1.95\times 10^{2}$	$1.92\times 10^{2}$	$2.52\times 10^{2}$	$3.52\times 10^{2}$	$1.43\times 10^{2}$
	Rank	4	8	3	2	5	6	7	9	1
F18	Ave	$2.32\times 10^{6}$	$2.57\times 10^{6}$	$4.25\times 10^{5}$	$5.51\times 10^{5}$	$1.24\times 10^{6}$	$4.42\times 10^{6}$	$4.34\times 10^{6}$	$2.88\times 10^{7}$	$3.80\times 10^{5}$
	Std	$2.31\times 10^{6}$	$3.34\times 10^{6}$	$2.87\times 10^{5}$	$5.74\times 10^{5}$	$1.04\times 10^{6}$	$4.33\times 10^{6}$	$3.72\times 10^{6}$	$2.70\times 10^{7}$	$2.59\times 10^{5}$
	Rank	5	6	2	3	4	8	7	9	1
F19	Ave	$1.52\times 10^{5}$	$1.16\times 10^{6}$	$3.05\times 10^{5}$	$1.21\times 10^{4}$	$2.43\times 10^{4}$	$2.50\times 10^{6}$	$1.99\times 10^{6}$	$3.74\times 10^{8}$	$4.23\times 10^{3}$
	Std	$2.59\times 10^{5}$	$8.48\times 10^{5}$	$2.62\times 10^{5}$	$7.95\times 10^{3}$	$3.29\times 10^{4}$	$2.61\times 10^{6}$	$2.17\times 10^{6}$	$2.66\times 10^{8}$	$2.88\times 10^{3}$
	Rank	4	6	5	2	3	8	7	9	1
F20	Ave	$2.66\times 10^{3}$	$2.76\times 10^{3}$	$2.44\times 10^{3}$	$2.41\times 10^{3}$	$2.61\times 10^{3}$	$2.60\times 10^{3}$	$2.62\times 10^{3}$	$3.07\times 10^{3}$	$2.34\times 10^{3}$
	Std	$2.64\times 10^{2}$	$2.02\times 10^{2}$	$1.47\times 10^{2}$	$1.31\times 10^{2}$	$1.61\times 10^{2}$	$1.41\times 10^{2}$	$1.73\times 10^{2}$	$2.16\times 10^{2}$	$1.52\times 10^{2}$
	Rank	7	8	3	2	5	4	6	9	1
F21	Ave	$2.49\times 10^{3}$	$2.57\times 10^{3}$	$2.41\times 10^{3}$	$2.40\times 10^{3}$	$2.48\times 10^{3}$	$2.48\times 10^{3}$	$2.50\times 10^{3}$	$2.73\times 10^{3}$	$2.36\times 10^{3}$
	Std	$3.87\times 10^{1}$	$4.93\times 10^{1}$	$2.94\times 10^{1}$	$2.17\times 10^{1}$	$1.85\times 10^{1}$	$2.79\times 10^{1}$	$4.96\times 10^{1}$	$3.05\times 10^{1}$	$1.03\times 10^{1}$
	Rank	6	8	3	2	5	4	7	9	1
F22	Ave	$6.44\times 10^{3}$	$6.72\times 10^{3}$	$5.06\times 10^{3}$	$3.43\times 10^{3}$	$9.42\times 10^{3}$	$2.76\times 10^{3}$	$2.57\times 10^{3}$	$9.11\times 10^{3}$	$3.17\times 10^{3}$
	Std	$2.14\times 10^{3}$	$2.04\times 10^{3}$	$1.62\times 10^{3}$	$1.60\times 10^{3}$	$1.52\times 10^{3}$	$9.12\times 10^{2}$	$1.68\times 10^{2}$	$9.22\times 10^{2}$	$1.28\times 10^{3}$
	Rank	6	7	5	4	9	2	1	8	3
F23	Ave	$2.88\times 10^{3}$	$3.26\times 10^{3}$	$2.78\times 10^{3}$	$2.78\times 10^{3}$	$2.83\times 10^{3}$	$2.96\times 10^{3}$	$2.96\times 10^{3}$	$3.52\times 10^{3}$	$2.70\times 10^{3}$
	Std	$4.60\times 10^{1}$	$1.48\times 10^{2}$	$3.79\times 10^{1}$	$2.55\times 10^{1}$	$3.93\times 10^{1}$	$5.97\times 10^{1}$	$6.45\times 10^{1}$	$1.50\times 10^{2}$	$1.45\times 10^{1}$
	Rank	5	8	2	3	4	6	7	9	1
F24	Ave	$3.01\times 10^{3}$	$3.49\times 10^{3}$	$2.95\times 10^{3}$	$3.00\times 10^{3}$	$3.04\times 10^{3}$	$3.11\times 10^{3}$	$3.09\times 10^{3}$	$3.80\times 10^{3}$	$2.91\times 10^{3}$
	Std	$4.71\times 10^{1}$	$1.72\times 10^{2}$	$4.50\times 10^{1}$	$3.68\times 10^{1}$	$2.24\times 10^{1}$	$6.35\times 10^{1}$	$6.71\times 10^{1}$	$3.39\times 10^{2}$	$2.15\times 10^{1}$
	Rank	4	8	2	3	5	7	6	9	1
F25	Ave	$2.99\times 10^{3}$	$2.96\times 10^{3}$	$2.89\times 10^{3}$	$2.90\times 10^{3}$	$2.88\times 10^{3}$	$3.00\times 10^{3}$	$2.99\times 10^{3}$	$4.65\times 10^{3}$	$2.89\times 10^{3}$
	Std	$5.76\times 10^{1}$	$2.97\times 10^{1}$	$9.01\times 10^{0}$	$1.74\times 10^{1}$	$6.24\times 10^{0}$	$4.17\times 10^{1}$	$3.36\times 10^{1}$	$2.95\times 10^{2}$	$2.58\times 10^{0}$
	Rank	6	5	3	4	1	8	7	9	2
F26	Ave	$5.59\times 10^{3}$	$7.81\times 10^{3}$	$4.37\times 10^{3}$	$4.34\times 10^{3}$	$5.27\times 10^{3}$	$5.71\times 10^{3}$	$6.09\times 10^{3}$	$1.06\times 10^{4}$	$3.73\times 10^{3}$
	Std	$8.91\times 10^{2}$	$1.29\times 10^{3}$	$9.65\times 10^{2}$	$1.02\times 10^{3}$	$4.18\times 10^{2}$	$1.50\times 10^{3}$	$1.54\times 10^{3}$	$5.59\times 10^{2}$	$7.16\times 10^{2}$
	Rank	5	8	3	2	4	6	7	9	1
F27	Ave	$3.28\times 10^{3}$	$3.46\times 10^{3}$	$3.23\times 10^{3}$	$3.24\times 10^{3}$	$3.20\times 10^{3}$	$3.37\times 10^{3}$	$3.36\times 10^{3}$	$4.16\times 10^{3}$	$3.22\times 10^{3}$
	Std	$3.44\times 10^{1}$	$9.56\times 10^{1}$	$1.79\times 10^{1}$	$1.25\times 10^{1}$	$1.13\times 10^{-4}$	$7.69\times 10^{1}$	$7.06\times 10^{1}$	$4.02\times 10^{2}$	$6.28\times 10^{0}$
	Rank	5	8	3	4	1	7	6	9	2
F28	Ave	$3.40\times 10^{3}$	$3.33\times 10^{3}$	$3.23\times 10^{3}$	$3.25\times 10^{3}$	$3.30\times 10^{3}$	$3.45\times 10^{3}$	$3.42\times 10^{3}$	$6.53\times 10^{3}$	$3.22\times 10^{3}$
	Std	$6.43\times 10^{1}$	$2.61\times 10^{1}$	$2.23\times 10^{1}$	$2.56\times 10^{1}$	$1.54\times 10^{0}$	$6.28\times 10^{1}$	$5.48\times 10^{1}$	$3.67\times 10^{2}$	$1.34\times 10^{1}$
	Rank	6	5	2	3	4	8	7	9	1
F29	Ave	$4.26\times 10^{3}$	$4.76\times 10^{3}$	$3.86\times 10^{3}$	$3.70\times 10^{3}$	$4.42\times 10^{3}$	$4.63\times 10^{3}$	$4.68\times 10^{3}$	$6.77\times 10^{3}$	$3.59\times 10^{3}$
	Std	$2.59\times 10^{2}$	$3.71\times 10^{2}$	$1.67\times 10^{2}$	$1.61\times 10^{2}$	$3.27\times 10^{2}$	$3.00\times 10^{2}$	$3.45\times 10^{2}$	$6.92\times 10^{2}$	$1.34\times 10^{2}$
	Rank	4	8	3	2	5	6	7	9	1
F30	Ave	$3.42\times 10^{6}$	$5.15\times 10^{6}$	$2.13\times 10^{6}$	$3.05\times 10^{4}$	$2.22\times 10^{4}$	$2.02\times 10^{7}$	$1.51\times 10^{7}$	$1.43\times 10^{9}$	$1.11\times 10^{4}$
	Std	$2.58\times 10^{6}$	$2.90\times 10^{6}$	$1.27\times 10^{6}$	$1.79\times 10^{4}$	$5.33\times 10^{4}$	$2.06\times 10^{7}$	$1.12\times 10^{7}$	$1.40\times 10^{9}$	$3.50\times 10^{3}$
	Rank	5	6	4	3	2	8	7	9	1

**Table 11 biomimetics-10-00620-t011:** Experimental results of MSAO and other algorithms on CEC2017 (50D).

		WOA	HHO	AOO	DOA	LSHADE	AO	LOBLAO	TEAO	MSAO
F1	Ave	$3.74\times 10^{9}$	$2.52\times 10^{8}$	$7.75\times 10^{4}$	$2.01\times 10^{8}$	$4.13\times 10^{7}$	$5.24\times 10^{9}$	$5.35\times 10^{9}$	$1.05\times 10^{11}$	$2.57\times 10^{5}$
	Std	$2.19\times 10^{9}$	$5.07\times 10^{7}$	$4.01\times 10^{4}$	$4.01\times 10^{7}$	$1.59\times 10^{8}$	$1.80\times 10^{9}$	$1.59\times 10^{9}$	$5.70\times 10^{9}$	$8.09\times 10^{4}$
	Rank	6	5	1	4	3	7	8	9	2
F3	Ave	$1.28\times 10^{5}$	$1.36\times 10^{5}$	$6.42\times 10^{4}$	$9.23\times 10^{4}$	$5.21\times 10^{5}$	$2.45\times 10^{5}$	$2.56\times 10^{5}$	$2.42\times 10^{5}$	$1.96\times 10^{5}$
	Std	$2.36\times 10^{4}$	$1.59\times 10^{4}$	$1.86\times 10^{4}$	$1.21\times 10^{4}$	$1.60\times 10^{5}$	$6.04\times 10^{4}$	$5.47\times 10^{4}$	$5.92\times 10^{4}$	$3.72\times 10^{4}$
	Rank	3	4	1	2	9	7	8	6	5
F4	Ave	$1.05\times 10^{3}$	$8.63\times 10^{2}$	$5.97\times 10^{2}$	$6.29\times 10^{2}$	$5.07\times 10^{2}$	$1.70\times 10^{3}$	$1.57\times 10^{3}$	$3.06\times 10^{4}$	$5.37\times 10^{2}$
	Std	$2.15\times 10^{2}$	$1.03\times 10^{2}$	$4.77\times 10^{1}$	$4.28\times 10^{1}$	$9.06\times 10^{1}$	$3.38\times 10^{2}$	$3.83\times 10^{2}$	$5.04\times 10^{3}$	$3.91\times 10^{1}$
	Rank	6	5	3	4	1	8	7	9	2
F5	Ave	$9.63\times 10^{2}$	$9.16\times 10^{2}$	$7.51\times 10^{2}$	$7.30\times 10^{2}$	$8.98\times 10^{2}$	$8.83\times 10^{2}$	$8.95\times 10^{2}$	$1.20\times 10^{3}$	$6.28\times 10^{2}$
	Std	$6.66\times 10^{1}$	$3.24\times 10^{1}$	$3.87\times 10^{1}$	$2.82\times 10^{1}$	$4.11\times 10^{1}$	$4.16\times 10^{1}$	$4.58\times 10^{1}$	$2.23\times 10^{1}$	$1.73\times 10^{1}$
	Rank	8	7	3	2	6	4	5	9	1
F6	Ave	$6.65\times 10^{2}$	$6.77\times 10^{2}$	$6.42\times 10^{2}$	$6.06\times 10^{2}$	$6.01\times 10^{2}$	$6.68\times 10^{2}$	$6.70\times 10^{2}$	$7.06\times 10^{2}$	$6.01\times 10^{2}$
	Std	$9.00\times 10^{0}$	$5.75\times 10^{0}$	$8.56\times 10^{0}$	$1.05\times 10^{0}$	$2.27\times 10^{0}$	$7.21\times 10^{0}$	$8.07\times 10^{0}$	$6.19\times 10^{0}$	$3.06\times 10^{-1}$
	Rank	5	8	4	3	2	6	7	9	1
F7	Ave	$1.40\times 10^{3}$	$1.85\times 10^{3}$	$1.02\times 10^{3}$	$1.08\times 10^{3}$	$1.17\times 10^{3}$	$1.58\times 10^{3}$	$1.58\times 10^{3}$	$1.98\times 10^{3}$	$9.09\times 10^{2}$
	Std	$9.54\times 10^{1}$	$8.55\times 10^{1}$	$5.10\times 10^{1}$	$3.68\times 10^{1}$	$3.30\times 10^{1}$	$8.74\times 10^{1}$	$8.65\times 10^{1}$	$5.72\times 10^{1}$	$3.13\times 10^{1}$
	Rank	5	8	2	3	4	6	7	9	1
F8	Ave	$1.23\times 10^{3}$	$1.20\times 10^{3}$	$1.04\times 10^{3}$	$1.05\times 10^{3}$	$1.20\times 10^{3}$	$1.21\times 10^{3}$	$1.21\times 10^{3}$	$1.52\times 10^{3}$	$9.32\times 10^{2}$
	Std	$8.24\times 10^{1}$	$2.69\times 10^{1}$	$4.06\times 10^{1}$	$2.49\times 10^{1}$	$4.16\times 10^{1}$	$4.44\times 10^{1}$	$4.72\times 10^{1}$	$2.96\times 10^{1}$	$1.84\times 10^{1}$
	Rank	8	5	2	3	4	6	7	9	1
F9	Ave	$1.83\times 10^{4}$	$2.89\times 10^{4}$	$9.99\times 10^{3}$	$8.10\times 10^{3}$	$1.48\times 10^{3}$	$2.39\times 10^{4}$	$2.48\times 10^{4}$	$4.01\times 10^{4}$	$4.42\times 10^{3}$
	Std	$4.77\times 10^{3}$	$2.74\times 10^{3}$	$2.81\times 10^{3}$	$2.31\times 10^{3}$	$7.41\times 10^{2}$	$4.32\times 10^{3}$	$4.05\times 10^{3}$	$3.45\times 10^{3}$	$1.73\times 10^{3}$
	Rank	5	8	4	3	1	6	7	9	2
F10	Ave	$1.00\times 10^{4}$	$9.62\times 10^{3}$	$7.51\times 10^{3}$	$7.58\times 10^{3}$	$1.51\times 10^{4}$	$9.47\times 10^{3}$	$1.00\times 10^{4}$	$1.55\times 10^{4}$	$5.57\times 10^{3}$
	Std	$1.10\times 10^{3}$	$1.12\times 10^{3}$	$1.09\times 10^{3}$	$6.04\times 10^{2}$	$9.66\times 10^{2}$	$8.37\times 10^{2}$	$1.19\times 10^{3}$	$6.83\times 10^{2}$	$5.71\times 10^{2}$
	Rank	7	5	2	3	8	4	6	9	1
F11	Ave	$3.91\times 10^{3}$	$1.79\times 10^{3}$	$1.42\times 10^{3}$	$1.49\times 10^{3}$	$3.12\times 10^{3}$	$3.45\times 10^{3}$	$3.26\times 10^{3}$	$2.27\times 10^{4}$	$1.48\times 10^{3}$
	Std	$1.68\times 10^{3}$	$1.50\times 10^{2}$	$6.86\times 10^{1}$	$1.39\times 10^{2}$	$1.89\times 10^{3}$	$8.54\times 10^{2}$	$6.29\times 10^{2}$	$2.18\times 10^{3}$	$6.80\times 10^{2}$
	Rank	8	4	1	3	5	7	6	9	2
F12	Ave	$4.46\times 10^{8}$	$2.75\times 10^{8}$	$4.95\times 10^{7}$	$1.97\times 10^{7}$	$4.13\times 10^{8}$	$1.17\times 10^{9}$	$1.07\times 10^{9}$	$7.24\times 10^{10}$	$9.80\times 10^{6}$
	Std	$2.75\times 10^{8}$	$1.63\times 10^{8}$	$2.87\times 10^{7}$	$9.26\times 10^{6}$	$5.01\times 10^{8}$	$6.54\times 10^{8}$	$5.15\times 10^{8}$	$1.38\times 10^{10}$	$4.19\times 10^{6}$
	Rank	6	4	3	2	5	8	7	9	1
F13	Ave	$4.65\times 10^{7}$	$5.33\times 10^{6}$	$1.31\times 10^{5}$	$5.25\times 10^{5}$	$1.54\times 10^{6}$	$7.72\times 10^{7}$	$6.44\times 10^{7}$	$4.00\times 10^{10}$	$7.33\times 10^{3}$
	Std	$7.57\times 10^{7}$	$3.36\times 10^{6}$	$7.18\times 10^{4}$	$2.41\times 10^{5}$	$4.18\times 10^{6}$	$1.22\times 10^{8}$	$5.44\times 10^{7}$	$1.30\times 10^{10}$	$7.84\times 10^{3}$
	Rank	6	5	2	3	4	8	7	9	1
F14	Ave	$1.21\times 10^{6}$	$3.42\times 10^{6}$	$1.80\times 10^{5}$	$1.24\times 10^{6}$	$1.15\times 10^{6}$	$4.43\times 10^{6}$	$3.84\times 10^{6}$	$5.41\times 10^{7}$	$7.57\times 10^{5}$
	Std	$7.97\times 10^{5}$	$3.63\times 10^{6}$	$1.39\times 10^{5}$	$1.01\times 10^{6}$	$9.89\times 10^{5}$	$3.73\times 10^{6}$	$3.71\times 10^{6}$	$4.12\times 10^{7}$	$5.57\times 10^{5}$
	Rank	4	6	1	5	3	8	7	9	2
F15	Ave	$5.32\times 10^{6}$	$8.72\times 10^{5}$	$6.40\times 10^{4}$	$6.47\times 10^{4}$	$4.64\times 10^{5}$	$1.36\times 10^{6}$	$1.02\times 10^{6}$	$4.65\times 10^{9}$	$4.72\times 10^{3}$
	Std	$1.15\times 10^{7}$	$4.80\times 10^{5}$	$4.66\times 10^{4}$	$3.82\times 10^{4}$	$1.05\times 10^{6}$	$1.87\times 10^{6}$	$6.06\times 10^{5}$	$1.90\times 10^{9}$	$3.62\times 10^{3}$
	Rank	8	5	2	3	4	7	6	9	1
F16	Ave	$4.22\times 10^{3}$	$4.43\times 10^{3}$	$3.28\times 10^{3}$	$3.25\times 10^{3}$	$5.70\times 10^{3}$	$4.57\times 10^{3}$	$4.76\times 10^{3}$	$8.87\times 10^{3}$	$3.00\times 10^{3}$
	Std	$4.65\times 10^{2}$	$6.45\times 10^{2}$	$4.09\times 10^{2}$	$3.63\times 10^{2}$	$3.46\times 10^{2}$	$4.89\times 10^{2}$	$6.01\times 10^{2}$	$1.08\times 10^{3}$	$2.87\times 10^{2}$
	Rank	4	5	3	2	8	6	7	9	1
F17	Ave	$3.68\times 10^{3}$	$3.74\times 10^{3}$	$3.06\times 10^{3}$	$2.84\times 10^{3}$	$4.16\times 10^{3}$	$3.73\times 10^{3}$	$3.84\times 10^{3}$	$1.07\times 10^{4}$	$2.67\times 10^{3}$
	Std	$3.51\times 10^{2}$	$4.34\times 10^{2}$	$3.26\times 10^{2}$	$3.02\times 10^{2}$	$2.19\times 10^{2}$	$3.30\times 10^{2}$	$3.85\times 10^{2}$	$6.23\times 10^{3}$	$1.96\times 10^{2}$
	Rank	4	6	3	2	8	5	7	9	1
F18	Ave	$6.37\times 10^{6}$	$7.68\times 10^{6}$	$2.12\times 10^{6}$	$3.03\times 10^{6}$	$2.13\times 10^{7}$	$1.66\times 10^{7}$	$1.43\times 10^{7}$	$1.24\times 10^{8}$	$1.79\times 10^{6}$
	Std	$6.48\times 10^{6}$	$6.72\times 10^{6}$	$1.47\times 10^{6}$	$1.84\times 10^{6}$	$1.49\times 10^{7}$	$6.23\times 10^{6}$	$8.36\times 10^{6}$	$5.52\times 10^{7}$	$2.14\times 10^{6}$
	Rank	4	5	2	3	8	7	6	9	1
F19	Ave	$1.44\times 10^{6}$	$1.36\times 10^{6}$	$7.12\times 10^{5}$	$3.79\times 10^{4}$	$4.74\times 10^{4}$	$3.38\times 10^{6}$	$3.48\times 10^{6}$	$3.59\times 10^{9}$	$7.43\times 10^{3}$
	Std	$1.98\times 10^{6}$	$1.30\times 10^{6}$	$6.04\times 10^{5}$	$1.39\times 10^{4}$	$1.03\times 10^{5}$	$3.88\times 10^{6}$	$3.27\times 10^{6}$	$1.21\times 10^{9}$	$6.88\times 10^{3}$
	Rank	6	5	4	2	3	7	8	9	1
F20	Ave	$3.52\times 10^{3}$	$3.42\times 10^{3}$	$3.11\times 10^{3}$	$2.96\times 10^{3}$	$4.17\times 10^{3}$	$3.43\times 10^{3}$	$3.36\times 10^{3}$	$4.27\times 10^{3}$	$2.81\times 10^{3}$
	Std	$3.32\times 10^{2}$	$2.78\times 10^{2}$	$3.67\times 10^{2}$	$2.38\times 10^{2}$	$2.52\times 10^{2}$	$2.65\times 10^{2}$	$3.28\times 10^{2}$	$2.61\times 10^{2}$	$2.19\times 10^{2}$
	Rank	7	5	3	2	8	6	4	9	1
F21	Ave	$2.75\times 10^{3}$	$2.91\times 10^{3}$	$2.52\times 10^{3}$	$2.53\times 10^{3}$	$2.71\times 10^{3}$	$2.77\times 10^{3}$	$2.75\times 10^{3}$	$3.21\times 10^{3}$	$2.43\times 10^{3}$
	Std	$8.03\times 10^{1}$	$5.89\times 10^{1}$	$4.36\times 10^{1}$	$2.79\times 10^{1}$	$4.64\times 10^{1}$	$9.49\times 10^{1}$	$5.69\times 10^{1}$	$7.63\times 10^{1}$	$1.75\times 10^{1}$
	Rank	5	8	2	3	4	7	6	9	1
F22	Ave	$1.13\times 10^{4}$	$1.16\times 10^{4}$	$9.22\times 10^{3}$	$9.26\times 10^{3}$	$1.64\times 10^{4}$	$1.14\times 10^{4}$	$1.16\times 10^{4}$	$1.75\times 10^{4}$	$7.16\times 10^{3}$
	Std	$6.72\times 10^{2}$	$1.01\times 10^{3}$	$1.73\times 10^{3}$	$1.54\times 10^{3}$	$8.80\times 10^{2}$	$1.58\times 10^{3}$	$1.71\times 10^{3}$	$5.24\times 10^{2}$	$1.11\times 10^{3}$
	Rank	4	6	2	3	8	5	7	9	1
F23	Ave	$3.29\times 10^{3}$	$3.89\times 10^{3}$	$3.02\times 10^{3}$	$3.03\times 10^{3}$	$3.10\times 10^{3}$	$3.57\times 10^{3}$	$3.53\times 10^{3}$	$4.56\times 10^{3}$	$2.87\times 10^{3}$
	Std	$1.52\times 10^{2}$	$1.72\times 10^{2}$	$6.30\times 10^{1}$	$3.56\times 10^{1}$	$4.63\times 10^{1}$	$1.31\times 10^{2}$	$1.11\times 10^{2}$	$2.37\times 10^{2}$	$3.29\times 10^{1}$
	Rank	5	8	2	3	4	7	6	9	1
F24	Ave	$3.40\times 10^{3}$	$4.32\times 10^{3}$	$3.18\times 10^{3}$	$3.28\times 10^{3}$	$3.33\times 10^{3}$	$3.57\times 10^{3}$	$3.55\times 10^{3}$	$4.79\times 10^{3}$	$3.09\times 10^{3}$
	Std	$1.02\times 10^{2}$	$2.84\times 10^{2}$	$7.98\times 10^{1}$	$8.21\times 10^{1}$	$4.67\times 10^{1}$	$1.19\times 10^{2}$	$1.23\times 10^{2}$	$3.97\times 10^{2}$	$3.97\times 10^{1}$
	Rank	5	8	2	3	4	7	6	9	1
F25	Ave	$3.57\times 10^{3}$	$3.28\times 10^{3}$	$3.07\times 10^{3}$	$3.15\times 10^{3}$	$2.96\times 10^{3}$	$3.71\times 10^{3}$	$3.69\times 10^{3}$	$1.37\times 10^{4}$	$3.05\times 10^{3}$
	Std	$2.87\times 10^{2}$	$6.47\times 10^{1}$	$2.93\times 10^{1}$	$3.44\times 10^{1}$	$6.17\times 10^{1}$	$2.17\times 10^{2}$	$1.54\times 10^{2}$	$1.08\times 10^{3}$	$2.77\times 10^{1}$
	Rank	6	5	3	4	1	8	7	9	2
F26	Ave	$9.06\times 10^{3}$	$1.09\times 10^{4}$	$5.72\times 10^{3}$	$6.95\times 10^{3}$	$7.46\times 10^{3}$	$9.93\times 10^{3}$	$9.01\times 10^{3}$	$1.63\times 10^{4}$	$5.24\times 10^{3}$
	Std	$1.12\times 10^{3}$	$1.90\times 10^{3}$	$2.09\times 10^{3}$	$3.51\times 10^{2}$	$6.09\times 10^{2}$	$1.90\times 10^{3}$	$2.26\times 10^{3}$	$6.08\times 10^{2}$	$2.92\times 10^{2}$
	Rank	6	8	2	3	4	7	5	9	1
F27	Ave	$3.79\times 10^{3}$	$4.52\times 10^{3}$	$3.56\times 10^{3}$	$3.50\times 10^{3}$	$3.20\times 10^{3}$	$4.28\times 10^{3}$	$4.27\times 10^{3}$	$6.14\times 10^{3}$	$3.34\times 10^{3}$
	Std	$1.48\times 10^{2}$	$3.70\times 10^{2}$	$1.04\times 10^{2}$	$5.97\times 10^{1}$	$8.66\times 10^{-5}$	$2.15\times 10^{2}$	$2.49\times 10^{2}$	$7.42\times 10^{2}$	$3.49\times 10^{1}$
	Rank	5	8	4	3	1	7	6	9	2
F28	Ave	$4.17\times 10^{3}$	$3.80\times 10^{3}$	$3.33\times 10^{3}$	$3.44\times 10^{3}$	$3.30\times 10^{3}$	$4.82\times 10^{3}$	$4.84\times 10^{3}$	$1.21\times 10^{4}$	$3.31\times 10^{3}$
	Std	$4.27\times 10^{2}$	$1.34\times 10^{2}$	$3.82\times 10^{1}$	$4.79\times 10^{1}$	$8.55\times 10^{-5}$	$3.47\times 10^{2}$	$3.59\times 10^{2}$	$6.22\times 10^{2}$	$2.71\times 10^{1}$
	Rank	6	5	3	4	1	7	8	9	2
F29	Ave	$6.06\times 10^{3}$	$6.23\times 10^{3}$	$4.75\times 10^{3}$	$4.15\times 10^{3}$	$5.88\times 10^{3}$	$7.05\times 10^{3}$	$6.71\times 10^{3}$	$3.15\times 10^{4}$	$3.89\times 10^{3}$
	Std	$6.78\times 10^{2}$	$5.63\times 10^{2}$	$2.91\times 10^{2}$	$2.85\times 10^{2}$	$6.56\times 10^{2}$	$1.00\times 10^{3}$	$7.60\times 10^{2}$	$1.37\times 10^{4}$	$1.89\times 10^{2}$
	Rank	5	6	3	2	4	8	7	9	1
F30	Ave	$5.66\times 10^{7}$	$6.85\times 10^{7}$	$3.30\times 10^{7}$	$1.79\times 10^{6}$	$9.53\times 10^{6}$	$1.45\times 10^{8}$	$1.50\times 10^{8}$	$6.75\times 10^{9}$	$9.67\times 10^{5}$
	Std	$2.18\times 10^{7}$	$1.95\times 10^{7}$	$8.03\times 10^{6}$	$2.65\times 10^{5}$	$1.70\times 10^{7}$	$5.91\times 10^{7}$	$4.96\times 10^{7}$	$1.81\times 10^{9}$	$1.40\times 10^{5}$
	Rank	5	6	4	2	3	7	8	9	1

**Table 12 biomimetics-10-00620-t012:** Experimental results of MSAO and other algorithms on CEC2017 (100D).

		WOA	HHO	AOO	DOA	LSHADE	AO	LOBLAO	TEAO	MSAO
F1	Ave	$4.89\times 10^{9}$	$2.73\times 10^{8}$	$8.19\times 10^{4}$	$2.13\times 10^{8}$	$2.16\times 10^{7}$	$5.40\times 10^{9}$	$5.58\times 10^{9}$	$1.06\times 10^{11}$	$2.43\times 10^{5}$
	Std	$2.74\times 10^{9}$	$8.54\times 10^{7}$	$2.19\times 10^{4}$	$4.95\times 10^{7}$	$6.72\times 10^{7}$	$1.40\times 10^{9}$	$1.46\times 10^{9}$	$5.48\times 10^{9}$	$6.58\times 10^{4}$
	Rank	6	5	1	4	3	7	8	9	2
F3	Ave	$1.28\times 10^{5}$	$1.33\times 10^{5}$	$6.68\times 10^{4}$	$9.29\times 10^{4}$	$5.78\times 10^{5}$	$2.57\times 10^{5}$	$2.55\times 10^{5}$	$2.64\times 10^{5}$	$1.97\times 10^{5}$
	Std	$2.44\times 10^{4}$	$1.32\times 10^{4}$	$1.54\times 10^{4}$	$1.90\times 10^{4}$	$1.64\times 10^{5}$	$4.68\times 10^{4}$	$6.29\times 10^{4}$	$7.36\times 10^{4}$	$4.58\times 10^{4}$
	Rank	3	4	1	2	9	7	6	8	5
F4	Ave	$1.16\times 10^{3}$	$8.19\times 10^{2}$	$5.91\times 10^{2}$	$6.27\times 10^{2}$	$4.85\times 10^{2}$	$1.62\times 10^{3}$	$1.56\times 10^{3}$	$3.07\times 10^{4}$	$5.38\times 10^{2}$
	Std	$3.95\times 10^{2}$	$9.62\times 10^{1}$	$5.44\times 10^{1}$	$5.02\times 10^{1}$	$4.34\times 10^{1}$	$3.52\times 10^{2}$	$3.45\times 10^{2}$	$4.45\times 10^{3}$	$3.13\times 10^{1}$
	Rank	6	5	3	4	1	8	7	9	2
F5	Ave	$9.29\times 10^{2}$	$9.06\times 10^{2}$	$7.28\times 10^{2}$	$7.32\times 10^{2}$	$8.82\times 10^{2}$	$8.79\times 10^{2}$	$8.92\times 10^{2}$	$1.20\times 10^{3}$	$6.33\times 10^{2}$
	Std	$8.75\times 10^{1}$	$3.15\times 10^{1}$	$4.86\times 10^{1}$	$2.81\times 10^{1}$	$4.82\times 10^{1}$	$4.26\times 10^{1}$	$3.87\times 10^{1}$	$2.86\times 10^{1}$	$2.17\times 10^{1}$
	Rank	8	7	2	3	5	4	6	9	1
F6	Ave	$6.66\times 10^{2}$	$6.76\times 10^{2}$	$6.42\times 10^{2}$	$6.06\times 10^{2}$	$6.01\times 10^{2}$	$6.69\times 10^{2}$	$6.69\times 10^{2}$	$7.07\times 10^{2}$	$6.01\times 10^{2}$
	Std	$1.01\times 10^{1}$	$5.50\times 10^{0}$	$9.45\times 10^{0}$	$1.11\times 10^{0}$	$9.78\times 10^{-1}$	$7.19\times 10^{0}$	$5.86\times 10^{0}$	$4.99\times 10^{0}$	$2.00\times 10^{-1}$
	Rank	5	8	4	3	2	7	6	9	1
F7	Ave	$1.38\times 10^{3}$	$1.86\times 10^{3}$	$1.03\times 10^{3}$	$1.08\times 10^{3}$	$1.17\times 10^{3}$	$1.58\times 10^{3}$	$1.61\times 10^{3}$	$1.98\times 10^{3}$	$9.16\times 10^{2}$
	Std	$1.00\times 10^{2}$	$7.35\times 10^{1}$	$6.82\times 10^{1}$	$2.99\times 10^{1}$	$4.18\times 10^{1}$	$1.09\times 10^{2}$	$9.64\times 10^{1}$	$5.26\times 10^{1}$	$2.99\times 10^{1}$
	Rank	5	8	2	3	4	6	7	9	1
F8	Ave	$1.22\times 10^{3}$	$1.22\times 10^{3}$	$1.06\times 10^{3}$	$1.04\times 10^{3}$	$1.20\times 10^{3}$	$1.23\times 10^{3}$	$1.22\times 10^{3}$	$1.53\times 10^{3}$	$9.38\times 10^{2}$
	Std	$5.90\times 10^{1}$	$3.58\times 10^{1}$	$5.48\times 10^{1}$	$2.68\times 10^{1}$	$5.01\times 10^{1}$	$4.89\times 10^{1}$	$4.28\times 10^{1}$	$3.03\times 10^{1}$	$2.35\times 10^{1}$
	Rank	7	5	3	2	4	8	6	9	1
F9	Ave	$2.12\times 10^{4}$	$2.88\times 10^{4}$	$1.00\times 10^{4}$	$7.54\times 10^{3}$	$1.72\times 10^{3}$	$2.47\times 10^{4}$	$2.55\times 10^{4}$	$3.96\times 10^{4}$	$4.53\times 10^{3}$
	Std	$5.91\times 10^{3}$	$2.66\times 10^{3}$	$3.44\times 10^{3}$	$2.21\times 10^{3}$	$1.56\times 10^{3}$	$4.10\times 10^{3}$	$4.16\times 10^{3}$	$3.04\times 10^{3}$	$1.46\times 10^{3}$
	Rank	5	8	4	3	1	6	7	9	2
F10	Ave	$1.00\times 10^{4}$	$9.68\times 10^{3}$	$7.49\times 10^{3}$	$7.93\times 10^{3}$	$1.52\times 10^{4}$	$1.00\times 10^{4}$	$9.92\times 10^{3}$	$1.55\times 10^{4}$	$5.57\times 10^{3}$
	Std	$9.79\times 10^{2}$	$1.02\times 10^{3}$	$9.68\times 10^{2}$	$6.83\times 10^{2}$	$6.81\times 10^{2}$	$1.08\times 10^{3}$	$8.08\times 10^{2}$	$7.18\times 10^{2}$	$5.83\times 10^{2}$
	Rank	7	4	2	3	8	6	5	9	1
F11	Ave	$2.94\times 10^{3}$	$1.77\times 10^{3}$	$1.42\times 10^{3}$	$1.62\times 10^{3}$	$2.72\times 10^{3}$	$3.33\times 10^{3}$	$3.28\times 10^{3}$	$2.20\times 10^{4}$	$1.33\times 10^{3}$
	Std	$1.27\times 10^{3}$	$1.86\times 10^{2}$	$7.09\times 10^{1}$	$3.79\times 10^{2}$	$1.52\times 10^{3}$	$4.80\times 10^{2}$	$5.75\times 10^{2}$	$1.88\times 10^{3}$	$1.43\times 10^{2}$
	Rank	6	4	2	3	5	8	7	9	1
F12	Ave	$4.84\times 10^{8}$	$2.27\times 10^{8}$	$4.94\times 10^{7}$	$2.01\times 10^{7}$	$2.05\times 10^{8}$	$1.04\times 10^{9}$	$1.19\times 10^{9}$	$7.74\times 10^{10}$	$8.52\times 10^{6}$
	Std	$5.02\times 10^{8}$	$1.57\times 10^{8}$	$2.34\times 10^{7}$	$6.67\times 10^{6}$	$2.48\times 10^{8}$	$5.15\times 10^{8}$	$6.32\times 10^{8}$	$1.16\times 10^{10}$	$3.80\times 10^{6}$
	Rank	6	5	3	2	4	7	8	9	1
F13	Ave	$5.68\times 10^{7}$	$5.64\times 10^{6}$	$1.52\times 10^{5}$	$6.52\times 10^{5}$	$4.13\times 10^{5}$	$6.92\times 10^{7}$	$9.10\times 10^{7}$	$3.94\times 10^{10}$	$5.60\times 10^{3}$
	Std	$8.71\times 10^{7}$	$5.60\times 10^{6}$	$1.11\times 10^{5}$	$3.36\times 10^{5}$	$9.09\times 10^{5}$	$7.11\times 10^{7}$	$1.60\times 10^{8}$	$1.25\times 10^{10}$	$3.73\times 10^{3}$
	Rank	6	5	2	4	3	7	8	9	1
F14	Ave	$1.15\times 10^{6}$	$2.77\times 10^{6}$	$2.16\times 10^{5}$	$1.33\times 10^{6}$	$1.13\times 10^{6}$	$5.10\times 10^{6}$	$5.47\times 10^{6}$	$4.52\times 10^{7}$	$7.47\times 10^{5}$
	Std	$1.08\times 10^{6}$	$2.19\times 10^{6}$	$1.96\times 10^{5}$	$9.42\times 10^{5}$	$1.38\times 10^{6}$	$5.28\times 10^{6}$	$4.15\times 10^{6}$	$3.18\times 10^{7}$	$6.29\times 10^{5}$
	Rank	4	6	1	5	3	7	8	9	2
F15	Ave	$4.13\times 10^{6}$	$1.00\times 10^{6}$	$4.60\times 10^{4}$	$9.06\times 10^{4}$	$7.23\times 10^{5}$	$1.35\times 10^{6}$	$1.33\times 10^{6}$	$5.22\times 10^{9}$	$7.26\times 10^{3}$
	Std	$9.75\times 10^{6}$	$4.43\times 10^{5}$	$2.55\times 10^{4}$	$4.91\times 10^{4}$	$1.83\times 10^{6}$	$1.10\times 10^{6}$	$1.05\times 10^{6}$	$2.66\times 10^{9}$	$5.61\times 10^{3}$
	Rank	8	5	2	3	4	7	6	9	1
F16	Ave	$4.40\times 10^{3}$	$4.52\times 10^{3}$	$3.29\times 10^{3}$	$3.25\times 10^{3}$	$5.58\times 10^{3}$	$4.44\times 10^{3}$	$4.66\times 10^{3}$	$8.45\times 10^{3}$	$3.09\times 10^{3}$
	Std	$5.76\times 10^{2}$	$7.07\times 10^{2}$	$3.73\times 10^{2}$	$3.88\times 10^{2}$	$3.23\times 10^{2}$	$4.49\times 10^{2}$	$4.67\times 10^{2}$	$9.51\times 10^{2}$	$2.84\times 10^{2}$
	Rank	4	6	3	2	8	5	7	9	1
F17	Ave	$3.78\times 10^{3}$	$3.67\times 10^{3}$	$2.98\times 10^{3}$	$2.87\times 10^{3}$	$4.17\times 10^{3}$	$3.59\times 10^{3}$	$3.75\times 10^{3}$	$8.82\times 10^{3}$	$2.67\times 10^{3}$
	Std	$4.25\times 10^{2}$	$3.31\times 10^{2}$	$2.87\times 10^{2}$	$2.64\times 10^{2}$	$2.67\times 10^{2}$	$4.78\times 10^{2}$	$4.52\times 10^{2}$	$3.58\times 10^{3}$	$1.63\times 10^{2}$
	Rank	7	5	3	2	8	4	6	9	1
F18	Ave	$5.53\times 10^{6}$	$5.19\times 10^{6}$	$1.94\times 10^{6}$	$3.65\times 10^{6}$	$2.50\times 10^{7}$	$1.38\times 10^{7}$	$1.42\times 10^{7}$	$1.25\times 10^{8}$	$1.36\times 10^{6}$
	Std	$5.56\times 10^{6}$	$4.54\times 10^{6}$	$1.32\times 10^{6}$	$2.83\times 10^{6}$	$1.44\times 10^{7}$	$9.92\times 10^{6}$	$8.90\times 10^{6}$	$5.86\times 10^{7}$	$1.06\times 10^{6}$
	Rank	5	4	2	3	8	6	7	9	1
F19	Ave	$9.47\times 10^{5}$	$1.37\times 10^{6}$	$7.59\times 10^{5}$	$3.74\times 10^{4}$	$5.49\times 10^{4}$	$2.48\times 10^{6}$	$2.26\times 10^{6}$	$3.67\times 10^{9}$	$8.87\times 10^{3}$
	Std	$8.30\times 10^{5}$	$9.68\times 10^{5}$	$5.23\times 10^{5}$	$1.39\times 10^{4}$	$1.20\times 10^{5}$	$3.10\times 10^{6}$	$1.97\times 10^{6}$	$1.38\times 10^{9}$	$7.19\times 10^{3}$
	Rank	5	6	4	2	3	8	7	9	1
F20	Ave	$3.49\times 10^{3}$	$3.57\times 10^{3}$	$3.17\times 10^{3}$	$2.96\times 10^{3}$	$4.12\times 10^{3}$	$3.44\times 10^{3}$	$3.29\times 10^{3}$	$4.23\times 10^{3}$	$2.77\times 10^{3}$
	Std	$2.80\times 10^{2}$	$3.71\times 10^{2}$	$3.02\times 10^{2}$	$2.74\times 10^{2}$	$2.23\times 10^{2}$	$3.23\times 10^{2}$	$2.71\times 10^{2}$	$2.35\times 10^{2}$	$1.92\times 10^{2}$
	Rank	6	7	3	2	8	5	4	9	1
F21	Ave	$2.74\times 10^{3}$	$2.88\times 10^{3}$	$2.54\times 10^{3}$	$2.53\times 10^{3}$	$2.71\times 10^{3}$	$2.77\times 10^{3}$	$2.76\times 10^{3}$	$3.22\times 10^{3}$	$2.44\times 10^{3}$
	Std	$7.99\times 10^{1}$	$9.29\times 10^{1}$	$5.48\times 10^{1}$	$3.00\times 10^{1}$	$6.01\times 10^{1}$	$6.31\times 10^{1}$	$7.12\times 10^{1}$	$5.66\times 10^{1}$	$2.24\times 10^{1}$
	Rank	5	8	3	2	4	7	6	9	1
F22	Ave	$1.13\times 10^{4}$	$1.21\times 10^{4}$	$9.04\times 10^{3}$	$9.57\times 10^{3}$	$1.68\times 10^{4}$	$1.16\times 10^{4}$	$1.17\times 10^{4}$	$1.73\times 10^{4}$	$7.27\times 10^{3}$
	Std	$1.21\times 10^{3}$	$1.19\times 10^{3}$	$1.51\times 10^{3}$	$6.99\times 10^{2}$	$5.12\times 10^{2}$	$2.03\times 10^{3}$	$2.32\times 10^{3}$	$6.99\times 10^{2}$	$6.62\times 10^{2}$
	Rank	4	7	2	3	8	5	6	9	1
F23	Ave	$3.22\times 10^{3}$	$3.95\times 10^{3}$	$3.02\times 10^{3}$	$3.03\times 10^{3}$	$3.13\times 10^{3}$	$3.54\times 10^{3}$	$3.53\times 10^{3}$	$4.52\times 10^{3}$	$2.87\times 10^{3}$
	Std	$9.09\times 10^{1}$	$2.03\times 10^{2}$	$6.19\times 10^{1}$	$6.49\times 10^{1}$	$4.55\times 10^{1}$	$1.08\times 10^{2}$	$1.56\times 10^{2}$	$2.23\times 10^{2}$	$2.30\times 10^{1}$
	Rank	5	8	2	3	4	7	6	9	1
F24	Ave	$3.40\times 10^{3}$	$4.38\times 10^{3}$	$3.18\times 10^{3}$	$3.28\times 10^{3}$	$3.35\times 10^{3}$	$3.58\times 10^{3}$	$3.60\times 10^{3}$	$4.80\times 10^{3}$	$3.08\times 10^{3}$
	Std	$8.89\times 10^{1}$	$2.86\times 10^{2}$	$6.57\times 10^{1}$	$6.28\times 10^{1}$	$4.43\times 10^{1}$	$1.22\times 10^{2}$	$1.11\times 10^{2}$	$4.73\times 10^{2}$	$3.41\times 10^{1}$
	Rank	5	8	2	3	4	6	7	9	1
F25	Ave	$3.50\times 10^{3}$	$3.27\times 10^{3}$	$3.07\times 10^{3}$	$3.15\times 10^{3}$	$2.96\times 10^{3}$	$3.70\times 10^{3}$	$3.71\times 10^{3}$	$1.40\times 10^{4}$	$3.05\times 10^{3}$
	Std	$1.76\times 10^{2}$	$6.68\times 10^{1}$	$3.07\times 10^{1}$	$3.39\times 10^{1}$	$4.85\times 10^{1}$	$1.69\times 10^{2}$	$1.73\times 10^{2}$	$8.76\times 10^{2}$	$2.09\times 10^{1}$
	Rank	6	5	3	4	1	7	8	9	2
F26	Ave	$8.91\times 10^{3}$	$1.01\times 10^{4}$	$6.33\times 10^{3}$	$6.91\times 10^{3}$	$7.63\times 10^{3}$	$8.57\times 10^{3}$	$9.06\times 10^{3}$	$1.63\times 10^{4}$	$5.09\times 10^{3}$
	Std	$1.15\times 10^{3}$	$2.17\times 10^{3}$	$1.72\times 10^{3}$	$3.50\times 10^{2}$	$6.31\times 10^{2}$	$2.40\times 10^{3}$	$2.40\times 10^{3}$	$7.72\times 10^{2}$	$4.45\times 10^{2}$
	Rank	6	8	2	3	4	5	7	9	1
F27	Ave	$3.83\times 10^{3}$	$4.62\times 10^{3}$	$3.52\times 10^{3}$	$3.50\times 10^{3}$	$3.20\times 10^{3}$	$4.22\times 10^{3}$	$4.31\times 10^{3}$	$6.49\times 10^{3}$	$3.34\times 10^{3}$
	Std	$1.78\times 10^{2}$	$4.02\times 10^{2}$	$9.37\times 10^{1}$	$7.59\times 10^{1}$	$1.17\times 10^{-4}$	$2.41\times 10^{2}$	$2.32\times 10^{2}$	$7.03\times 10^{2}$	$3.57\times 10^{1}$
	Rank	5	8	4	3	1	6	7	9	2
F28	Ave	$4.10\times 10^{3}$	$3.84\times 10^{3}$	$3.34\times 10^{3}$	$3.43\times 10^{3}$	$3.30\times 10^{3}$	$4.71\times 10^{3}$	$4.72\times 10^{3}$	$1.21\times 10^{4}$	$3.31\times 10^{3}$
	Std	$3.25\times 10^{2}$	$1.60\times 10^{2}$	$3.70\times 10^{1}$	$4.62\times 10^{1}$	$1.06\times 10^{-4}$	$3.70\times 10^{2}$	$3.39\times 10^{2}$	$8.51\times 10^{2}$	$2.24\times 10^{1}$
	Rank	6	5	3	4	1	7	8	9	2
F29	Ave	$5.92\times 10^{3}$	$6.39\times 10^{3}$	$4.77\times 10^{3}$	$4.04\times 10^{3}$	$5.84\times 10^{3}$	$7.22\times 10^{3}$	$7.17\times 10^{3}$	$4.06\times 10^{4}$	$3.85\times 10^{3}$
	Std	$6.12\times 10^{2}$	$6.23\times 10^{2}$	$3.61\times 10^{2}$	$2.51\times 10^{2}$	$6.23\times 10^{2}$	$8.18\times 10^{2}$	$9.37\times 10^{2}$	$3.35\times 10^{4}$	$2.01\times 10^{2}$
	Rank	5	6	3	2	4	8	7	9	1
F30	Ave	$8.01\times 10^{7}$	$7.29\times 10^{7}$	$3.55\times 10^{7}$	$1.84\times 10^{6}$	$1.19\times 10^{7}$	$1.48\times 10^{8}$	$1.42\times 10^{8}$	$5.42\times 10^{9}$	$9.29\times 10^{5}$
	Std	$3.85\times 10^{7}$	$2.17\times 10^{7}$	$1.07\times 10^{7}$	$3.36\times 10^{5}$	$1.79\times 10^{7}$	$5.75\times 10^{7}$	$4.41\times 10^{7}$	$1.58\times 10^{9}$	$1.13\times 10^{5}$
	Rank	6	5	4	2	3	8	7	9	1

**Table 13 biomimetics-10-00620-t013:** Wilcoxon rank sum test of MSOA and other algorithms on CEC2017 (10D).

MSAO vs.	WOA	HHO	AOO	DOA	LSHADE	AO	LOBLAO	TEAO
F1	$4.08\times 10^{-11}$	$3.02\times 10^{-11}$	$3.37\times 10^{-4}$	$3.02\times 10^{-11}$	$4.80\times 10^{-7}$	$3.02\times 10^{-11}$	$3.02\times 10^{-11}$	$3.02\times 10^{-11}$
F2	$2.15\times 10^{-10}$	$5.49\times 10^{-11}$	$1.87\times 10^{-7}$	$3.20\times 10^{-9}$	$1.27\times 10^{-2}$	$3.02\times 10^{-11}$	$3.02\times 10^{-11}$	$3.02\times 10^{-11}$
F3	$3.02\times 10^{-11}$	$3.02\times 10^{-11}$	$1.56\times 10^{-8}$	$3.02\times 10^{-11}$	$5.49\times 10^{-11}$	$3.02\times 10^{-11}$	$3.02\times 10^{-11}$	$3.02\times 10^{-11}$
F4	$2.60\times 10^{-8}$	$1.43\times 10^{-5}$	$1.25\times 10^{-5}$	$2.01\times 10^{-4}$	$2.39\times 10^{-8}$	$7.09\times 10^{-8}$	$3.35\times 10^{-8}$	$3.02\times 10^{-11}$
F5	$2.61\times 10^{-10}$	$3.02\times 10^{-11}$	$1.73\times 10^{-7}$	$4.35\times 10^{-5}$	$1.56\times 10^{-8}$	$3.16\times 10^{-10}$	$4.50\times 10^{-11}$	$3.02\times 10^{-11}$
F6	$3.02\times 10^{-11}$	$3.02\times 10^{-11}$	$3.02\times 10^{-11}$	$3.02\times 10^{-11}$	$2.43\times 10^{-11}$	$3.02\times 10^{-11}$	$3.02\times 10^{-11}$	$3.02\times 10^{-11}$
F7	$7.39\times 10^{-11}$	$3.02\times 10^{-11}$	$5.60\times 10^{-7}$	$6.84\times 10^{-1}$	$5.46\times 10^{-9}$	$3.02\times 10^{-11}$	$3.02\times 10^{-11}$	$3.02\times 10^{-11}$
F8	$8.10\times 10^{-10}$	$4.08\times 10^{-11}$	$6.05\times 10^{-7}$	$1.00\times 10^{0}$	$1.85\times 10^{-8}$	$1.96\times 10^{-10}$	$1.29\times 10^{-9}$	$3.02\times 10^{-11}$
F9	$3.02\times 10^{-11}$	$3.02\times 10^{-11}$	$1.17\times 10^{-2}$	$3.02\times 10^{-11}$	$1.72\times 10^{-10}$	$3.02\times 10^{-11}$	$3.02\times 10^{-11}$	$3.02\times 10^{-11}$
F10	$8.35\times 10^{-8}$	$5.49\times 10^{-11}$	$3.96\times 10^{-8}$	$8.31\times 10^{-3}$	$3.69\times 10^{-11}$	$4.08\times 10^{-11}$	$1.07\times 10^{-9}$	$3.02\times 10^{-11}$
F11	$1.78\times 10^{-10}$	$3.02\times 10^{-11}$	$8.10\times 10^{-10}$	$8.84\times 10^{-7}$	$2.05\times 10^{-3}$	$3.02\times 10^{-11}$	$3.02\times 10^{-11}$	$3.02\times 10^{-11}$
F12	$1.01\times 10^{-8}$	$6.07\times 10^{-11}$	$2.15\times 10^{-10}$	$1.09\times 10^{-1}$	$1.02\times 10^{-1}$	$6.70\times 10^{-11}$	$3.02\times 10^{-11}$	$3.02\times 10^{-11}$
F13	$5.53\times 10^{-8}$	$1.86\times 10^{-9}$	$1.86\times 10^{-9}$	$6.77\times 10^{-5}$	$8.15\times 10^{-11}$	$2.92\times 10^{-9}$	$2.03\times 10^{-9}$	$7.12\times 10^{-9}$
F14	$3.92\times 10^{-2}$	$8.65\times 10^{-1}$	$1.70\times 10^{-2}$	$1.68\times 10^{-3}$	$5.19\times 10^{-7}$	$8.77\times 10^{-2}$	$1.05\times 10^{-1}$	$2.58\times 10^{-1}$
F15	$3.26\times 10^{-1}$	$1.03\times 10^{-6}$	$6.00\times 10^{-1}$	$5.49\times 10^{-1}$	$8.99\times 10^{-11}$	$6.28\times 10^{-6}$	$1.25\times 10^{-5}$	$4.12\times 10^{-6}$
F16	$7.60\times 10^{-7}$	$2.23\times 10^{-9}$	$1.41\times 10^{-4}$	$7.98\times 10^{-2}$	$9.05\times 10^{-2}$	$2.68\times 10^{-4}$	$4.35\times 10^{-5}$	$1.96\times 10^{-10}$
F17	$3.34\times 10^{-11}$	$3.02\times 10^{-11}$	$3.02\times 10^{-11}$	$7.30\times 10^{-4}$	$4.80\times 10^{-7}$	$3.02\times 10^{-11}$	$3.02\times 10^{-11}$	$3.02\times 10^{-11}$
F18	$3.52\times 10^{-7}$	$7.96\times 10^{-3}$	$1.47\times 10^{-7}$	$2.23\times 10^{-1}$	$3.02\times 10^{-11}$	$2.60\times 10^{-8}$	$1.70\times 10^{-8}$	$3.57\times 10^{-6}$
F19	$1.89\times 10^{-4}$	$3.52\times 10^{-7}$	$5.59\times 10^{-1}$	$2.15\times 10^{-2}$	$2.15\times 10^{-10}$	$4.69\times 10^{-8}$	$1.73\times 10^{-6}$	$8.15\times 10^{-5}$
F20	$3.02\times 10^{-11}$	$3.02\times 10^{-11}$	$3.02\times 10^{-11}$	$1.09\times 10^{-5}$	$4.64\times 10^{-3}$	$3.02\times 10^{-11}$	$3.02\times 10^{-11}$	$3.02\times 10^{-11}$
F21	$8.66\times 10^{-5}$	$1.49\times 10^{-6}$	$5.56\times 10^{-4}$	$9.63\times 10^{-2}$	$5.86\times 10^{-6}$	$5.01\times 10^{-2}$	$6.84\times 10^{-1}$	$1.27\times 10^{-2}$
F22	$5.60\times 10^{-7}$	$5.07\times 10^{-10}$	$2.57\times 10^{-7}$	$5.46\times 10^{-9}$	$2.32\times 10^{-2}$	$4.18\times 10^{-9}$	$7.77\times 10^{-9}$	$1.21\times 10^{-10}$
F23	$5.49\times 10^{-11}$	$3.02\times 10^{-11}$	$1.43\times 10^{-5}$	$2.50\times 10^{-3}$	$2.02\times 10^{-8}$	$1.61\times 10^{-10}$	$4.08\times 10^{-11}$	$3.02\times 10^{-11}$
F24	$8.84\times 10^{-7}$	$9.76\times 10^{-10}$	$8.31\times 10^{-3}$	$3.03\times 10^{-3}$	$1.17\times 10^{-5}$	$1.20\times 10^{-8}$	$2.68\times 10^{-6}$	$1.73\times 10^{-6}$
F25	$2.88\times 10^{-6}$	$2.38\times 10^{-7}$	$1.62\times 10^{-1}$	$1.78\times 10^{-4}$	$2.32\times 10^{-2}$	$5.19\times 10^{-7}$	$9.53\times 10^{-7}$	$3.02\times 10^{-11}$
F26	$8.29\times 10^{-6}$	$1.11\times 10^{-6}$	$1.22\times 10^{-2}$	$6.84\times 10^{-1}$	$2.46\times 10^{-1}$	$4.22\times 10^{-4}$	$6.97\times 10^{-3}$	$4.08\times 10^{-11}$
F27	$7.66\times 10^{-5}$	$5.49\times 10^{-11}$	$7.51\times 10^{-1}$	$1.34\times 10^{-5}$	$3.02\times 10^{-11}$	$5.53\times 10^{-8}$	$8.35\times 10^{-8}$	$3.02\times 10^{-11}$
F28	$2.49\times 10^{-6}$	$2.19\times 10^{-8}$	$6.10\times 10^{-3}$	$1.44\times 10^{-2}$	$3.59\times 10^{-4}$	$1.07\times 10^{-9}$	$2.67\times 10^{-9}$	$3.02\times 10^{-11}$
F29	$1.73\times 10^{-7}$	$3.34\times 10^{-11}$	$6.35\times 10^{-2}$	$6.52\times 10^{-1}$	$7.96\times 10^{-3}$	$4.31\times 10^{-8}$	$7.77\times 10^{-9}$	$3.02\times 10^{-11}$
F30	$7.28\times 10^{-1}$	$5.57\times 10^{-10}$	$6.38\times 10^{-3}$	$2.24\times 10^{-2}$	$3.02\times 10^{-11}$	$8.84\times 10^{-7}$	$6.36\times 10^{-5}$	$3.34\times 10^{-11}$
w/t/l	27/2/1	29/1/0	22/5/3	20/9/1	16/3/11	28/2/0	28/2/0	29/1/0

**Table 14 biomimetics-10-00620-t014:** Wilcoxon rank sum test of MSOA and other algorithms on CEC2017 (30D).

MSAO vs.	WOA	HHO	AOO	DOA	LSHADE	AO	LOBLAO	TEAO
F1	$3.02\times 10^{-11}$	$3.02\times 10^{-11}$	$3.99\times 10^{-4}$	$3.02\times 10^{-11}$	$6.67\times 10^{-3}$	$3.02\times 10^{-11}$	$3.02\times 10^{-11}$	$3.02\times 10^{-11}$
F2	$3.02\times 10^{-11}$	$3.02\times 10^{-11}$	$2.71\times 10^{-1}$	$1.87\times 10^{-5}$	$3.02\times 10^{-11}$	$3.02\times 10^{-11}$	$3.02\times 10^{-11}$	$3.02\times 10^{-11}$
F3	$1.71\times 10^{-1}$	$4.64\times 10^{-5}$	$3.02\times 10^{-11}$	$2.15\times 10^{-10}$	$3.16\times 10^{-10}$	$4.29\times 10^{-1}$	$5.20\times 10^{-1}$	$1.17\times 10^{-5}$
F4	$3.02\times 10^{-11}$	$3.02\times 10^{-11}$	$3.20\times 10^{-9}$	$2.39\times 10^{-8}$	$3.99\times 10^{-4}$	$3.02\times 10^{-11}$	$3.02\times 10^{-11}$	$3.02\times 10^{-11}$
F5	$3.02\times 10^{-11}$	$3.02\times 10^{-11}$	$3.69\times 10^{-11}$	$9.76\times 10^{-10}$	$3.02\times 10^{-11}$	$3.02\times 10^{-11}$	$3.02\times 10^{-11}$	$3.02\times 10^{-11}$
F6	$3.02\times 10^{-11}$	$3.02\times 10^{-11}$	$3.02\times 10^{-11}$	$3.02\times 10^{-11}$	$5.09\times 10^{-6}$	$3.02\times 10^{-11}$	$3.02\times 10^{-11}$	$3.02\times 10^{-11}$
F7	$3.02\times 10^{-11}$	$3.02\times 10^{-11}$	$5.57\times 10^{-10}$	$5.49\times 10^{-11}$	$3.02\times 10^{-11}$	$3.02\times 10^{-11}$	$3.02\times 10^{-11}$	$3.02\times 10^{-11}$
F8	$3.02\times 10^{-11}$	$3.02\times 10^{-11}$	$9.06\times 10^{-8}$	$2.23\times 10^{-9}$	$4.20\times 10^{-10}$	$3.02\times 10^{-11}$	$3.02\times 10^{-11}$	$3.02\times 10^{-11}$
F9	$3.02\times 10^{-11}$	$3.02\times 10^{-11}$	$4.18\times 10^{-9}$	$7.20\times 10^{-5}$	$3.47\times 10^{-10}$	$3.02\times 10^{-11}$	$3.02\times 10^{-11}$	$3.02\times 10^{-11}$
F10	$3.02\times 10^{-11}$	$3.02\times 10^{-11}$	$7.12\times 10^{-9}$	$4.74\times 10^{-6}$	$3.02\times 10^{-11}$	$3.02\times 10^{-11}$	$3.02\times 10^{-11}$	$3.02\times 10^{-11}$
F11	$3.02\times 10^{-11}$	$6.07\times 10^{-11}$	$1.61\times 10^{-10}$	$4.42\times 10^{-6}$	$1.25\times 10^{-4}$	$3.02\times 10^{-11}$	$3.02\times 10^{-11}$	$3.02\times 10^{-11}$
F12	$2.61\times 10^{-10}$	$5.49\times 10^{-11}$	$4.31\times 10^{-8}$	$4.03\times 10^{-3}$	$4.12\times 10^{-6}$	$3.02\times 10^{-11}$	$3.02\times 10^{-11}$	$3.02\times 10^{-11}$
F13	$3.02\times 10^{-11}$	$3.02\times 10^{-11}$	$3.34\times 10^{-11}$	$4.50\times 10^{-11}$	$2.20\times 10^{-7}$	$3.02\times 10^{-11}$	$3.02\times 10^{-11}$	$3.02\times 10^{-11}$
F14	$2.71\times 10^{-1}$	$1.17\times 10^{-5}$	$8.84\times 10^{-7}$	$4.64\times 10^{-1}$	$5.00\times 10^{-9}$	$4.98\times 10^{-4}$	$8.15\times 10^{-5}$	$8.35\times 10^{-8}$
F15	$3.02\times 10^{-11}$	$3.02\times 10^{-11}$	$3.02\times 10^{-11}$	$6.01\times 10^{-8}$	$3.35\times 10^{-8}$	$3.02\times 10^{-11}$	$3.02\times 10^{-11}$	$3.02\times 10^{-11}$
F16	$6.72\times 10^{-10}$	$3.02\times 10^{-11}$	$1.17\times 10^{-4}$	$6.35\times 10^{-2}$	$3.02\times 10^{-11}$	$4.98\times 10^{-11}$	$3.69\times 10^{-11}$	$3.02\times 10^{-11}$
F17	$2.02\times 10^{-8}$	$4.50\times 10^{-11}$	$3.63\times 10^{-1}$	$9.71\times 10^{-1}$	$6.52\times 10^{-9}$	$4.62\times 10^{-10}$	$2.60\times 10^{-8}$	$3.02\times 10^{-11}$
F18	$7.09\times 10^{-8}$	$1.25\times 10^{-4}$	$2.06\times 10^{-1}$	$6.73\times 10^{-1}$	$1.34\times 10^{-5}$	$9.26\times 10^{-9}$	$1.20\times 10^{-8}$	$3.69\times 10^{-11}$
F19	$3.69\times 10^{-11}$	$3.02\times 10^{-11}$	$3.02\times 10^{-11}$	$1.21\times 10^{-10}$	$1.56\times 10^{-8}$	$3.02\times 10^{-11}$	$3.02\times 10^{-11}$	$3.02\times 10^{-11}$
F20	$4.08\times 10^{-5}$	$4.57\times 10^{-9}$	$2.17\times 10^{-1}$	$3.04\times 10^{-1}$	$3.56\times 10^{-4}$	$2.53\times 10^{-4}$	$6.28\times 10^{-6}$	$3.34\times 10^{-11}$
F21	$3.02\times 10^{-11}$	$3.02\times 10^{-11}$	$4.31\times 10^{-8}$	$2.19\times 10^{-8}$	$3.69\times 10^{-11}$	$3.02\times 10^{-11}$	$6.72\times 10^{-10}$	$3.02\times 10^{-11}$
F22	$1.31\times 10^{-8}$	$2.38\times 10^{-7}$	$8.30\times 10^{-1}$	$1.12\times 10^{-1}$	$3.02\times 10^{-11}$	$2.71\times 10^{-2}$	$5.55\times 10^{-2}$	$3.02\times 10^{-11}$
F23	$3.02\times 10^{-11}$	$3.02\times 10^{-11}$	$2.15\times 10^{-10}$	$6.72\times 10^{-10}$	$2.92\times 10^{-9}$	$3.02\times 10^{-11}$	$3.02\times 10^{-11}$	$3.02\times 10^{-11}$
F24	$3.82\times 10^{-10}$	$3.02\times 10^{-11}$	$4.46\times 10^{-4}$	$1.78\times 10^{-10}$	$3.02\times 10^{-11}$	$3.02\times 10^{-11}$	$3.02\times 10^{-11}$	$3.02\times 10^{-11}$
F25	$3.02\times 10^{-11}$	$3.02\times 10^{-11}$	$3.09\times 10^{-6}$	$3.02\times 10^{-11}$	$2.24\times 10^{-2}$	$3.02\times 10^{-11}$	$3.02\times 10^{-11}$	$3.02\times 10^{-11}$
F26	$3.02\times 10^{-11}$	$1.10\times 10^{-8}$	$1.08\times 10^{-2}$	$3.99\times 10^{-4}$	$4.08\times 10^{-11}$	$1.73\times 10^{-6}$	$5.61\times 10^{-5}$	$3.02\times 10^{-11}$
F27	$3.02\times 10^{-11}$	$3.02\times 10^{-11}$	$2.39\times 10^{-8}$	$4.18\times 10^{-9}$	$3.02\times 10^{-11}$	$3.02\times 10^{-11}$	$3.02\times 10^{-11}$	$3.02\times 10^{-11}$
F28	$3.02\times 10^{-11}$	$3.02\times 10^{-11}$	$2.89\times 10^{-3}$	$8.35\times 10^{-8}$	$3.02\times 10^{-11}$	$3.02\times 10^{-11}$	$3.02\times 10^{-11}$	$3.02\times 10^{-11}$
F29	$3.02\times 10^{-11}$	$3.02\times 10^{-11}$	$6.01\times 10^{-8}$	$4.92\times 10^{-1}$	$5.49\times 10^{-11}$	$3.02\times 10^{-11}$	$3.02\times 10^{-11}$	$3.02\times 10^{-11}$
F30	$3.02\times 10^{-11}$	$3.02\times 10^{-11}$	$3.02\times 10^{-11}$	$7.12\times 10^{-9}$	$1.50\times 10^{-2}$	$3.02\times 10^{-11}$	$3.02\times 10^{-11}$	$3.02\times 10^{-11}$
w/t/l	28/2/0	29/1/0	22/5/3	22/7/1	24/0/6	28/1/1	28/2/0	30/0/0

**Table 15 biomimetics-10-00620-t015:** Wilcoxon rank sum test of MSOA and other algorithms on CEC2017 (50D).

MSAO vs.	WOA	HHO	AOO	DOA	LSHADE	AO	LOBLAO	TEAO
F1	$3.02\times 10^{-11}$	$3.02\times 10^{-11}$	$3.02\times 10^{-11}$	$3.02\times 10^{-11}$	$2.28\times 10^{-1}$	$3.02\times 10^{-11}$	$3.02\times 10^{-11}$	$3.02\times 10^{-11}$
F2	$3.02\times 10^{-11}$	$3.02\times 10^{-11}$	$3.63\times 10^{-1}$	$4.73\times 10^{-1}$	$3.02\times 10^{-11}$	$3.02\times 10^{-11}$	$3.02\times 10^{-11}$	$3.02\times 10^{-11}$
F3	$6.12\times 10^{-10}$	$1.41\times 10^{-9}$	$3.34\times 10^{-11}$	$5.49\times 10^{-11}$	$8.99\times 10^{-11}$	$2.62\times 10^{-3}$	$1.95\times 10^{-3}$	$3.52\times 10^{-7}$
F4	$3.02\times 10^{-11}$	$3.02\times 10^{-11}$	$4.69\times 10^{-8}$	$7.38\times 10^{-10}$	$1.30\times 10^{-3}$	$3.02\times 10^{-11}$	$3.02\times 10^{-11}$	$3.02\times 10^{-11}$
F5	$3.02\times 10^{-11}$	$3.02\times 10^{-11}$	$9.92\times 10^{-11}$	$3.69\times 10^{-11}$	$3.02\times 10^{-11}$	$3.02\times 10^{-11}$	$3.02\times 10^{-11}$	$3.02\times 10^{-11}$
F6	$3.02\times 10^{-11}$	$3.02\times 10^{-11}$	$3.02\times 10^{-11}$	$3.02\times 10^{-11}$	$1.68\times 10^{-4}$	$3.02\times 10^{-11}$	$3.02\times 10^{-11}$	$3.02\times 10^{-11}$
F7	$3.02\times 10^{-11}$	$3.02\times 10^{-11}$	$2.39\times 10^{-8}$	$3.02\times 10^{-11}$	$3.02\times 10^{-11}$	$3.02\times 10^{-11}$	$3.02\times 10^{-11}$	$3.02\times 10^{-11}$
F8	$3.02\times 10^{-11}$	$3.02\times 10^{-11}$	$8.10\times 10^{-10}$	$6.70\times 10^{-11}$	$3.02\times 10^{-11}$	$3.02\times 10^{-11}$	$3.02\times 10^{-11}$	$3.02\times 10^{-11}$
F9	$3.34\times 10^{-11}$	$3.02\times 10^{-11}$	$7.39\times 10^{-11}$	$9.26\times 10^{-9}$	$3.57\times 10^{-6}$	$3.02\times 10^{-11}$	$3.02\times 10^{-11}$	$3.02\times 10^{-11}$
F10	$3.02\times 10^{-11}$	$3.02\times 10^{-11}$	$2.44\times 10^{-9}$	$8.99\times 10^{-11}$	$3.02\times 10^{-11}$	$3.02\times 10^{-11}$	$3.02\times 10^{-11}$	$3.02\times 10^{-11}$
F11	$1.86\times 10^{-9}$	$7.74\times 10^{-6}$	$1.71\times 10^{-1}$	$8.68\times 10^{-3}$	$2.39\times 10^{-8}$	$1.61\times 10^{-10}$	$1.21\times 10^{-10}$	$3.02\times 10^{-11}$
F12	$3.02\times 10^{-11}$	$3.02\times 10^{-11}$	$5.07\times 10^{-10}$	$2.44\times 10^{-9}$	$3.02\times 10^{-11}$	$3.02\times 10^{-11}$	$3.02\times 10^{-11}$	$3.02\times 10^{-11}$
F13	$3.02\times 10^{-11}$	$3.02\times 10^{-11}$	$3.02\times 10^{-11}$	$3.02\times 10^{-11}$	$3.81\times 10^{-7}$	$3.02\times 10^{-11}$	$3.02\times 10^{-11}$	$3.02\times 10^{-11}$
F14	$2.75\times 10^{-3}$	$3.16\times 10^{-10}$	$1.60\times 10^{-3}$	$1.87\times 10^{-7}$	$8.50\times 10^{-2}$	$4.18\times 10^{-9}$	$1.21\times 10^{-10}$	$3.02\times 10^{-11}$
F15	$3.02\times 10^{-11}$	$3.02\times 10^{-11}$	$6.70\times 10^{-11}$	$3.02\times 10^{-11}$	$8.48\times 10^{-9}$	$3.02\times 10^{-11}$	$3.02\times 10^{-11}$	$3.02\times 10^{-11}$
F16	$2.15\times 10^{-10}$	$8.10\times 10^{-10}$	$3.37\times 10^{-4}$	$1.71\times 10^{-1}$	$3.02\times 10^{-11}$	$2.15\times 10^{-10}$	$4.08\times 10^{-11}$	$3.02\times 10^{-11}$
F17	$3.02\times 10^{-11}$	$3.02\times 10^{-11}$	$5.56\times 10^{-4}$	$1.76\times 10^{-2}$	$3.02\times 10^{-11}$	$2.15\times 10^{-10}$	$3.02\times 10^{-11}$	$3.02\times 10^{-11}$
F18	$1.11\times 10^{-6}$	$2.49\times 10^{-6}$	$8.77\times 10^{-2}$	$4.98\times 10^{-4}$	$2.44\times 10^{-9}$	$3.47\times 10^{-10}$	$2.61\times 10^{-10}$	$3.02\times 10^{-11}$
F19	$3.02\times 10^{-11}$	$3.02\times 10^{-11}$	$3.02\times 10^{-11}$	$3.02\times 10^{-11}$	$2.87\times 10^{-10}$	$3.02\times 10^{-11}$	$3.02\times 10^{-11}$	$3.02\times 10^{-11}$
F20	$4.50\times 10^{-11}$	$4.20\times 10^{-10}$	$4.80\times 10^{-7}$	$9.03\times 10^{-4}$	$3.02\times 10^{-11}$	$1.41\times 10^{-9}$	$4.57\times 10^{-9}$	$3.02\times 10^{-11}$
F21	$3.02\times 10^{-11}$	$3.02\times 10^{-11}$	$6.07\times 10^{-11}$	$4.08\times 10^{-11}$	$5.57\times 10^{-10}$	$3.02\times 10^{-11}$	$3.02\times 10^{-11}$	$3.02\times 10^{-11}$
F22	$3.02\times 10^{-11}$	$3.02\times 10^{-11}$	$1.39\times 10^{-6}$	$4.98\times 10^{-11}$	$3.02\times 10^{-11}$	$5.07\times 10^{-10}$	$5.57\times 10^{-10}$	$3.02\times 10^{-11}$
F23	$3.02\times 10^{-11}$	$3.02\times 10^{-11}$	$4.08\times 10^{-11}$	$3.02\times 10^{-11}$	$5.57\times 10^{-10}$	$3.02\times 10^{-11}$	$3.02\times 10^{-11}$	$3.02\times 10^{-11}$
F24	$3.02\times 10^{-11}$	$3.02\times 10^{-11}$	$2.92\times 10^{-9}$	$3.02\times 10^{-11}$	$3.02\times 10^{-11}$	$3.02\times 10^{-11}$	$3.02\times 10^{-11}$	$3.02\times 10^{-11}$
F25	$3.02\times 10^{-11}$	$3.02\times 10^{-11}$	$4.03\times 10^{-3}$	$5.49\times 10^{-11}$	$2.03\times 10^{-7}$	$3.02\times 10^{-11}$	$3.02\times 10^{-11}$	$3.02\times 10^{-11}$
F26	$3.02\times 10^{-11}$	$1.33\times 10^{-10}$	$1.07\times 10^{-7}$	$5.57\times 10^{-10}$	$3.02\times 10^{-11}$	$2.87\times 10^{-10}$	$3.34\times 10^{-11}$	$3.02\times 10^{-11}$
F27	$3.02\times 10^{-11}$	$3.02\times 10^{-11}$	$1.21\times 10^{-10}$	$1.61\times 10^{-10}$	$3.02\times 10^{-11}$	$3.02\times 10^{-11}$	$3.02\times 10^{-11}$	$3.02\times 10^{-11}$
F28	$3.02\times 10^{-11}$	$3.02\times 10^{-11}$	$1.86\times 10^{-1}$	$4.98\times 10^{-11}$	$1.86\times 10^{-1}$	$3.02\times 10^{-11}$	$3.02\times 10^{-11}$	$3.02\times 10^{-11}$
F29	$3.02\times 10^{-11}$	$3.02\times 10^{-11}$	$2.61\times 10^{-10}$	$2.02\times 10^{-8}$	$3.02\times 10^{-11}$	$3.02\times 10^{-11}$	$3.02\times 10^{-11}$	$3.02\times 10^{-11}$
F30	$3.02\times 10^{-11}$	$3.02\times 10^{-11}$	$3.02\times 10^{-11}$	$3.69\times 10^{-11}$	$3.95\times 10^{-1}$	$3.02\times 10^{-11}$	$3.02\times 10^{-11}$	$3.02\times 10^{-11}$
w/t/l	29/0/1	29/0/1	23/4/3	27/2/1	21/4/5	30/0/0	30/0/0	30/0/0

**Table 16 biomimetics-10-00620-t016:** Wilcoxon rank sum test of MSOA and other algorithms on CEC2017 (100D).

MSAO vs.	WOA	HHO	AOO	DOA	LSHADE	AO	LOBLAO	TEAO
F1	$3.02\times 10^{-11}$	$3.02\times 10^{-11}$	$3.67\times 10^{-3}$	$3.02\times 10^{-11}$	$3.02\times 10^{-11}$	$3.02\times 10^{-11}$	$3.02\times 10^{-11}$	$3.02\times 10^{-11}$
F2	$3.02\times 10^{-11}$	$3.02\times 10^{-11}$	$2.00\times 10^{-5}$	$6.10\times 10^{-3}$	$3.02\times 10^{-11}$	$3.02\times 10^{-11}$	$3.02\times 10^{-11}$	$3.02\times 10^{-11}$
F3	$3.85\times 10^{-3}$	$3.02\times 10^{-11}$	$1.96\times 10^{-10}$	$9.92\times 10^{-11}$	$5.49\times 10^{-11}$	$3.02\times 10^{-11}$	$3.02\times 10^{-11}$	$4.98\times 10^{-11}$
F4	$3.02\times 10^{-11}$	$3.02\times 10^{-11}$	$3.37\times 10^{-4}$	$3.02\times 10^{-11}$	$1.54\times 10^{-1}$	$3.02\times 10^{-11}$	$3.02\times 10^{-11}$	$3.02\times 10^{-11}$
F5	$3.02\times 10^{-11}$	$3.02\times 10^{-11}$	$9.92\times 10^{-11}$	$3.02\times 10^{-11}$	$3.02\times 10^{-11}$	$3.02\times 10^{-11}$	$3.02\times 10^{-11}$	$3.02\times 10^{-11}$
F6	$3.02\times 10^{-11}$	$3.02\times 10^{-11}$	$3.02\times 10^{-11}$	$3.02\times 10^{-11}$	$8.77\times 10^{-2}$	$3.02\times 10^{-11}$	$3.02\times 10^{-11}$	$3.02\times 10^{-11}$
F7	$3.02\times 10^{-11}$	$3.02\times 10^{-11}$	$3.69\times 10^{-11}$	$3.02\times 10^{-11}$	$7.38\times 10^{-10}$	$3.02\times 10^{-11}$	$3.02\times 10^{-11}$	$3.02\times 10^{-11}$
F8	$3.02\times 10^{-11}$	$3.02\times 10^{-11}$	$3.16\times 10^{-10}$	$3.02\times 10^{-11}$	$5.07\times 10^{-10}$	$3.02\times 10^{-11}$	$3.02\times 10^{-11}$	$3.02\times 10^{-11}$
F9	$4.08\times 10^{-11}$	$3.02\times 10^{-11}$	$5.55\times 10^{-2}$	$2.57\times 10^{-7}$	$1.17\times 10^{-5}$	$3.02\times 10^{-11}$	$3.02\times 10^{-11}$	$3.02\times 10^{-11}$
F10	$3.02\times 10^{-11}$	$3.02\times 10^{-11}$	$1.87\times 10^{-7}$	$3.02\times 10^{-11}$	$3.02\times 10^{-11}$	$3.02\times 10^{-11}$	$3.02\times 10^{-11}$	$3.02\times 10^{-11}$
F11	$4.98\times 10^{-11}$	$3.34\times 10^{-11}$	$3.34\times 10^{-11}$	$1.34\times 10^{-5}$	$3.02\times 10^{-11}$	$3.02\times 10^{-11}$	$3.02\times 10^{-11}$	$3.02\times 10^{-11}$
F12	$3.02\times 10^{-11}$	$3.02\times 10^{-11}$	$1.61\times 10^{-10}$	$4.08\times 10^{-11}$	$3.02\times 10^{-11}$	$3.02\times 10^{-11}$	$3.02\times 10^{-11}$	$3.02\times 10^{-11}$
F13	$3.02\times 10^{-11}$	$3.02\times 10^{-11}$	$3.69\times 10^{-11}$	$3.02\times 10^{-11}$	$3.96\times 10^{-8}$	$3.02\times 10^{-11}$	$3.02\times 10^{-11}$	$3.02\times 10^{-11}$
F14	$5.19\times 10^{-7}$	$3.83\times 10^{-5}$	$2.13\times 10^{-4}$	$3.77\times 10^{-4}$	$2.15\times 10^{-10}$	$3.69\times 10^{-11}$	$4.98\times 10^{-11}$	$3.02\times 10^{-11}$
F15	$3.02\times 10^{-11}$	$3.02\times 10^{-11}$	$3.02\times 10^{-11}$	$3.02\times 10^{-11}$	$4.44\times 10^{-7}$	$3.02\times 10^{-11}$	$3.02\times 10^{-11}$	$3.02\times 10^{-11}$
F16	$3.02\times 10^{-11}$	$3.02\times 10^{-11}$	$1.68\times 10^{-3}$	$5.27\times 10^{-5}$	$3.02\times 10^{-11}$	$3.02\times 10^{-11}$	$3.02\times 10^{-11}$	$3.02\times 10^{-11}$
F17	$4.08\times 10^{-11}$	$3.02\times 10^{-11}$	$1.47\times 10^{-7}$	$1.43\times 10^{-5}$	$3.02\times 10^{-11}$	$3.02\times 10^{-11}$	$3.02\times 10^{-11}$	$3.02\times 10^{-11}$
F18	$1.11\times 10^{-6}$	$6.38\times 10^{-3}$	$1.15\times 10^{-1}$	$7.70\times 10^{-4}$	$3.69\times 10^{-11}$	$5.19\times 10^{-7}$	$1.07\times 10^{-9}$	$3.02\times 10^{-11}$
F19	$3.02\times 10^{-11}$	$3.02\times 10^{-11}$	$3.02\times 10^{-11}$	$3.02\times 10^{-11}$	$4.21\times 10^{-2}$	$3.02\times 10^{-11}$	$3.02\times 10^{-11}$	$3.02\times 10^{-11}$
F20	$3.69\times 10^{-11}$	$3.02\times 10^{-11}$	$1.70\times 10^{-8}$	$1.60\times 10^{-7}$	$3.02\times 10^{-11}$	$3.34\times 10^{-11}$	$3.69\times 10^{-11}$	$3.02\times 10^{-11}$
F21	$3.02\times 10^{-11}$	$3.02\times 10^{-11}$	$8.89\times 10^{-10}$	$3.02\times 10^{-11}$	$3.02\times 10^{-11}$	$3.02\times 10^{-11}$	$3.02\times 10^{-11}$	$3.02\times 10^{-11}$
F22	$3.02\times 10^{-11}$	$3.02\times 10^{-11}$	$1.29\times 10^{-6}$	$3.02\times 10^{-11}$	$3.02\times 10^{-11}$	$3.02\times 10^{-11}$	$3.02\times 10^{-11}$	$3.02\times 10^{-11}$
F23	$3.02\times 10^{-11}$	$3.02\times 10^{-11}$	$3.02\times 10^{-11}$	$3.02\times 10^{-11}$	$8.15\times 10^{-11}$	$3.02\times 10^{-11}$	$3.02\times 10^{-11}$	$3.02\times 10^{-11}$
F24	$3.02\times 10^{-11}$	$3.02\times 10^{-11}$	$3.02\times 10^{-11}$	$3.02\times 10^{-11}$	$3.02\times 10^{-11}$	$3.02\times 10^{-11}$	$3.02\times 10^{-11}$	$3.02\times 10^{-11}$
F25	$3.02\times 10^{-11}$	$3.02\times 10^{-11}$	$4.08\times 10^{-5}$	$3.02\times 10^{-11}$	$1.73\times 10^{-7}$	$3.02\times 10^{-11}$	$3.02\times 10^{-11}$	$3.02\times 10^{-11}$
F26	$3.02\times 10^{-11}$	$3.02\times 10^{-11}$	$1.11\times 10^{-6}$	$3.02\times 10^{-11}$	$3.02\times 10^{-11}$	$3.02\times 10^{-11}$	$3.02\times 10^{-11}$	$3.02\times 10^{-11}$
F27	$3.02\times 10^{-11}$	$3.02\times 10^{-11}$	$3.02\times 10^{-11}$	$3.69\times 10^{-11}$	$3.02\times 10^{-11}$	$3.02\times 10^{-11}$	$3.02\times 10^{-11}$	$3.02\times 10^{-11}$
F28	$3.02\times 10^{-11}$	$3.02\times 10^{-11}$	$1.37\times 10^{-3}$	$3.02\times 10^{-11}$	$3.02\times 10^{-11}$	$3.02\times 10^{-11}$	$3.02\times 10^{-11}$	$3.02\times 10^{-11}$
F29	$3.02\times 10^{-11}$	$3.02\times 10^{-11}$	$1.09\times 10^{-10}$	$8.48\times 10^{-9}$	$3.02\times 10^{-11}$	$3.02\times 10^{-11}$	$3.02\times 10^{-11}$	$3.02\times 10^{-11}$
F30	$3.02\times 10^{-11}$	$3.02\times 10^{-11}$	$3.02\times 10^{-11}$	$9.92\times 10^{-11}$	$1.84\times 10^{-2}$	$3.02\times 10^{-11}$	$3.02\times 10^{-11}$	$3.02\times 10^{-11}$
w/t/l	29/0/1	29/0/1	24/2/4	28/0/2	25/2/3	29/0/1	29/0/1	29/0/1

**Table 17 biomimetics-10-00620-t017:** Friedman test rank test of MSOA and other algorithms on CEC2017 (10D).

	WOA	HHO	AOO	DOA	LSHADE	AO	LOBLAO	TEAO	MSAO
F1	4.07	6.57	2.67	4.97	1.17	7	7.37	9	2.2
F3	6.07	3.27	1.07	4.7	4.67	7.17	7.03	9	2.03
F4	5.9	5.03	3.67	2.93	4.7	5.6	6.6	9	1.57
F5	5.43	7.83	4.27	2.23	3.73	5.4	5.97	8.9	1.23
F6	4.9	8.13	4.03	3.2	1	6.3	6.67	8.77	2
F7	4.87	7.8	3.77	2.13	3.47	5.97	6.4	8.9	1.7
F8	5.93	6.93	4.67	1.83	4.37	5.37	5.43	8.93	1.53
F9	5.13	8.4	3.4	3.5	1	6.63	6.2	8.5	2.23
F10	3.8	6.3	4.67	2.03	6.67	5.67	5.3	8.93	1.63
F11	4.67	5.97	4.9	3.1	1.33	7.13	7.07	8.83	2
F12	4.63	6.3	5.23	2.33	1.97	6.73	6.7	8.9	2.2
F13	5.5	5.87	6.67	3.87	1	6.2	6.3	6.67	2.93
F14	3.27	5.2	3.37	7.6	1.2	6.97	6.83	5.07	5.5
F15	3.6	7.33	3.8	3.73	1	7.13	6.33	7.4	4.67
F16	5.27	7.33	4.4	4.03	1.67	5.2	5.9	8.4	2.8
F17	4.8	6.63	5.57	2.27	2.73	6.67	6.53	8.43	1.37
F18	5.97	5.63	6.67	3.1	1.07	7	6.87	5.9	2.8
F19	5.87	6.6	3.77	4.67	1	6.4	6.7	6.77	3.23
F20	4.7	7.83	4.8	2.57	2.4	6.83	6.33	8.13	1.4
F21	5.53	6.57	4.57	4.17	5.57	5.33	5.13	5.4	2.73
F22	4	7.1	3.6	4.57	2.03	6.17	6.47	8.93	2.13
F23	4.03	7.33	3.63	3.6	3.9	6.03	6.2	8.93	1.33
F24	5.6	8.13	3.43	3.8	4.13	6.03	5.53	6.4	1.93
F25	6.1	5.1	3.33	3.83	4.03	5.83	5.4	9	2.37
F26	5.97	6.37	3.5	3.43	4.2	5.53	5.27	8.53	2.2
F27	4.27	8.4	2.77	4.9	1	5.83	5.67	8.4	3.77
F28	5.1	6.07	3.7	3.2	3.83	5.87	6	8.83	2.4
F29	5.87	7.27	3.77	2.93	3.1	5.9	5.77	8.53	1.87
F30	5.03	6.87	4.57	4.23	1	5.77	5.77	8.53	3.23
Mean	5.03	6.70	4.08	3.57	2.72	6.20	6.20	8.13	2.38
Rank	5	8	4	3	2	6	7	9	1

**Table 18 biomimetics-10-00620-t018:** Friedman test rank test of MSOA and other algorithms on CEC2017 (30D).

	WOA	HHO	AOO	DOA	LSHADE	AO	LOBLAO	TEAO	MSAO
F1	7.17	4.93	1.43	3.93	2.7	6.73	7.1	9	2
F3	5.5	3.53	1	2	9	5.37	5.3	7.8	5.5
F4	5.83	5.77	3.33	3.4	1.2	6.97	7.3	9	2.2
F5	5.87	7.5	2.9	2.33	4.97	5.67	5.73	9	1.03
F6	6.03	7.7	4.07	3	1.07	6.33	5.87	9	1.93
F7	5	8	2.63	2.53	4	6.43	6.4	8.93	1.07
F8	6.53	6	2.8	2.37	6.87	5.17	5.07	9	1.2
F9	5.8	7.33	3.87	3.03	1	6.23	6.5	9	2.23
F10	5.57	5.47	3.03	2.37	8.03	5.13	5.37	8.9	1.13
F11	6.03	4.8	3.87	2.7	2.33	7.33	7.47	9	1.47
F12	5.67	5.43	3.37	2.2	3.9	7	7.03	9	1.4
F13	4.23	6.47	3.5	2.7	4.2	7.03	6.87	9	1
F14	5.53	7.03	2.3	4.4	1.43	6.2	6.37	7.67	4.07
F15	4.9	5.97	4.63	1.93	3.9	6.73	6.7	9	1.23
F16	4.73	6.77	2.57	2.4	6	5.9	5.93	9	1.7
F17	4.6	7.47	2.43	2.57	4.93	6.07	5.67	8.93	2.33
F18	4.7	5.7	2.9	2.73	4.9	6.27	6.77	8.6	2.43
F19	4.17	6.33	5.13	2.63	2.43	6.87	7.17	9	1.27
F20	5.4	6.83	3.6	2.53	4.57	5.5	5.2	8.7	2.67
F21	5.53	7.77	2.9	2.43	5.8	5	5.37	9	1.2
F22	5.8	6.07	4.27	2.93	8.7	2.83	2.9	8.23	3.27
F23	4.63	8	2.6	2.73	4.17	6.37	6.53	8.9	1.07
F24	4.1	8.03	2.27	3.83	4.83	5.87	5.8	8.97	1.3
F25	6.43	5.43	3.3	3.6	1.47	6.8	7.13	9	1.83
F26	5.7	7.17	3.1	3.3	5.03	5.13	4.43	9	2.13
F27	5.1	7.57	3.3	3.67	1.03	6.8	6.4	9	2.13
F28	6.07	5.27	2.03	2.67	4.23	7.2	7.13	9	1.4
F29	5.1	6.77	3.1	1.83	5.1	5.83	6.67	9	1.6
F30	5	5.9	4.63	2.63	1.53	7.17	7.3	9	1.83
Mean	5.40	6.45	3.13	2.81	4.11	6.14	6.19	8.85	1.92
Rank	5	8	3	2	4	6	7	9	1

**Table 19 biomimetics-10-00620-t019:** Friedman test rank test of MSOA and other algorithms on CEC2017 (50D).

	WOA	HHO	AOO	DOA	LSHADE	AO	LOBLAO	TEAO	MSAO
F1	6.43	4.63	1.23	4.2	2.7	7.27	7.3	9	2.23
F3	3.23	3.7	1.07	2.13	8.97	7.1	6.1	7	5.7
F4	6.1	5.2	3.1	3.63	1.33	7.47	7.2	9	1.97
F5	6.77	6.63	2.77	2.33	5.87	5.43	5.17	9	1.03
F6	6.13	7.3	4	2.97	1.37	6.4	6.17	9	1.67
F7	5.23	8.13	2.4	2.7	3.9	6.37	6.5	8.77	1
F8	6.23	5.97	2.77	2.23	6.2	5.9	5.7	9	1
F9	5.77	7.43	3.63	3.13	1.13	6.63	6.23	8.93	2.1
F10	5.7	5	2.7	2.87	8.33	5.13	5.57	8.67	1.03
F11	6.43	4.3	2.17	2.63	5.57	6.87	6.43	9	1.6
F12	5.73	5	3.03	2.2	4.63	7.3	7.07	9	1.03
F13	6.43	5.23	2.67	3.83	2.57	6.97	7.17	9	1.13
F14	4.3	6.23	1.3	4.17	3.43	6.97	6.53	8.83	3.23
F15	6.07	6.33	2.83	3.37	3.33	6.4	6.6	9	1.07
F16	4.7	5.83	2.7	2.07	7.83	5.3	5.87	9	1.7
F17	5.23	5.9	3.07	2.03	7.1	4.83	6.17	9	1.67
F18	4.47	4.67	2.27	3.23	6.4	7.03	6.07	8.87	2
F19	5.5	5.9	5.2	2.77	2.17	6.57	6.77	9	1.13
F20	6	6.1	3.27	2.3	8.13	4.63	4.4	8.57	1.6
F21	5.47	7.83	2.4	2.63	5.23	5.67	5.73	9	1.03
F22	5.03	5.6	2.5	2.53	8.07	5.63	5.63	8.93	1.07
F23	5.2	8	2.43	2.83	3.73	6.4	6.43	8.97	1
F24	4.33	8.17	2.17	3.4	4.27	6.5	6.3	8.83	1.03
F25	6.23	5.13	2.63	4.07	1.07	7.4	7.17	9	2.3
F26	5.8	7.4	2.6	3.1	4.23	5.97	5.57	9	1.33
F27	5.03	7.6	3.9	3.3	1	6.6	6.63	8.93	2
F28	6.2	5.13	2.7	3.93	1.27	7.43	7.23	9	2.1
F29	5	5.77	2.97	1.97	5.27	6.9	7	9	1.13
F30	5.63	5.47	4.13	2.63	1.97	7.27	7.33	9	1.57
Mean	5.53	6.05	2.78	2.94	4.38	6.43	6.35	8.87	1.67
Rank	5	6	2	3	4	8	7	9	1

**Table 20 biomimetics-10-00620-t020:** Friedman test rank test of MSOA and other algorithms on CEC2017 (100D).

	WOA	HHO	AOO	DOA	LSHADE	AO	LOBLAO	TEAO	MSAO
F1	6.43	5	1.4	3.9	3.03	7.37	7.2	9	1.67
F3	6.73	1.63	4.97	5.03	9	3.03	3.27	3.6	7.73
F4	5.9	5.03	2.23	3.67	2.73	7.53	7.4	9	1.5
F5	6.77	5.87	2.13	2.87	4.83	6.37	6.17	9	1
F6	6.23	7.2	4	2.8	1.83	6.2	6.37	9	1.37
F7	5.17	8.07	2.9	2.97	3.13	6.43	6.4	8.93	1
F8	5.9	6.9	2.2	2.8	4.1	6.5	6.6	9	1
F9	6	7.2	2.8	3.6	1.37	6.6	6.27	8.83	2.33
F10	5.2	5.33	1.87	3.4	8.43	5.6	5.47	8.57	1.13
F11	4.47	4.57	1.1	2	8.27	6.9	6.8	8	2.9
F12	5.97	4.53	2.73	2.37	4.63	7.4	7.37	9	1
F13	6.93	5	2.3	3.9	2.73	7.17	6.9	9	1.07
F14	4.63	3.83	1.33	3.4	7.8	6.43	6.33	8.87	2.37
F15	6.57	5.2	2.7	3.83	2.43	7.03	7.03	9	1.2
F16	5.1	4.5	2.47	2.13	7.77	6.4	6.2	9	1.43
F17	4.63	4.63	2.73	2.13	7.2	6.9	6.47	9	1.3
F18	4.57	3.73	1.5	4.4	7.67	5.67	6.1	8.87	2.5
F19	7.03	5.23	3.97	2.97	1.77	6.93	6.8	9	1.3
F20	5.73	5.57	2.9	2.73	8.6	4.57	5.27	8.33	1.3
F21	5.03	7.37	2.07	3.03	3.97	6.73	6.8	9	1
F22	4.73	5.4	2	3.2	8.43	5.8	5.83	8.57	1.03
F23	5	8	2.97	2.17	3.87	6.47	6.57	8.97	1
F24	5	7.93	2.83	2.73	3.43	6.67	6.4	9	1
F25	6.3	4.97	2.1	3.53	2.97	7.27	7.43	9	1.43
F26	5.03	6.47	2.3	2.97	3.63	7.17	7.3	9	1.13
F27	4.93	6.6	3.57	3.5	1	7.3	7.1	9	2
F28	6.17	5.07	2.6	4	1	7.53	7.23	9	2.4
F29	5.27	5.03	2.9	1.97	4.9	7.3	7.5	9	1.13
F30	6.13	5.07	4.07	2.9	1.4	7.3	7.43	9	1.7
Mean	5.64	5.55	2.61	3.13	4.55	6.57	6.55	8.71	1.69
Rank	6	5	2	3	4	8	7	9	1

**Table 21 biomimetics-10-00620-t021:** Optimization results for tension/compression spring design problem.

Algorithm	Design Variable			Optimal Value	Rank
	$x_{1}$	$x_{2}$	$x_{3}$		
WOA	0.05	0.317221	14.05509	0.012733	4
HHO	0.051346	0.32855	14.07487	0.013924	5
AOO	0.05	0.317403	14.03077	0.012721	3
DOA	0.060519	0.60568	4.349242	0.014085	6
LSHADE	0.052559	0.377994	10.144	0.012681	2
AO	0.061821	0.644598	4.407184	0.015784	8
LOBLAO	0.062055	0.632119	4.405829	0.015593	7
TEAO	0.0654	0.785028	2.714485	0.01583	9
MSAO	0.051124	0.343278	12.12271	0.012671	1

**Table 22 biomimetics-10-00620-t022:** Optimization results for pressure vessel design problem.

Algorithm	Design Variable				Optimal Value	>Rank
	$z_{1}$	$z_{2}$	$x_{3}$	$x_{4}$		
WOA	0.81426	0.40257	42.18945	175.5317	5950.671	4
HHO	1.096096	0.554776	56.79018	52.19098	6755.636	7
AOO	0.781801	0.386498	40.5078	197.4321	5892.49	3
DOA	1.231969	0.60835	63.66243	16.93532	7209.205	8
LSHADE	0.734706	0.376211	38.06765	233.9139	5848.727	2
AO	1.258394	0.630418	62.38845	24.53348	7644.994	9
LOBLAO	0.85456	0.486076	44.13857	153.5328	6282.705	5
TEAO	0.937033	0.450725	47.24577	121.8173	6307.177	6
MSAO	0.738992	0.365349	38.28975	230.325	5821.851	1

**Table 23 biomimetics-10-00620-t023:** Optimization results for three-bar truss design problem.

Algorithm	Design Variable		Optimal Value	Rank
	$x_{1}$	$x_{2}$		
WOA	0.787405	0.411854	263.8973	4
HHO	0.799045	0.379682	263.9721	7
AOO	0.787313	0.412114	263.8972	3
DOA	0.789191	0.406828	263.8998	5
LSHADE	0.787337	0.412047	263.8972	2
AO	0.777693	0.441216	264.0865	8
LOBLAO	0.805812	0.362841	264.2021	9
TEAO	0.788682	0.408895	263.9626	6
MSAO	0.788675	0.408248	263.8958	1

**Table 24 biomimetics-10-00620-t024:** Optimization results for welded beam design problem.

Algorithm	Design Variable				Optimal Value	Rank
	$x_{1}$	$x_{2}$	$x_{3}$	$x_{4}$		
WOA	0.202974	3.288993	9.044687	0.20569	1.697121	3
HHO	0.128243	9.052858	9.014212	0.206754	2.231482	5
AOO	0.238953	2.88184	8.471793	0.238971	1.826056	4
DOA	0.389963	2.091398	6.572499	0.390919	2.340397	7
LSHADE	0.203559	3.280222	9.036589	0.205731	1.695725	2
AO	0.13532	9.833804	8.816318	0.219159	2.414442	8
LOBLAO	0.126397	8.058519	8.35209	0.246175	2.324208	6
TEAO	0.384237	2.470936	5.788766	0.52854	2.827475	9
MSAO	0.205728	3.234919	9.036825	0.205729	1.692794	1

**Table 25 biomimetics-10-00620-t025:** Optimization results for speed reducer design problem.

Algorithm	Design Variable							Optimal Value	Rank
	$x_{1}$	$x_{2}$	$x_{3}$	$x_{4}$	$x_{5}$	$x_{6}$	$x_{7}$		
WOA	3.500364	0.7	17	7.3	7.744215	3.35024	5.286705	2995.287	4
HHO	3.50624	0.7	17	7.346998	7.967452	3.436743	5.286741	3025.564	6
AOO	3.500001	0.7	17	7.556774	7.898512	3.350962	5.286717	3000.988	5
DOA	3.50039	0.7	17	7.3	7.718313	3.350814	5.287142	2995.152	3
LSHADE	5.635129	0.603753	13.91125	6.933583	7.716773	3.355721	5.287968	2592.06	2
AO	3.6	0.7	17	8.3	7.990018	3.483046	5.314859	3102.403	8
LOBLAO	3.6	0.7	17	8.3	7.87192	3.698459	5.340921	3180.6	9
TEAO	3.596694	0.700001	17.00006	7.559103	8.3	3.350707	5.286859	3047.838	7
MSAO	8.395648	0.699651	9.834684	7.022819	7.718347	3.362394	5.289249	2500.976	1

## Data Availability

The original contributions presented in the study are included in the article. Further inquiries can be directed to the corresponding authors first.

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
