# Peer review of "Improved Multi-Strategy Aquila Optimizer for Engineering Optimization Problems"

_biomimetics, 2025, doi:10.3390/biomimetics10090620_

Round 1
Reviewer 1 Report
Comments and Suggestions for Authors
The proposed Multi-Strategy Aquila Optimizer (MSAO) introduces interesting hybrid techniques to enhance the original AO algorithm.
However, to strengthen the quality and clarity of the paper, the following points are recommended for revision:
- The literature review requires expansion to provide a more comprehensive analysis, especially regarding different variants of the AO
algorithm. A sub-paragraph should be added to introduce the research gap and justify the need for a new variant. Additionally, a brief
summary of the proposed method should be included at the end of the Introduction section.
- Please revise the subheadings in Section 2. For instance, "2.2" should correspond to the "Initialization" step, "2.3" to "Expanded
Exploration", and continue consistently up to "2.6".
- Provide appropriate references for all governing equations, especially those adapted or inspired by existing algorithms.
- The paper compares MSAO only with two AO variants. Several recent AO-based methods should also be considered as additional baselines
for a more robust and fair comparison.
- While the authors describe MSAO as an improved version of AO, it appears to be more of a hybrid with the DOA algorithm, or at least
strongly influenced by it. A brief description of DOA should be included, along with a clear explanation of how MSAO differs from it—
preferably illustrated with a flowchart.
- To present an unbiased view, the paper should include a discussion on the potential limitations or drawbacks of the proposed MSAO
method.
Author Response
Comments 1: The literature review requires expansion to provide a more comprehensive analysis, especially regarding different variants of the AO algorithm. A sub-paragraph should be added to introduce the research gap and justify the need for a new variant. Additionally, a brief summary of the proposed method should be included at the end of the Introduction section.
Response 1: Thank you for pointing this out. We agree with this comment. Accordingly, we have expanded the introduction of the main variants of the Aquila Optimizer (AO) (page 3 and lines 106-117) conducted a more comprehensive comparative analysis, and added a paragraph that details the shortcomings of current AO variants (pages 3-4 and lines 119-141)), analyzes their causes, and explains how the proposed MSAO variant addresses these deficiencies. Additionally, we have added a brief summary at the end of the Introduction, concisely outlining the core ideas, primary innovations, and anticipated contributions of the proposed method (page 4 and lines 144-151).
Comments 2: Please revise the subheadings in Section 2. For instance, "2.2" should correspond to the "Initialization" step, "2.3" to "Expanded Exploration", and continue consistently up to "2.6".
Response 2: Thank you for pointing out the issue with the section numbering. We agree with this comment. We have adjusted it accordingly and ensured that all section numbers are now correct. We have accordingly revised the subsection headings and reorganized all section titles. The specific changes can be found in the revised manuscript at page 4, line 181 (2.1.1. Initialization); page 5, lines 192 (2.1.2. Expanded exploration) and 200 (2.1.3. Narrowed exploration); and page 6, lines 216 (2.1.4. Expanded exploitation), 223 (2.1.5. Narrowed exploitation), and 230. (2.1.6. Algorithm procedure)
Comments 3: Provide appropriate references for all governing equations, especially those adapted or inspired by existing algorithms.
Response 3: Thank you for pointing out the adjustments needed in the references. We agree with this comment. We have accordingly added appropriate references for all governing equations.
Comments 4: The paper compares MSAO only with two AO variants. Several recent AO-based methods should also be considered as additional baselines for a more robust and fair comparison.
Response 4: Thank you for your valuable comments. We agree with this comment. We have added the comparison of AO variants as requested, introducing Section 4.4 “Performance Analysis of MSAO and AO’s variants in CEC2017” (page 19, line 478-492). Table 2 in Section 4 “Experimental Results and Analysis” now includes the parameter settings for each algorithm. In Section 4.4, we compare MSAO with five AO variants across all test dimensions of CEC2017 (10D, 30D, 50D, and 100D) and provide an in-depth analysis of the results. The findings indicate that, out of the 29 benchmark functions, MSAO achieves the lowest mean fitness in most cases: 25 functions (86%) in 10D, 22 functions (76%) in 30D, 23 functions (79%) in 50D, and 28 functions (97%) in 100D. Moreover, MSAO consistently attains the best average ranking, with scores of 1.34, 1.28, 1.34, and 1.03 in these dimensions, respectively. These results clearly demonstrate the superior performance of MSAO compared to other AO variants.
Comments 5: While the authors describe MSAO as an improved version of AO, it appears to be more of a hybrid with the DOA algorithm, or at least strongly influenced by it. A brief description of DOA should be included, along with a clear explanation of how MSAO differs from it—preferably illustrated with a flowchart.
Response 5: Thank you for your insightful comment. We agree with this comment. To present the Dream Optimization Algorithm (DOA) more clearly, we have optimized the structure of our headings by grouping the introductions to AO and DOA under the “Related Work” section. Specifically, we have added a new subsection “2.2 Overview of the DOA” (pages 7-8, line 253-271), which briefly describes the DOA algorithm workflow, its core strategies, and their roles, and included a new flowchart of DOA (Figure 1). Furthermore, to highlight the distinctions between DOA and the proposed MSAO, we have added a discussion at the end of Section 3.4 (page 12, line 384-392), where we compare their workflows and strategies using the respective flowcharts.
Comments 6: To present an unbiased view, the paper should include a discussion on the potential limitations or drawbacks of the proposed MSAO method.
Response 6: Thank you for your comments. We agree with this comment. In the Conclusion (page 49, lines 661-667), we have provided a more detailed analysis and discussion of the proposed algorithm’s limitations. We observed that MSAO exhibits suboptimal performance on certain hybrid functions, which may stem from the strict division of exploration and exploitation into two separate phases governed by a fixed threshold. In hybrid functions, MSAO cannot leverage a flexible phase-transition mechanism, potentially leading to premature phase changes or excessive focus on one region and thereby overlooking other promising areas of the search space. Addressing this issue will be a primary focus of our future work to enhance the algorithm’s performance on complex hybrid functions.
Reviewer 2 Report
Comments and Suggestions for Authors
Abstract
The abstract must clearly present the key numerical findings that validate the effectiveness of the proposed approach. Without this, the abstract lacks impact and fails to justify the study’s significance.
Literature Support (Lines 28–39)
The references cited in this section are not effectively integrated into the discussion. They appear to be included for formality rather than to substantiate the claims made. This undermines the credibility of the argument. You must expand on the methodologies referenced and critically evaluate their relevance to your work. A superficial citation approach is unacceptable in scholarly writing.
Mathematical Rigor and Notation
The mathematical expressions lack clarity in distinguishing between scalar and vector quantities. For instance, Equation (3) should be in vector form and formatted in bold to reflect this. This issue persists throughout the manuscript and must be addressed comprehensively to avoid ambiguity and ensure mathematical precision.
Equations (6) and (7) are particularly problematic. The use of dot and cross products appears arbitrary and unjustified. Equation (6), for example, should likely be expressed as a dot product. You must rigorously verify the mathematical operations used throughout the paper and ensure they are contextually and mathematically sound.
Section 2.7 – Methodological Clarity
This section lacks sufficient detail. Merely directing the reader to pseudocode is inadequate. The procedures must be explicitly described and justified. Pseudocode is a supplementary tool, not a substitute for a thorough methodological explanation.
Algorithm 1 – Parameter Justification
The constants 2/3 and 0.5 used in Algorithm 1 are introduced without any justification. This raises serious concerns about the empirical or theoretical basis for these values. You must explain their origin and rationale to establish the credibility of your algorithm.
Section 4 – Results and Analysis
The title of Section 4 should be revised to "Computational Results and Analysis" to better reflect its content. While comparative studies are presented, the criteria for comparison are vague, and the numerical results lack physical interpretation. It is essential to explain what these numbers represent and how they support your conclusions. Without this, the analysis remains superficial.
Section 4.4 – Interpretation Deficiency
This section presents numerical outputs without any meaningful interpretation. The absence of detailed discussion renders the results ineffective. You must provide a thorough analysis of what these values imply in the context of your study.
Table 17 and Tables 18–21 – Ranking and Redundancy Issues
Table 17 attempts to rank algorithm performance in a spring design problem, but the ranking methodology is unclear and unconvincing. Furthermore, Tables 18–21 appear to present similar problems without adequate differentiation. This deficiency must be addressed, and a compelling justification for their inclusion is required.
Conclusion – Lack of Depth
The conclusion is overly generic and fails to deliver a strong closing argument. It should clearly summarize the comparative performance of the algorithms and highlight the key numerical findings. A conclusion that merely restates general observations does not meet the standard expected in technical research.
Author Response
Comments 1: Abstract
The abstract must clearly present the key numerical findings that validate the effectiveness of the proposed approach. Without this, the abstract lacks impact and fails to justify the study’s significance.
Response 1: Thank you for pointing this out. We agree with this comment. We have revised the abstract to include the most significant validation results: MSAO outperform 8 state-of-the-art algorithms, including CEC-winning and enhanced AO variants, achieving the best optimization results on 55%, 69%, 69% and 72% of the benchmark functions, respectively, which demonstrates its outstanding performance. (page 1, lines 13–16 of the revised manuscript). These findings underscore the substantial contributions and practical significance of our work.
Comments 2: Literature Support (Lines 28–39)
The references cited in this section are not effectively integrated into the discussion. They appear to be included for formality rather than to substantiate the claims made. This undermines the credibility of the argument. You must expand on the methodologies referenced and critically evaluate their relevance to your work. A superficial citation approach is unacceptable in scholarly writing.
Response 2: Thank you for highlighting this issue. We agree with this comment. We have systematically reorganized the referenced literature in this section into five major application areas: computational intelligence and data mining, transportation, task planning, resource management, and UAV path planning. We provide a detailed overview of how current metaheuristic algorithms are applied in each of these fields (pages 1–2, lines 31–58).
Comments 3: Mathematical Rigor and Notation
The mathematical expressions lack clarity in distinguishing between scalar and vector quantities. For instance, Equation (3) should be in vector form and formatted in bold to reflect this. This issue persists throughout the manuscript and must be addressed comprehensively to avoid ambiguity and ensure mathematical precision.
Equations (6) and (7) are particularly problematic. The use of dot and cross products appears arbitrary and unjustified. Equation (6), for example, should likely be expressed as a dot product. You must rigorously verify the mathematical operations used throughout the paper and ensure they are contextually and mathematically sound.
Response 3: Thank you for pointing out the issues in the mathematical expressions. We agree with this comment. We have expanded and refined the content of Equation 6 (page 6, lines 208–210) by adding a detailed derivation and comprehensive notation explanations and adjusted all mathematical expressions to ensure greater rigor and clarity. Specifically, we distinguished between scalars and vectors by using boldface for all vector quantities. We also clarified the multiplication operations: the symbol “×” denotes multiplication between scalars or between a scalar and a vector, while “*” indicates element-wise multiplication between vectors. This mathematical representation is now noted at the beginning of each algorithm description section (page 4, lines 178–180 and page 8, lines 279–281).
Comments 4: Section 2.7 – Methodological Clarity
This section lacks sufficient detail. Merely directing the reader to pseudocode is inadequate. The procedures must be explicitly described and justified. Pseudocode is a supplementary tool, not a substitute for a thorough methodological explanation.
Response 4: Thank you for your careful review and valuable feedback on the references section. We agree with this comment. In Section 2.1.7 (pages 6-7, lines 232–243), we have provided a clearer and more detailed description of the AO workflow, elucidating the specific role and significance of each step in enhancing overall algorithm performance. Following your suggestion, we have similarly refined and expanded the description of the MSAO workflow in Section 3.4 (page 11, lines 364–382), explaining in detail how each process component works together to improve the algorithm’s effectiveness.
Comments 5: Algorithm 1 – Parameter Justification
The constants 2/3 and 0.5 used in Algorithm 1 are introduced without any justification. This raises serious concerns about the empirical or theoretical basis for these values. You must explain their origin and rationale to establish the credibility of your algorithm.
Response 5: Thank you for your insightful suggestion. We agree with this comment. We have provided a more comprehensive explanation of the two parameters in Algorithm 1 (page 7, lines 244–252), clarifying their origin, underlying principles, and specific roles within the algorithm, thereby further enhancing its credibility.
Comments 6: Section 4 – Results and Analysis
The title of Section 4 should be revised to "Computational Results and Analysis" to better reflect its content. While comparative studies are presented, the criteria for comparison are vague, and the numerical results lack physical interpretation. It is essential to explain what these numbers represent and how they support your conclusions. Without this, the analysis remains superficial.
Response 6: Thank you for your valuable comments. We agree with this comment. We have renamed Section 4 to “Computational Results and Analysis” (page 13, line 401) and clearly defined the comparison criteria and their meanings within this section. We have provided detailed interpretations of all the numerical results and expounded the specific conclusions they support. To further improve the manuscript’s quality, we have optimized and adjusted all experiments in Section 4 (see each subsection for specific changes), added illustrative explanations to some tables to clarify the significance of each experiment, and reorganized the experimental results with comprehensive statistical descriptions, emphasizing how these findings demonstrate the superior performance of MSAO.
Comments 7: Section 4.4 – Interpretation Deficiency
This section presents numerical outputs without any meaningful interpretation. The absence of detailed discussion renders the results ineffective. You must provide a thorough analysis of what these values imply in the context of your study.
Response 7: Thank you for your careful review and constructive suggestions. We agree with this comment. We have revised and refined the analysis in Section 4.4 (pages 33-37, lines 498–541), providing a more detailed discussion and a comprehensive analysis of the significance of these numeric values in the context of our study.
Comments 8: Table 17 and Tables 18–21 – Ranking and Redundancy Issues
Table 17 attempts to rank algorithm performance in a spring design problem, but the ranking methodology is unclear and unconvincing. Furthermore, Tables 18–21 appear to present similar problems without adequate differentiation. This deficiency must be addressed, and a compelling justification for their inclusion is required.
Response 8: Thank you for identifying this issue. We agree with this comment. We have comprehensively reviewed all engineering problems presented in Section 4.7. In the opening paragraph (page 43, lines 559–566), we have added a more detailed description of engineering design problem and clarified the criteria and methods used for performance comparison. Subsequently, in each subsection (page 44, lines 572–581; page 45, lines 587–595; page 46, lines 602–609 ; pages 47-48, lines 615–624 and page 48, lines 628–635), we have provided deeper discussions of the experimental results and conclusions, included additional numerical results, and conducted a thorough analysis of these data to better demonstrate the superior performance of MSAO on the respective engineering problems.
Comments 9: Lack of Depth
The conclusion is overly generic and fails to deliver a strong closing argument. It should clearly summarize the comparative performance of the algorithms and highlight the key numerical findings. A conclusion that merely restates general observations does not meet the standard expected in technical research.
Response 9: Thank you for your valuable comments on the Conclusion section. We agree with this comment. We have provided a more detailed exposition in the Conclusion (page 49, lines 647–660). By further summarizing and analyzing the most representative numerical results from our experiments, we present a clear demonstration of MSAO’s balanced performance and superiority across different problem scales. We elaborate on each key metric to reinforce the argument for MSAO’s excellent performance, thereby providing robust quantitative support for the validity of our conclusions.
Round 2
Reviewer 1 Report
Comments and Suggestions for Authors
I appreciate the authors' efforts in revising the manuscript. They have carefully considered my comments and provided satisfactory responses to my concerns.
Reviewer 2 Report
Comments and Suggestions for Authors
I am happy with the correction being made.